# PerturbDiff: Functional Diffusion for Single-Cell Perturbation Modeling

Xinyu Yuan [1 2 *]  Xixian Liu [1 2 *]  Yashi Zhang [1 2 *]  Zuobai Zhang [1 2]  Hongyu Guo [3 4]  Jian Tang [1 5 6 †]

## Abstract

Building *Virtual Cells* that can accurately simulate cellular responses to perturbations is a long-standing goal in systems biology. A fundamental challenge is that high-throughput single-cell sequencing is destructive: the same cell cannot be observed both before and after a perturbation. Thus, perturbation prediction requires mapping unpaired control and perturbed populations. Existing models address this by learning maps between distributions, but typically assume a single fixed response distribution when conditioned on observed cellular context (*e.g.*, cell type) and the perturbation type. In reality, responses vary systematically due to unobservable latent factors such as microenvironmental fluctuations and complex batch effects, forming a *manifold* of possible distributions for the same observed conditions. To account for this variability, we introduce PerturbDiff, which shifts modeling from individual cells to entire distributions. By embedding distributions as points in a Hilbert space, we define a diffusion-based generative process operating directly over probability distributions. This allows PerturbDiff to capture population-level response shifts across hidden factors. Benchmarks on established datasets show that PerturbDiff achieves state-of-the-art performance in single-cell response prediction and generalizes substantially better to unseen perturbations. See our project page, where code and data are publicly available.

## 1. Introduction

The human body is an ensemble of cells, each evolutionarily shaped to respond to disturbances. Recent advances in high-throughput single-cell perturbation sequencing have enabled systematic characterization of cellular responses to controlled interventions (Zhang et al., 2025). This gives a powerful tool to study relationships between diverse perturbations and phenotypic outcomes, including immune signaling through cytokines (Biosciences, 2023), drug responses for therapeutic discovery (Zhang et al., 2025), and genetic modifications to uncover gene function (Nadig et al., 2025).

Nevertheless, although modern platforms can assay hundreds or even thousands of perturbation conditions, the space of possible interventions—spanning genes, drugs, dosages, and cellular contexts—grows combinatorially and remains far beyond what can be explored exhaustively. As a result, experimental measurements remain costly and labor-intensive. This gap creates a strong need for computational methods that generalize beyond observed conditions to predict cellular responses. In response, *virtual cell models* (Bunne et al., 2024) have emerged as a new frontier at the intersection of artificial intelligence and biology. These models aim to simulate cellular state changes under unseen perturbations by predicting transcriptomic gene expressions.

Early methods, such as linear models (Ahlmann-Eltze et al., 2024) and foundation models like scGPT (Cui et al., 2024), often rely on performing regression between randomly paired control and perturbed cells. This paradigm is fundamentally limited because single-cell sequencing is destructive: the same cell cannot be observed both before and after perturbation. With no true cell-to-cell correspondence in the data, random pairing drives models to learn an average response across cells, thereby failing to capture cellular heterogeneity. To address this, more recent methods shift towards learning transitions between control and perturbed distributions. For example, STATE (Adduri et al., 2025) aligns populations using a kernel-based objective, CellFlow (Klein et al., 2025) uses flow matching, and Squidiff (He et al., 2025) leverages diffusion-based modeling. While effective, these methods commonly assume that, conditioned on observed cellular context, the control or perturbed cell distribution is fixed (Fig. 1(a)). In reality, however, unobserved latent factors, such as microenvironmental fluctuations and complex batch effects, introduce systematic variability that reshapes the underlying cell distributions (Fig. 1(b)). By overlooking this distribution variability, current methods tend to predict a static response distribution that marginal-

---

*Equal contribution; the first author led the project, and the other two authors are ordered alphabetically. †Corresponding author. [1]Mila - Québec AI Institute [2]University of Montréal [3]University of Ottawa [4]National Research Council of Canada [5]HEC Montréal [6]CIFAR AI Chair. Correspondence to: Jian Tang <tangjian@mila.quebec>.

*Proceedings of the 43rd International Conference on Machine Learning*, Seoul, South Korea. PMLR 306, 2026. Copyright 2026 by the author(s).

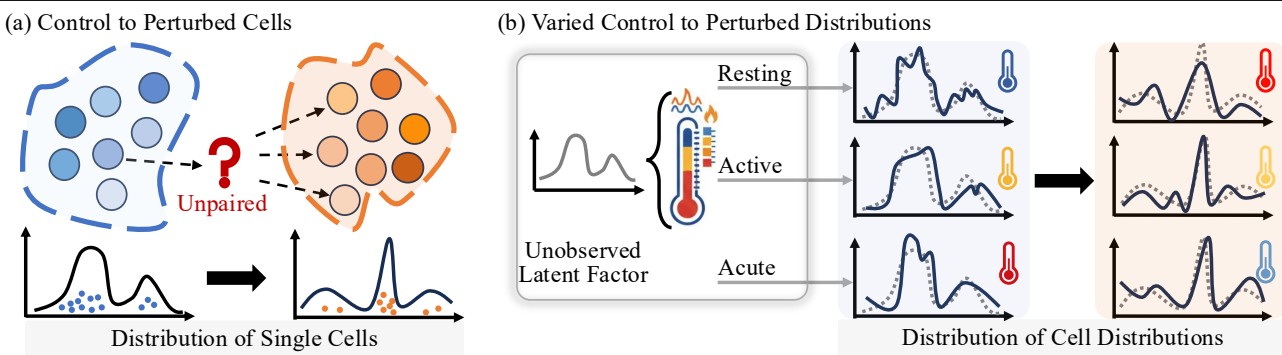

*Figure 1.* **Distributional variability in single-cell perturbation data.** (a) Traditional methods operate on unpaired control and perturbed cells, learning to map a control cell distribution to a perturbed one. (b) However, variations in cell distributions arise from unobserved latent factors, inducing a family of distinct cell distributions and shifting the objective to learning a distribution over cell distributions.

izes over these unobserved latents, limiting generalization to unseen perturbations or cellular contexts.

To address this limitation, we introduce PerturbDiff. By treating the population distribution itself as a random variable, PerturbDiff models a family of plausible response distributions, corresponding to variability induced by unobserved latent factors. In particular, as illustrated in Fig. 2, we first embed cell distributions into a reproducing kernel Hilbert space (RKHS) (Berlinet & Thomas-Agnan, 2011), where each distribution is represented by its kernel mean embedding as a single point in this function space. We then define a diffusion process directly over these function-valued representations, yielding a principled generative framework at the distribution level rather than the level of individual cells, as in existing methods. Importantly, this construction naturally induces a Maximum Mean Discrepancy (MMD) objective for diffusion training, providing a direct and well-founded measure for aligning source and target cell populations. Compared to pointwise MSE-based losses in standard diffusions, the resulting MMD loss is better suited to sparse, high-dimensional single-cell features. Empirically, our method achieves state-of-the-art performance on large-scale signaling, drug and genetic perturbation benchmarks, including PBMC (Biosciences, 2023), Tahoe100M (Zhang et al., 2025) and Replogle (Nadig et al., 2025), demonstrating strong and well-balanced results across 14 diverse metrics (see Fig. 3).

Furthermore, to address the scarcity of perturbed data, such as limited cell types, we introduce a novel pretraining strategy that learns marginal single-cell distributions by integrating perturbation data with large-scale unperturbed RNA-seq data, thereby covering a broader range of cellular states. Empirically, the pretrained model exhibits non-trivial zero-shot predictive capability, and fine-tuning consistently outperforms training from scratch when data is scarce.

## 2. Preliminary

### 2.1. Notations: Cells, Populations, and Distributions

We consider a fixed set of genes $\mathcal{G}$, with size $|\mathcal{G}|$ ranging from thousands to tens of thousands. A *single cell* is represented by its gene expression vector $\mathbf{x} \in \mathbb{R}^{|\mathcal{G}|}$, where each dimension corresponds to the expression of one gene. The set of all possible cell profiles constitutes the *cell space* $\mathcal{X} \subset \mathbb{R}^{|\mathcal{G}|}$. A finite collection of cells $X = \{\mathbf{x}_1, \ldots, \mathbf{x}_N\}$ is referred to as a *cell population*. We use $P$ to denote a *probability distribution* over $\mathcal{X}$, and let $\mathcal{P}(\mathcal{X})$ be the space of all such distributions.

### 2.2. Problem Formulation

In perturbation datasets, we observe two cell populations: the control cells $X_{\text{ctrl}}$ and the perturbed cells $X_{\text{pert}}$. Cells in $X_{\text{ctrl}}$ and $X_{\text{pert}}$ are unpaired: no cell-to-cell correspondence is assumed. Given $X_{\text{ctrl}}$, the goal of perturbation modeling is to predict the cellular response after perturbation.

Both $X_{\text{ctrl}}$ and $X_{\text{pert}}$ are associated with *cellular context* labels $c$, which describes *observable* experimental and biological conditions such as cell type, donor, experimental batch, or their combinations. $X_{\text{pert}}$ is additionally associated with *perturbation* labels $\tau$ (*e.g.*, a cytokine, drug, or genetic intervention). We use $P_c$ and $P_{c,\tau}$ to denote the (unknown) cell distributions conditioned on $c$ and $(c, \tau)$, respectively.

### 2.3. Diffusion Models

Diffusion models (Ho et al., 2020; Song et al., 2021b) are a family of generative models that learn data distributions from empirical samples by reversing a fixed forward Markov chain $q(\mathbf{x}_t | \mathbf{x}_{t-1})$ that gradually corrupts data $\mathbf{x}_0 \sim p_{\text{data}}$. The transition kernel at timestep $t \in \{1, \ldots, T\}$ admits the closed form: $q(\mathbf{x}_t | \mathbf{x}_0) = \mathcal{N}(\mathbf{x}_t; \gamma_t \mathbf{x}_0, \sigma_t^2 \mathbf{I})$, where $\gamma_t$ and $\sigma_t$ define the noise schedule. One way to parameterize the reverse process is to reconstruct the clean sample $\mathbf{x}_0$ from its noisy state $\mathbf{x}_t$ (Salimans & Ho, 2022), given time $t$ and all the condition information $\eta$, trained via the simplified denoising objective (Ho et al., 2020):

$$\mathbb{E}_{\mathbf{x}_0 \sim p_{\text{data}}, \varepsilon \sim \mathcal{N}(0, \mathbf{I}), t} \left[ \left\| \mathbf{x}_0 - \mathbf{x}_\theta(\mathbf{x}_t, t, \eta) \right\|_2^2 \right]. \quad (1)$$

To strengthen conditional fidelity, classifier-free guidance (CFG) (Ho & Salimans, 2022) is widely used. The model is trained with a mixture of conditional and unconditional inputs by replacing $\eta$ with a null token $\varnothing$ with probability $p_{\text{drop}}$. At sampling time, the prediction is steered toward the condition by linearly extrapolating between the conditional and unconditional estimates: $\hat{\mathbf{x}}_\theta(\mathbf{x}_t, t, \eta) = (1 + w)\,\mathbf{x}_\theta(\mathbf{x}_t, t, \eta) - w\,\mathbf{x}_\theta(\mathbf{x}_t, t, \varnothing)$, where $w \geq 0$ controls the guidance strength.

## 3. Related Work

Single-cell perturbation prediction has been extensively studied. Early methods perform deterministic cell-wise mapping via random pairing, differing mainly in modeling design. Linear models (Ahlmann-Eltze et al., 2024) regress perturbed expressions, offering a simple yet strong baseline. GEARS (Roohani et al., 2024) incorporates gene–gene interaction graphs. scGPT (Cui et al., 2024) finetunes pretrained cell foundation models. Due to random pairing, these models are driven to capture average perturbation effects. Probabilistic conditional generative models treat single cells as random variables, capturing single-cell stochasticity. scGen (Lotfollahi et al., 2019) models perturbation as latent shifts in a VAE, CPA (Lotfollahi et al., 2023) further disentangles perturbation and biological covariates in latent space. Squidiff (He et al., 2025) applies diffusion models to capture nonlinear effects. Population-level methods avoid random pairing by directly mapping control and perturbed distributions. CellOT (Bunne et al., 2023) uses an optimal transport framework. STATE (Adduri et al., 2025) aligns populations via a kernel-based discrepancy objective. CellFlow (Klein et al., 2025) learns a continuous-time flow via vector fields, while Unlasting (Chi et al., 2025) adopts diffusion bridges to model stochastic transitions between distributions.

Most existing methods assume a single perturbed distribution $P_{c,\tau}$ conditioned on observed context $c$ and perturbation type $\tau$, implicitly marginalizing variability from unobserved factors. In contrast, we directly model the distribution-level variability driven by latent factors (Fig. 1(b)), by learning a diffusion process directly over the space of cell distributions.

## 4. Method

We present PerturbDiff, a diffusion framework over cell distributions for perturbation prediction (Fig. 2). We model *distributional variability* by treating cell distributions as random variables (Sec. 4.1), represent them via kernel mean embeddings (Sec. 4.2), and define diffusion over these embeddings (Sec. 4.3) with a tractable training and sampling scheme (Sec. 4.4). We then describe the model architecture (Sec. 4.5), a pretraining strategy using large-scale single-cell atlases (Sec. 4.6), and conclude with a discussion (Sec. 4.7).

### 4.1. Motivation

We frame perturbation prediction as a conditional diffusion-based generative modeling problem. A key design choice in diffusion is the definition of the *random variable* on which the stochastic process is defined. Most existing methods take a single-cell state $\mathbf{x} \in \mathcal{X}$ as the random variable and learn a conditional distribution over cells (He et al., 2025) (Klein et al., 2025). Under this formulation, for a given observed condition $(c, \tau)$, these methods model a *single* perturbed cell distribution $P_{c,\tau}$ (Fig. 1(a)). In practice, however, cell samples collected under the same $(c, \tau)$ can be influenced by unobservable latent factors, such as microenvironmental fluctuations and complex batch effects. These latents induce variability at the *distribution level*, corresponding to a family of plausible response distributions associated with the same $(c, \tau)$ (Fig. 1(b)). Existing methods do not model this *distributional variability*, but collapse heterogeneous responses into a single distribution $P_{c,\tau}$, potentially limiting generalization to unseen conditions.

**Cell distribution as random variable.** To capture such distributional variability, we shift the random variable from individual cells to cell populations. Concretely, instead of assuming a single perturbed distribution $P_{c,\tau}$, we consider the perturbed population to be a random variable $D_{c,\tau}$ taking values in $\mathcal{P}(\mathcal{X})$. The realizations of $D_{c,\tau}$ are cell distributions induced by unobserved latents. Analogously, we denote the corresponding control distribution-valued random variable by $D_c$. Under this view, $D_{c,\tau}$ captures the family of possible perturbed distributions observed under the same condition $(c, \tau)$; the commonly learned target $P_{c,\tau}$ corresponds to the expectation of this random variable $\mathbb{E}[D_{c,\tau}]$.

**Goal.** We aim to model the conditional law of this perturbed distribution-valued random variable conditioned on the control one, with $\mathcal{F}_\theta$ defining a diffusion-based generative process on the space of distributions:

$$D_{c,\tau} \sim \mathcal{F}_\theta(\cdot \mid D_c, c, \tau). \quad (2)$$

### 4.2. Representing Cell Distributions

**Empirical distributions from cell batches.** While $D_{c,\tau}$ is conceptually a distribution-valued random variable supported on $\mathcal{P}(\mathcal{X})$, it is not directly observable. In practice, experiments under a fixed condition $(c, \tau)$ provide only *finite batches of cells*, each offering a partial and noisy view of the underlying perturbed population. Importantly, different batches might be implicitly influenced by different latent factors. Given a perturbed cell batch $B_{\text{pert}} = \{\mathbf{x}_1, \ldots, \mathbf{x}_m\}$, we construct the empirical distribution $\tilde{P}_{\text{pert}} := \frac{1}{m} \sum_{j=1}^{m} \delta_{\mathbf{x}_j}$, which serves as a finite-sample Monte-Carlo estimate of a realization of $D_{c,\tau}$. Note that $\delta_{\mathbf{x}}$ is the Dirac delta distribution at point $\mathbf{x}$. Analogously, batches of control cells yield empirical distributions $\tilde{P}_{\text{ctrl}}$, corresponding to a realization of the control variable $D_c$.

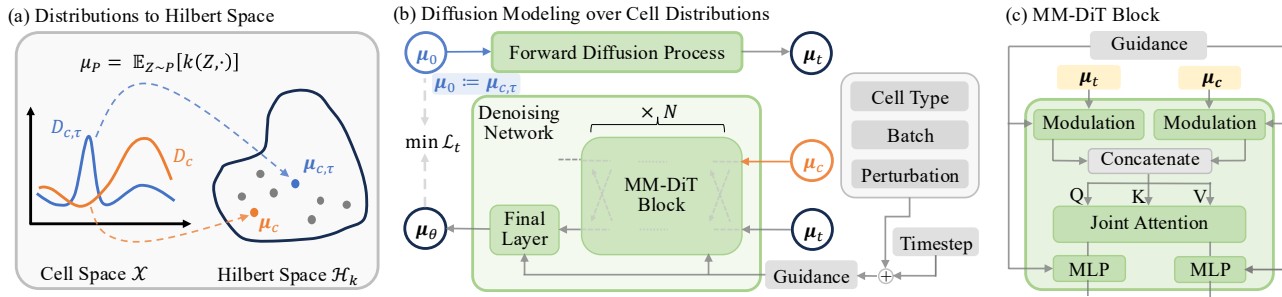

*Figure 2.* **Overview of the PerturbDiff framework.** (a) Distribution-valued random variables $D_{c,\tau}$ and $D_{c,\tau}$ in cell space are mapped to Hilbert-space elements $\boldsymbol{\mu}_{c,\tau}$ and $\boldsymbol{\mu}_c \in \mathcal{H}_k$ via kernel mean embedding. (b) Diffusion is defined on perturbed embeddings $\boldsymbol{\mu}_0 := \boldsymbol{\mu}_{c,\tau}$, with a denoising network predicting the target $\boldsymbol{\mu}_\theta$. (c) Each MM-DiT block performs joint attention over control and perturbed token streams.

For a given condition $(c, \tau)$, the collections $\{\tilde{P}_{\text{pert}}\}$ and $\{\tilde{P}_{\text{ctrl}}\}$ provide observed samples of the distribution-valued objects $D_{c,\tau}$ and $D_c$, respectively. These empirical distributions form the data used to learn our generative model.

**Hilbert space representation of cell distributions.** To enable a diffusion process over distributions, we embed probability distributions into a reproducing kernel Hilbert space (RKHS) (Berlinet & Thomas-Agnan, 2011) $\mathcal{H}_k$ induced by a positive-definite kernel $k : \mathcal{X} \times \mathcal{X} \to \mathbb{R}$. Each probability distribution $P \in \mathcal{P}(\mathcal{X})$ is mapped to its *kernel mean embedding* $\boldsymbol{\mu}_P := \mathbb{E}_{Z \sim P}[k(Z, \cdot)] \in \mathcal{H}_k$, which is well-defined under mild conditions (see App. B.1.2). This mapping places distributions as points in a well-behaved function space, enabling operations between distributions such as interpolation and discrepancy directly in $\mathcal{H}_k$. We summarize the key properties below.

*Remark* 4.1 (**Geometric Properties**). For distributions $P, Q$ on $\mathcal{X}$, kernel mean embeddings satisfy: **(i)** linearity: $\boldsymbol{\mu}_{\alpha P + (1-\alpha)Q} = \alpha \boldsymbol{\mu}_P + (1-\alpha)\boldsymbol{\mu}_Q$; **(ii)** kernel geometry: $\langle \boldsymbol{\mu}_P, \boldsymbol{\mu}_Q \rangle_{\mathcal{H}_k} = \mathbb{E}_{Z \sim P, Z' \sim Q}[k(Z, Z')]$; **(iii)** induced distance: $\|\boldsymbol{\mu}_P - \boldsymbol{\mu}_Q\|_{\mathcal{H}_k}^2 = \text{MMD}_k^2(P, Q)$, where MMD denotes Maximum Mean Discrepancy (Borgwardt et al., 2006).

Applying the embedding to $D_{c,\tau}$ and $D_c$ yields random elements $\boldsymbol{\mu}_{D_{c,\tau}}$ and $\boldsymbol{\mu}_{D_c}$ taking values in $\mathcal{H}_k$, which define the state space for our diffusion model (denoted $\boldsymbol{\mu}_{c,\tau}$ and $\boldsymbol{\mu}_c$ for brevity). Accordingly, the observed data for training, which is originally a collection of empirical cell distributions $\{\tilde{P}_{\text{pert}}\}$ and $\{\tilde{P}_{\text{ctrl}}\}$, is mapped into kernel mean embeddings in $\mathcal{H}_k$, $\{\boldsymbol{\mu}_{\tilde{P}_{\text{pert}}}\}$ and $\{\boldsymbol{\mu}_{\tilde{P}_{\text{ctrl}}}\}$, respectively. We discuss tractable training and sampling of $\boldsymbol{\mu}_{c,\tau}$ and $\boldsymbol{\mu}_c$ in Sec. 4.4.

### 4.3. Diffusion Modeling on Cell Distributions

**Forward diffusion process over cell distributions.** We construct a diffusion model directly in the RKHS $\mathcal{H}_k$, designed to mirror the standard DDPM (Ho et al., 2020), but operating on *kernel mean embeddings of cell distributions* rather than on finite-dimensional vectors. The forward diffusion process starts from the perturbed embedding $\boldsymbol{\mu}_0 := \boldsymbol{\mu}_{c,\tau}$, and progressively injects noise to transform it

towards a Gaussian-like reference measure[1] in $\mathcal{H}_k$.

In Euclidean space, sampling noise from Gaussian distributions is tractable and straightforward. However, $\mathcal{H}_k$ is infinite-dimensional. *Gaussian random elements*, which are generalized Gaussian random variables that take values in $\mathcal{H}_k$, are easy to characterize with a mean element and covariance operator (see App. B.1.4).

**Definition 4.2** (Forward diffusion process). Let $\{\beta_t\}_{t=1}^T \subset (0, 1)$ be a variance schedule and let $C : \mathcal{H}_k \to \mathcal{H}_k$ be a self-adjoint, positive semi-definite, trace-class covariance operator. We define the forward Markov chain

$$\boldsymbol{\mu}_t = \sqrt{1 - \beta_t}\boldsymbol{\mu}_{t-1} + \sqrt{\beta_t}\,\Xi_t, \quad \Xi_t \sim \mathcal{N}_{\mathcal{H}_k}(0, C), \quad (3)$$

where each $\Xi_t$ is a Gaussian random element in $\mathcal{H}_k$ following the Gaussian measure $\mathcal{N}_{\mathcal{H}_k}(0, C)$ (App. B.1.5). We discuss tractable sampling of $\Xi_t$ in Sec. 4.4.

By closure of Gaussian measures under affine transformation and summation (Lem. B.7 and Lem. B.8) in $\mathcal{H}_k$, the forward process remains Gaussian across steps $t$, yielding the closed-form marginal

$$\boldsymbol{\mu}_t \mid \boldsymbol{\mu}_0 \sim \mathcal{N}_{\mathcal{H}_k}\left(\sqrt{\alpha_t}\boldsymbol{\mu}_0, (1 - \alpha_t)C\right), \quad (4)$$

where $\alpha_t := \prod_{s=1}^t (1 - \beta_s)$. Thus, this establishes a diffusion process over cell distributions, seamlessly extending the DDPM framework to $\mathcal{H}_k$.

**Conditional reverse process and variational objective.** Given the forward diffusion defined above, the true reverse conditional $P_{t-1|t,0}(\boldsymbol{\mu}_{t-1}|\boldsymbol{\mu}_t, \boldsymbol{\mu}_0)$ admits a closed-form Gaussian measure in $\mathcal{H}_k$, whose mean is a linear combination of $\boldsymbol{\mu}_t$ and $\boldsymbol{\mu}_0$ and whose covariance is determined by the forward noise schedule (detailed in Prop. B.10).

Directly sampling from this conditional distribution is intractable since $\boldsymbol{\mu}_0$ is unknown during generation. As in DDPM, we therefore adopt a variational formulation and parameterize the reverse process using a learnable mean function $\boldsymbol{\mu}_\theta(\cdot)$ (details in App. B.1.6). This yields a simplified denoising objective, expressed in the RKHS norm:

---

[1] A *measure* refers to a probability distribution, not a density.

$$\mathcal{L}_t \propto ||\boldsymbol{\mu}_0 - \boldsymbol{\mu}_\theta(\boldsymbol{\mu}_t, t)||^2_{\mathcal{H}_k}. \tag{5}$$

To model perturbation responses, we extend the unconditional reverse process above to a conditional one by incorporating the control embedding $\boldsymbol{\mu}_c$, and the covariates $(c, \tau)$. We implement this via classifier-free guidance (Ho & Salimans, 2022), which leads to a conditional denoising objective (full derivations in App. B.1.6):

$$\mathcal{L}_t \propto \left|\left|\boldsymbol{\mu}_0 - \boldsymbol{\mu}_\theta(\boldsymbol{\mu}_t, t, \boldsymbol{\mu}_c, c, \tau)\right|\right|^2_{\mathcal{H}_k}. \tag{6}$$

This reverse process enables stochastic, conditional generation of perturbed cell distributions, while remaining well-defined in the infinite-dimensional RKHS.

### 4.4. Training and Sampling

**Distribution-aware loss via cell batches.** Our training objective (Eqn. (6)) is defined in $\mathcal{H}_k$, which is infinite-dimensional and therefore intractable. We thus instantiate all *kernel mean embedding random elements* appearing in the objective (namely $\boldsymbol{\mu}_0, \boldsymbol{\mu}_c$, and $\boldsymbol{\mu}_\theta$) via empirical cell batches. Recall that $\boldsymbol{\mu}_0$ is the perturbed kernel mean embedding $\boldsymbol{\mu}_{c,\tau}$. A perturbed batch $B_{\text{pert}}$ of size $m$ yields a finite-sample realization of this random element by inducing an empirical distribution $\tilde{P}_{\text{pert}}$, whose empirical kernel mean embedding $\boldsymbol{\mu}_{\tilde{P}_{\text{pert}}}$ acts as a Monte Carlo estimate of $\boldsymbol{\mu}_{c,\tau}$, converging as $m \to \infty$. Analogously, the control embedding $\boldsymbol{\mu}_c$ is instantiated from a control batch $B_{\text{ctrl}}$. And we let a neural network output a batch of predicted perturbed cells $B_\theta = \{\mathbf{x}^\theta_1, \ldots, \mathbf{x}^\theta_m\}$, which is associated with $\tilde{P}_\theta$, to model $\boldsymbol{\mu}_\theta$. Leveraging the RKHS distance property in Rem. 4.1, the reverse objective admits an exact equivalence:

$$||\boldsymbol{\mu}_0 - \boldsymbol{\mu}_\theta(\boldsymbol{\mu}_t, t, \boldsymbol{\mu}_{\mathbf{P}_c}, c, \tau)||^2_{\mathcal{H}_k} = \text{MMD}^2_k(\tilde{P}_{\text{pert}}, \tilde{P}_\theta). \tag{7}$$

As a result, training reduces to matching the predicted and real cell populations via MMD, providing a distribution-aware objective beyond pointwise cell-wise reconstruction. Network details for constructing $B_\theta$ are deferred to Sec. 4.5.

**Tractable approximation of Gaussian noise in RKHS.** Our diffusion process is defined over kernel mean embeddings in $\mathcal{H}_k$, which requires adding Gaussian random elements $\Xi_t$ during the forward noising process. Direct sampling from function-valued Gaussian measures is intractable. Therefore, we construct a tractable approximation by injecting additive Gaussian noise independently to each cell in the original space and recomputing the empirical kernel mean embeddings. As proved in App. B.1.7, under a first-order linearization of the kernel feature map, this procedure induces an $\mathcal{H}_k$-valued Gaussian random element whose covariance operator can be characterized explicitly (Prop. B.14). This yields a theoretically grounded and tractable realization of the Gaussian random element required for diffusion noising.

**Multi-scale training objective.** In practice, we use a hybrid loss combining the distribution-aware loss (Eqn. (7)) with

a standard DDPM loss that treats single cells as random variables to make our framework more compact:

$$\mathcal{L}_{\text{total}} = \text{MMD}^2_k(\tilde{P}_{\text{pert}}, \tilde{P}_\theta) + \frac{1}{m} \sum_{j=1}^m ||\mathbf{x}_j - \mathbf{x}^\theta_j||^2_2. \tag{8}$$

Empirically, we observe that optimization is dominated by the MMD term and the MSE term adds simple regularization on the global batch centroid (see Sec. 5.5).

**Sampling.** Since $\mathcal{H}_k$ is intractable, we perform deterministic DDIM (Song et al., 2020) sampling directly in cell space, consistent with our training procedure.

### 4.5. Architecture Design

As shown in Fig. 2(c), our denoising network jointly encodes the perturbed and control cell batches $(B_{\text{pert}}, B_{\text{ctrl}})$ via a Multi-Modal Diffusion Transformer (MM-DiT) (Esser et al., 2024), which maintains two parallel token streams that interact via joint attention in each block. This design represents perturbation effects as structured deviations from the control distribution. The timestep $t$ and covariate $(c, \tau)$ embeddings are combined and injected into every block via adaptive LayerNorm with zero initialization (AdaLN-Zero) (Peebles & Xie, 2023), following standard DiT-style conditioning. Architectural details are provided in App. B.3.

### 4.6. Marginal Pretraining as a Prior

While perturbation datasets are often limited in cell type and experimental batch diversity, large-scale single-cell atlases span a much broader range of cell states across both observable and latent contexts. Although unperturbed, these data capture rich population structures and biologically plausible cell states, providing a broad prior over the cell manifold.

Motivated by this, we introduce a *marginal pretraining* stage in PerturbDiff, where the model first learns the unperturbed cell distribution conditioned solely on cellular context $c$,

$$D_c \sim \mathcal{F}_\theta(\cdot \mid c), \tag{9}$$

before being finetuned on perturbation data to model perturbation-induced distributional shifts. As in Eqn. (2), $\mathcal{F}_\theta$ denotes the diffusion process defined in $\mathcal{H}_k$.

This stage uses all available training cells from perturbation datasets spanning dozens of cell types, alongside 61 million cells across hundreds of cell types from single-cell RNA-seq data curated in CellxGene (Program et al., 2025). This two-stage paradigm improves data efficiency and generalization, particularly when perturbation data are scarce.

### 4.7. Discussion

**Relationship with function space diffusion.** Existing functional diffusion models (Kerrigan et al., 2022) aim to generate a *single continuous function* (*e.g.*, audio waves) from

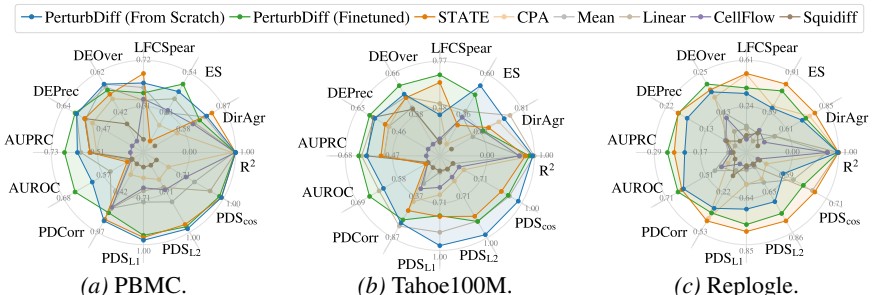

*(a) PBMC.*     *(b) Tahoe100M.*     *(c) Replogle.*

*Figure 3.* **Perturbation prediction results across methods, metrics, and datasets.** Each axis represents a performance metric, with higher values indicating better performance.

*Table 1.* **Data statistics.** We report the number of cells (#Cells), perturbations (#Pert.), cell types (#CT.), and experimental batches (#Batches) for each dataset, highlighting data diversity. See detailed data analyses in App. A.4, including zero-expression sparsity, sample imbalance, and distribution shifts.

| Dataset | #Cells | #Pert. | #CT. | #Batches |
|---|---|---|---|---|
| PBMC | 9.7M | 90 | 18 | 12 |
| Tahoe100M | 101.2M | 1,137 | 50 | 14 |
| Replogle | 0.6M | 2,023 | 4 | 56 |
| CellxGene | 60.9M | / | 662 | 10,887 |

finite and noisy observations. In contrast, our method generates *cell distributions*, which are represented by kernel mean embeddings in RKHS, to solve perturbation predictions. Although both methods define diffusion processes in Hilbert space, they fundamentally differ in their modeling targets and the specific spaces they operate in (*i.e.*, Sobolev spaces *v.s.* RKHS). As a result, diffusion formulations, derivations, and training objectives are inherently different.

**Relationship with STATE.** STATE (Adduri et al., 2025), as an emerging and increasingly adopted method, uses MMD for population matching to train transformers that align control and perturbed cell populations. Our work offers a fundamentally different modeling perspective. Rather than treating population matching as a deterministic objective, we formulate perturbation prediction as learning a *stochastic distribution over cell distributions*. In our framework, MMD is not externally imposed, but arises naturally from the equivalence property of squared RKHS distance between kernel mean embeddings, which coincides with MMD by construction. We also discuss a score-matching (Song et al., 2021a) perspective to interpret our MMD in App B.4.

## 5. Experiment

### 5.1. Experimental Setup

**Downstream and pretraining datasets.** We evaluate perturbation prediction on three widely used benchmark datasets following STATE, including signaling (PBMC), drug (Tahoe100M), and genetic (Replogle) perturbations. All tasks predict perturbed gene expressions over 2,000 highly variable genes (HVGs), with preprocessing and test splits following STATE (App. A.2). For pretraining, we combine all training cells from these three perturbation datasets with 61 million single-cell RNA-seq cells from CellxGene (statistics in Tab. 1). To enable cross-dataset pretraining, we unify gene spaces with a merge-then-select strategy (App. A.3.2), yielding 12,626 informative genes with high overlap across datasets (Tab. 3; >98% overlap for PBMC and Tahoe100M, >45% for Replogle and Cellx-Gene), enabling effective knowledge transfer.

**Training Configuration.** We report two variants of our method: *PerturbDiff (From Scratch)* (trained from scratch) and *PerturbDiff (Finetuned)* (pretrained then finetuned). Unless otherwise specified in App. C.1, all training stages (training from scratch, pretraining and finetuning) share the same diffusion pipeline, optimizer, training and sampling procedures, and checkpoint selection. Diffusion-specific settings are detailed in App. B.2.

**Baselines.** We compare against diverse strong baselines, including the widely used *mean* baseline (Kernfeld et al., 2025) and a *linear* model (Ahlmann-Eltze et al., 2024), both previously shown to outperform many existing methods, including foundation models such as scGPT (Cui et al., 2024) and scFoundation (Hao et al., 2024). For the mean baseline, we average expression profiles per perturbation over training cells (variants aggregated by cell type or experimental batch are reported in App. C.3). We further compare against leading perturbation models[2]: CPA (Lotfollahi et al., 2023), STATE (Adduri et al., 2025), CellFlow (Klein et al., 2025), and Squidiff (He et al., 2025). All baselines are trained under the same data split using their official implementations and evaluated under a unified protocol for fair comparison.

**Evaluation Metrics.** We adopt the Cell-Eval framework (Adduri et al., 2025) for evaluation following STATE, alongside the Coefficient of Determination ($R^2$) metric from CellFlow. Metrics assess performance from two perspectives: (1) **average expression accuracy across cells**—measuring how well the predicted average expression shift matches the ground truth, using $R^2$, Pearson Delta Correlation (PDCorr), Mean Absolute Error (MAE), Mean Squared Error (MSE), and the Perturbation Discrimination Score ($PDS_{L1}$, $PDS_{L2}$, $PDS_{cos}$) (2) **biologically meaningful differential gene patterns across genes**—evaluating whether predicted cells capture true differential expression (DE) patterns, using DE overlap (DEOver), DE precision (DEPrec), direction agreement (DirAgr), log-fold-change Spearman correlation (LFCSpear), AUROC, AUPRC, and effect size correlation (ES). Among these, *DE-related metrics* are par-

---
[2]Unlasting (Chi et al., 2025) is excluded (no public code).

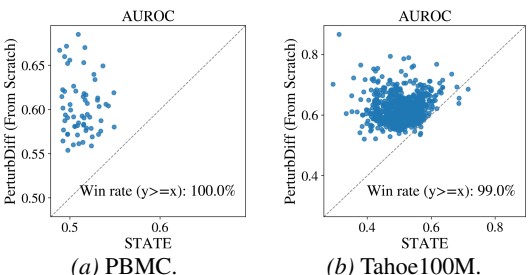

*(a)* PBMC.   *(b)* Tahoe100M.

*Figure 4.* **Per-metric scatter plots comparing PerturbDiff (From Scratch) and STATE**. Each point denotes one held-out condition (62 conditions for PBMC; 735 for Tahoe100M).

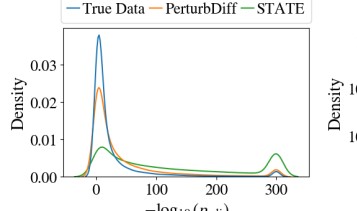 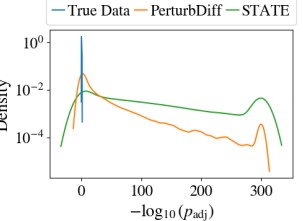

*(a)* Density over DE genes.   *(b)* Density over non-DE genes.

*Figure 5.* **Distribution of** $-\log_{10}(p_{\text{adj}})$ **for true DE and non-DE genes on PBMC** across true data, PerturbDiff (From Scratch), and STATE. Larger values indicate a gene is more likely DE.

ticularly important, as the core of perturbation modeling is to recover biologically meaningful gene responses rather than merely matching average expression levels. See detailed metric definitions in App. C.2.

## 5.2. Main Results: Perturbation Performance

**Overall performance across datasets.** Across 12 diverse metrics[3] (Fig. 3), PerturbDiff (From Scratch) consistently outperforms all baselines across nearly all metrics on the large-scale PBMC and Tahoe100M datasets, and achieves the second best performance on the smaller Replogle dataset. This demonstrates the effectiveness of our diffusion framework that models cell distributions as random variables rather than single cells. In particular, PerturbDiff (From Scratch) shows substantial gains on differential expression (DE)-related metrics like AUPRC and AUROC, indicating improved capture of biologically meaningful, population-level response shifts beyond average expression matching.

While PerturbDiff (From Scratch) slightly trails behind STATE on Reploge, its pretrained and finetuned counterpart, PerturbDiff (Finetuned), consistently improves performance across all metrics and closes the gap, matching STATE on Replogle. This highlights the benefit of marginal pretraining for learning transferable biological priors. On PBMC and Tahoe100M, PerturbDiff (Finetuned) further improves DE-related metrics, while showing mixed behavior on average expression accuracy metrics: it remains comparable to PerturbDiff (From Scratch) on PBMC, but slightly underperforms on Tahoe100M. This suggests that finetuning primarily enhances conditional distributional shifts rather than averaged cell-level accuracy.

Squidiff and CellFlow exhibit the weakest overall performance, ranking last on multiple evaluation metrics across datasets. While simpler baselines may perform well on a single dataset, their performance often collapses when evaluated on others. For example, the Linear model achieves competitive results on Tahoe100M but collapses on Re-

---
[3]We report higher-is-better metrics in Fig. 3; lower-is-better metrics (MSE, MAE) and full numerical results are in App C.3.

plogle, with all three PDS metrics dropping to around 0.5, comparable to random predictions (App. C.2.1). Similarly, the Mean baseline performs reasonably on PBMC but degrades substantially on Tahoe100M, with both AUPRC and AUROC approaching random levels. Together, these results underscore the need for a unified model that generalizes robustly across datasets and metrics.

**Consistent performance across perturbations.** We further analyze per-perturbation performance on PBMC and Tahoe100M (Fig. 4): PerturbDiff (From Scratch) outperforms STATE on nearly all test perturbation types for DE-related metrics (like AUROC, AUPRC, and DEPrec), achieving win rates above 96–100%. This indicates improved model capability to understand differential expression patterns. These metrics require coherent modeling of perturbation effects across cells, rather than matching average expressions, further supporting the advantage of diffusion over cell distributions. More metrics are reported in App. C.4.

**Accurate recovery of perturbation-driven differential expression (DE) patterns.** We evaluate gene-level recovery of perturbation-driven DE patterns to compare our model with the strongest baseline, STATE. In Cell-Eval, genes are identified as DE using a Wilcoxon rank-sum test (Soneson & Robinson, 2018) with adjusted $p$-value $p_{\text{adj}} < 0.05$. In Fig. 5, PerturbDiff (From Scratch) clearly separates DE and non-DE genes, closely matching the ground truth. In contrast, STATE predicts most genes as DE by assigning large $-\log_{10}(p_{\text{adj}})$ values, including many true non-DE genes. This systematic overconfidence suggests that STATE fails to learn the perturbation driven DE patterns.

## 5.3. Pretraining Improves Low-Data Adaption

**Leveraging pretrained marginal manifolds for zero-shot prediction.** Marginal pretraining induces cell state manifolds that may serve as priors for perturbation prediction. We test this hypothesis by probe the pretrained model in a zero-shot setting, where no perturbation-specific supervision is provided at test time. On PBMC and Replogle, the pretrained model yields substantially higher $R^2$ (Fig. 6) and consistently better performance across other metrics

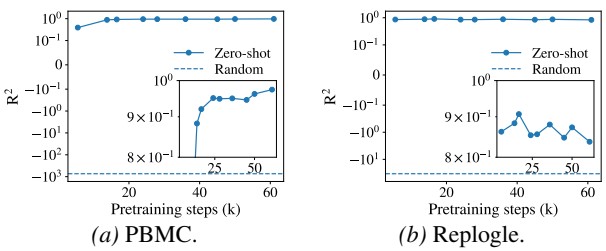

*(a)* PBMC.          *(b)* Replogle.

*Figure 6.* **Zero-shot $R^2$ performance across pretraining steps** versus a random baseline. Insets plot the curve on a linear $y$-axis.

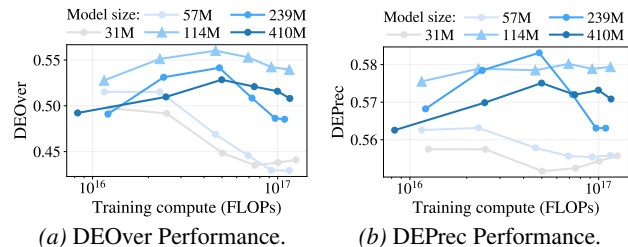

*(a)* DEOver Performance.     *(b)* DEPrec Performance.

*Figure 7.* **Scaling behavior of PerturbDiff (From Scratch)** on PBMC with respect to training compute (FLOPs) and model size.

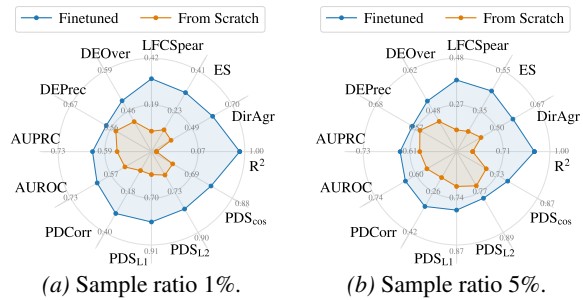

*(a)* Sample ratio 1%.     *(b)* Sample ratio 5%.

*Figure 8.* **Performance on downsampled PBMC**, comparing PerturbDiff (From Scratch) and PerturbDiff (Finetuned) across metrics.

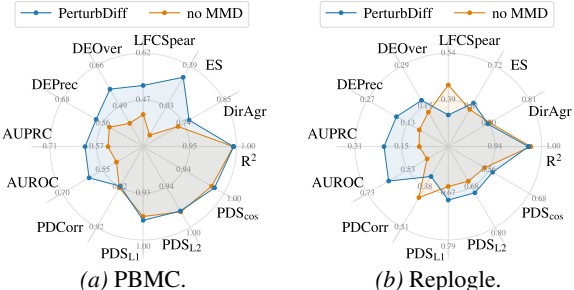

*(a)* PBMC.          *(b)* Replogle.

*Figure 9.* **Ablation study of PerturbDiff (From Scratch)** with and without the MMD objective.

(App. C.5) compared to random initialization. This performance gap implies that biological perturbations do not move cells into arbitrary regions of the expression space; rather, perturbation-induced shifts partially align with structured directions already encoded in the marginal cell distributions.

**Adaptation under Naturally Limited Data.** Unlike PBMC and Tahoe100M, Replogle provides a naturally low-data scenario, with only ∼300 cells per gene perturbation on average (Tab. 1). In this setting, as discussed in Sec. 5.2, PerturbDiff (From Scratch) achieves competitive performance but trails STATE, indicating that training from scratch struggles to learn stable representations from limited samples. Pretraining, however, substantially improves performance and closes the gap to STATE, improving sample efficiency during downstream fine-tuning.

**Adaptation under Controlled Low-Data Regimes.** To examine data scarcity in a controlled setting, we downsample PBMC by uniformly subsampling cells per perturbation at fixed ratios (1%, 5%), ensuring that all perturbation types remain represented. In Fig. 8, across both ratios, PerturbDiff (Finetuned) substantially outperforms PerturbDiff (From Scratch) on all metrics, confirming the sample-efficiency advantage of marginal pretraining. Performance improves for both models as the data ratio increases, and the performance gap between them narrows as expected. Additional comparisons across training steps are provided in App. C.6.

These results show that pretraining provides transferable priors, enabling adaptation when only few perturbed cells are available. Together with the natural low-data regime in Replogle, these support that pretraining enhances perturbation

modeling under limited data.

### 5.4. Scaling Study

To evaluate the scaling of PerturbDiff (From Scratch), we conduct a study on PBMC by varying model size and training compute. Results are in Fig. 7.

As shown in Fig. 7, scaling is not monotonic along either dimension. Increasing model size does not uniformly improve performance: the medium model (114M) achieves the best and most stable results across compute budges. Increasing compute also does not uniformly help: the large model (239M) peaks at intermediate compute and then degrades, exhibiting commonly observed overfitting behaviors (Nichol & Dhariwal, 2021). Thus, perturbation modeling benefits from moderate model capacity and compute, rather than aggressively scaling. This also motivates our marginal pretraining strategy, which aims to extract transferable biological priors to improve data efficiency without relying on brute-force scaling, by simply enlarging models and mixing all datasets. More per-metric curves are in App. C.7.

### 5.5. Ablation Study

In Fig. 9, we compare PerturbDiff (From Scratch) with a variant where the MMD loss is removed from Eqn. (8).

Fig. 9 indicates that removing MMD consistently degrades performance across datasets, especially on DE-metrics. Our further analysis shows that MSE performs poorly on highly sparse single-cell data: over 95% of expressions are zero in PBMC and over 60% in Replogle (App. A.4), likely causing the MSE loss to be dominated by shared zeros.

## 5.6. More Experiments

We clarify the implementation pseudo code in App. D.1, and provide additional analyses in App. D to further validate PerturbDiff beyond the main benchmarks. These include a clarified held-out-context evaluation protocol (App. D.2.1), unseen-cell-line generalization on Replogle (App. D.2.2), population-level discrepancy metrics (App. D.2.3), and ablations isolating the role of diffusion from the MMD objective (App. D.2.4). We also report robustness studies on kernel choice (App. D.2.5), latent batch variability (App. D.2.6), sampling cost (App. D.2.7), batch-size effects (App. D.2.8), and seed variability (App. D.2.9). Finally, App. D.3 presents an explicit Nyström KME variant (Chatalic et al., 2022), showing that a more direct RKHS realization can further improve performance while being less scalable than our default cell-space implementation, which further highlights the computational advantages of our RKHS formulation.

# 6. Conclusion

We proposed PerturbDiff, a functional diffusion framework for single-cell perturbation modeling. In contrast to existing methods, which treat individual cells as random variables, PerturbDiff operates on entire cell distributions as random variables, allowing the model to capture response variability induced by unobserved biological and technical latent factors, rather than collapsing responses into a single static distribution. To do so, we embed empirical cell distributions into a RKHS and define diffusion directly in this function space, which naturally induces an MMD-based denoising objective with a tractable approach for scalable learning.

Empirically, across signaling, drug, and genetic perturbation benchmarks, PerturbDiff achieves state-of-the-art performance, particularly in recovering perturbation-driven differential expression and efficiently adapting to low-data regimes. These results establish PerturbDiff as an effective and principled paradigm for virtual cell modeling.

## Impact Statement

This paper focuses on improving single-cell perturbation modeling by introducing a functional diffusion framework defined over cell populations, aiming to better predict population-level responses for unseen perturbations. Such models have the potential to accelerate biological discovery and experimental design in areas such as functional genomics and drug discovery, where large-scale perturbation assays are costly and limited. While the work is methodological, its downstream use requires care: biases in training data and over-interpretation of predicted effects could lead to misleading biological conclusions. Accordingly, these models should be used as decision-support tools alongside experimental validation.

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

# Appendix Overview

# A. Datasets

## A.1. Data Sources

### A.1.1. SINGLE CELL PERTURBATION DATA

**PBMC** (Biosciences, 2023). The PBMC dataset is a large-scale single-cell signaling perturbation dataset, comprising approximately 10 million human cells collected from 12 healthy donors. It covers 18 immune cell types and includes 90 distinct cytokines perturbation conditions alongside a *PBS* control condition. After quality control, around 9.7 million cells were retained, with a mean sequencing depth of ∼31,000 reads per cell, ensuring high data quality. Prior analyses also show that many cytokine treatments induce strong transcriptional responses, characterized by the upregulation of more than 50 genes (Norman et al., 2019). Owning to its scale, diversity of perturbations, and strong perturbation-induced expression shifts, this dataset provides a representative benchmark for studying perturbation-driven distributional changes in cellular populations, and has been widely adopted in many prior works (Adduri et al., 2025; Klein et al., 2025).

**Tahoe100M** (Zhang et al., 2025). The Tahoe100M atlas represents a transformative leap in single-cell perturbation research. As the largest perturbation dataset ever constructed, it comprises over 100 million single cell expression profiles collected from 50 human cancer cell lines exposed to more than 1,100 small-molecule drug treatment and dosage combinations. Critically, its innovative multiplexed cell-village experimental design during sequencing substantially reduced batch effects, yielding good data quality. Collectively, Tahoe100M has been recognized as a key resource for AI-driven cellular modeling, and has been widely adopted in large-scale foundation models for single-cell biology (Adduri et al., 2025; Rizvi et al., 2025).

**Replogle** (Nadig et al., 2025). The Replogle dataset is a landmark, large-scale resource for studying single-cell genetic perturbation effects. It integrates multiple experiments across four human cell lines (K562, RPE1, Jurkat, and HepG2), providing rich cellular-context diversity. Generated via pooled CRISPR interference (CRISPRi) (Gilbert et al., 2013) to systematically repress thousands of essential genes, this data resource captures transcriptome-wide expression profiles. Prior analyses indicate that its perturbed essential genes often lead to broad transcriptional changes, impacting the expression of hundreds of downstream target genes (Nadig et al., 2025).

### A.1.2. SINGLE CELL RNA-SEQ DATA

**CellxGene** (Program et al., 2025). CellxGene is a large-scale, community-driven single-cell transcriptomics platform that aggregates and standardizes publicly available single-cell RNA-seq datasets across diverse experiments. As one of the earliest initiatives to systematically consolidate single-cell datasets at the scale of tens of millions of cells, CellxGene played a pivotal role in enabling the pretraining of large-scale foundation models for single-cell biology. For example, Geneformer (Theodoris et al., 2023), one of the first transcripotmic foundation models published in *Nature*, relied on CellxGene as a primary data source. Importantly, CellxGene covers more than 600 cell types and 10,000 experimental batches, and a wide range of biological context as its meta data, including cellular development stage and disease, providing exceptionally rich contextual diversity.

## A.2. Downstream Data for Perturbation Prediction

For the perturbation response prediction task, we consider three perturbation datasets (PBMC, Tahoe100M, and Replogle) for evaluation following STATE (Adduri et al., 2025). As detailed in Tab. 2(a), the sample sizes for these datasets range from hundreds of thousands to tens of millions of single cells, together representing a comprehensive evaluation across multiple scales. Furthermore, all these datasets exhibit high heterogeneity across perturbation conditions, cell types, and experimental batches, providing a rigorous testbed for modeling complex context-dependent perturbation responses.

### A.2.1. RAW COUNT PREPROCESSING

We applied standard preprocessing steps to the raw transcriptomic expression counts. Specifically, for PBMC and Tahoe100M, we performed per-cell library-size normalization (using 'scanpy.pp.normalize_total'), followed by a log1p transformation (using 'scanpy.pp.log1p'). The transformation brings gene expression values to a moderate range, which lies within $[0, 10)$ empirically. We further rescaled the log-transformed expressions by a constant factor of 10 to map the values approximately into the range $[0, 1)$, which enables effective learning for the diffusion models.

For the Replogle dataset, we followed the preprocessing protocol of STATE (Adduri et al., 2025), and directly adopted their processed data. In particular, STATE first applies a filtering procedure based on on-target knockdown efficacy to retain only

*Table 2.* Summary statistics of both perturbation and single-cell RNA-seq datasets used in this study.

*(a)* Perturbation datasets. Statistics include counts of perturbed cell across train/valid/test splits and counts of control cells, total gene vocabulary size and HVGs, and dataset diversity in terms of perturbations, cell types, and experimental batches.

| Dataset | Perturbation Type | #Perturbed Cells | | | #Control Cells | #Genes | #HVGs | #Perturbations | #Cell Types | #Batches |
|---|---|---|---|---|---|---|---|---|---|---|
| | | Train | Valid | Test | | | | | | |
| PBMC | cytokine | 6,657,336 | 150,484 | 2,260,453 | 629,701 | 62,710 | 2,000 | 90 | 18 | 12 |
| Tahoe100M | drug | 89,495,239 | 553,076 | 8,790,272 | 2,330,156 | 40,352 | 2,000 | 1,137 | 50 | 14 |
| Replogle | gene | 572,545 | 4,825 | 26,878 | 39,165 | 6,642 | 2,000 | 2,023 | 4 | 56 |

*(b)* Single-cell RNA-seq datasets. Statistics include counts of cells and the number of datasets before and after merging, averaged gene vocabulary size across datasets, the union of HVGs for pretraining, the average number of genes shared between each dataset and the pretraining HVG set, and dataset diversity in terms of cell types and experimental batches.

| Dataset | #Cells | #Original Datasets | #Merged Datasets | Avg. #Genes | #HVGs | Avg. #Shared Genes | #Cell Types | #Batches |
|---|---|---|---|---|---|---|---|---|
| CellxGene | 60,901,077 | 1,139 | 23 | 7,344.3 | 10,045 | 6,867.5 | 662 | 10,887 |

*Table 3.* Overlap between dataset-specific gene vocabularies and the pretraining gene set (12,626 genes). Shared gene counts and ratios are reported; CellxGene values are averaged across the 23 merged single-cell RNA-seq datasets.

| Dataset | PBMC | Tahoe100M | Replogle | CellxGene |
|---|---|---|---|---|
| #Shared Genes | 12,478 | 12,578 | 5,760 | 6,958.0 |
| Shared Ratio (%) | 98.8 | 99.6 | 45.6 | 55.1 |

perturbations and cells exhibiting sufficient repression of the targeted gene, thereby improving the signal-to-noise ratio of perturbation effects. After filtering, they also applied library-size normalization prior to log-transformation. We later similarly applied the constant rescaling by a factor of 10 to their processed data.

### A.2.2. Highly Variable Gene (HVG) Selection

After raw count preprocessing for each dataset, we followed the standard practice (Wolf et al., 2018) and extracted the top 2,000 HVGs from the data's original transcriptome gene vocabulary (using 'sc.pp.highly_variable_genes'). Excluding low-variance genes can effectively enhance the signal-to-noise ratio without sacrificing biological fidelity, and ensure that the task focuses on the most information-rich features of the data.

### A.2.3. Data Splits for Task Evaluation

We follow the same data-splitting method as STATE (Adduri et al., 2025), which adopt dataset-specific holdout strategies to evaluate model generalization. For the PBMC dataset, 4 of the 12 donor cell lines were reserved for testing. To test the models under conditions of partial perturbation coverage, 30% of the perturbations from these held-out donors were moved into the training set, leaving the remaining 70% for testing. For Tahoe100M, PCA was first performed on pseudobulked expression profiles of the 50 cell lines to identify distinct phenotypic clusters. From these clusters, five cell lines were chosen as a held-out test set. For the Replogle dataset, one cell line (hepg2) was held out for testing, while models were trained on the remaining three cell lines together with 30% of perturbations randomly sampled from the held-out cell line. The number of cells across splits is summarized in Tab. 2(a).

### A.3. Pretraining Data for Marginal Distribution Learning

For the unconditional pretraining stage, we incorporated two data sources: both the three aforementioned perturbation datasets and the single-cell RNA-seq data. For the former, we used the same train/validation/test split to ensure that there is no information leakage. For the latter, we followed the same curation protocol as the SE cell foundation model (Adduri et al., 2025) and aggregated expression profiles from 1,139 diverse studies in CellxGene (Program et al., 2025). This corpus contains over 60 million cells spanning more than 600 cell types and 10,000 experimental batches, offering substantially broader biological coverage and experimental diversity compared to the perturbation datasets alone (Tab. 2(b)). As such, it provides a strong prior for learning general-purpose cellular representations, together with the observable perturbation cells.

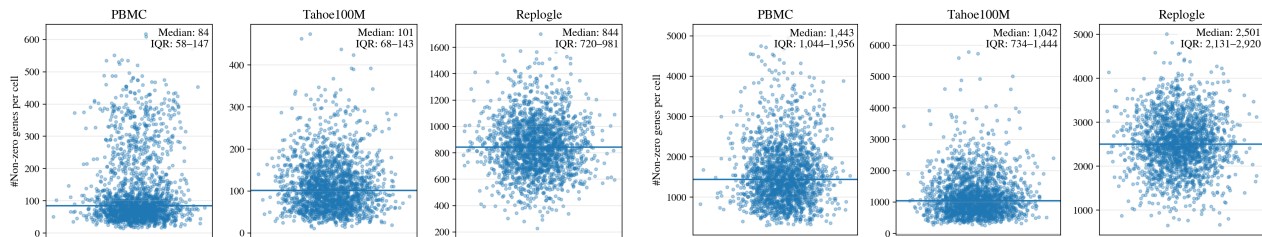

*(a)* Using the downstream HVG gene set (2, 000 genes).  *(b)* Using the pretraining gene set (12, 626 genes).

*Figure 10.* The number of non-zero genes per cell across datasets under different gene sets: (a) downstream HVGs (b) pretraining gene set.

### A.3.1. RAW COUNT PREPROCESSING

In the pretraining stage, the three perturbation datasets applied the same preprocessing pipeline as discussed in App. A.2.1. And all the single-cell RNA-seq datasets were preprocessed using the same pipeline as PBMC and Tahoe100M.

### A.3.2. HIGHLY VARIABLE GENE (HVG) SELECTION

**HVG selection for perturbation datasets.** We follow the same HVG selection strategy for preprocessing perturbation downstream data as in App. A.2.2.

**HVG selection for single-cell RNA-seq datasets.** However, retrieving a unified HVG feature space across such a diverse collection of studies presents significant challenges. Directly computing HVGs on the entire corpus is computationally prohibitive and risks out-of-memory failures. On the other hand, taking the union of top 2,000 HVGs computed independently for all 1,139 studies leads to an excessively high-dimensional vocabulary (more than 30k genes), which would be likely dominated by study-specific noise and hinder effective representation learning.

To balance HVG coverage and computational tractability, we adopted a *merge-then-select* HVG selection strategy. Specifically, we merged the 1,139 studies into 23 composite datasets and independently selected the top 2,000 HVGs with each composite. Taking the union of these HVG sets yields a global vocabulary of 10,045 genes for the CellxGene corpus. On average, each merged dataset covers approximately 6,867 genes from this global HVG vocabulary.

**Construction of the unified pretraining gene vocabulary.** Finally, we constructed the gene space used for pretraining by taking the union of the three 2,000-HVG sets from the perturbation datasets and the 10,045 global HVGs from CellxGene, resulting in the pretraining gene vocabulary $\mathcal{G}_{\text{pretrain}}$ of 12,626 genes. Notably, even when considering only the HVGs derived from the perturbation datasets, Tab. 3 shows that PBMC and Tahoe100M share more than 98% of their genes within this pretraining vocabulary, while Replogle and CellxGene retain around 50% coverage. This substantial shared. gene space provides a solid foundation for learning transferable representations from large-scale single-cell RNA-seq data and effectively adapting them to perturbation settings.

### A.4. Data Statistics Analysis for Perturbation Datasets

Perturbation datasets are highly structured, heterogeneous, and severely imbalanced across multiple levels, including individual cells, perturbations, cell types, and experimental batches. These characteristics fundamentally motivate modeling perturbation responses at the distribution level, rather than at the level of individual cells alone.

**Cell-level sparsity.** As shown in Fig. 10(a), when considering 2,000 HVGs, PBMC and Tahoe100M contain on average 100 non-zero genes per cell, resulting in extreme sparsity. Such sparsity makes cell-level objectives, *i.e.*, the per-cell MSE loss for diffusion model, dominated by zero entries and prone to suboptimal solutions. This observation directly motivates modeling distributions of cells besides individual cells alone. When considering the full pretraining gene vocabulary $\mathcal{G}_{\text{pretrain}}$ (Fig. 10(b)), a consistent pattern emerges across datasets: PBMC and Tahoe100M remain highly sparse (approximately 5% non-zero entries), while Replogle exhibits substantially lower sparsity (around 40% non-zero entries). These dataset-specific zero-inflation patterns further complicate joint pretraining and underscore the necessity of distribution-aware modeling.

**Imbalance across perturbations and their cell-type- and batch-specific subpopulations.** Perturbation datasets also exhibit severe imbalance in sample sizes. As shown in Fig. 11, the number of cells per perturbation follows a heavy-tailed distribution in Replogle and Tahoe100M, with many perturbations supported by only a limited number of cells. This

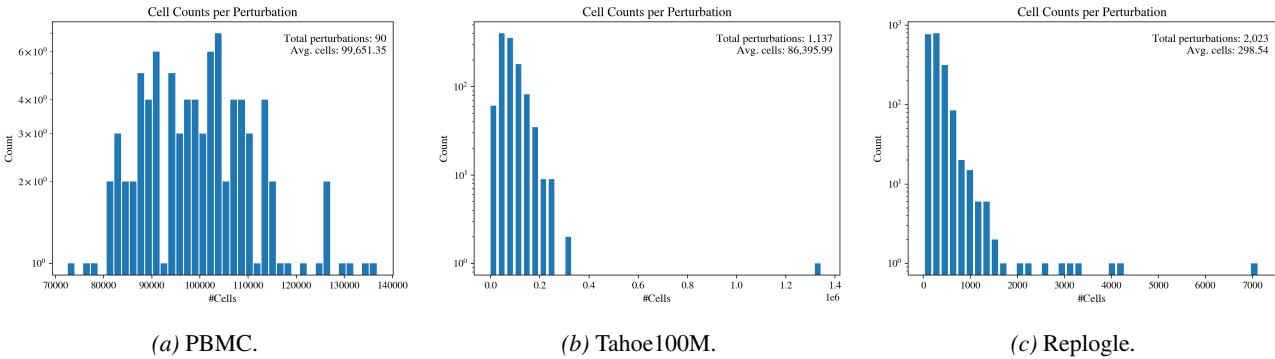

*(a)* PBMC.        *(b)* Tahoe100M.        *(c)* Replogle.

*Figure 11.* Distribution of cell counts per perturbation condition across datasets. Insets report the total number of perturbations and the average number of cells per perturbation. Lower within-perturbation MMD reflects regular cellular responses under the same perturbation, while higher between-perturbation MMD indicates distinct perturbation-specific effects.

imbalance becomes even more pronounced when conditioning on finer-grained subpopulations, such as (perturbation, cell type, experimental batch) combinations (Fig. 12). These observations substantially complicate learning at the individual-cell level, as sparse subpopulations provide noisy and unstable supervision. This also motivates modeling perturbation effects at the distribution level, which allows aggregating information across cells and stabilizing learning under severe data imbalance.

**Perturbation heterogeneity.** Fig. 13 compares distributional differences measured by Maximum Mean Discrepancy (MMD) within and between perturbations. Across datasets, inter-perturbation distances consistently and substantially exceed intra-perturbation distances, indicating that perturbations induce evident structured shifts in cellular distributions rather than stochastic cell-level noise.

Notably, PBMC and Tahoe100M exhibit both substantially smaller mean inter-perturbation MMD and smaller variance compared to Replogle. This is likely not indicative of weaker biological effects, but rather reflects the high sparsity of single-cell profiles, which compresses the effective support of these distributions and suppresses the observable distribution divergence. For Replogle, the large variance of inter-perturbation MMD further suggests pronounced heterogeneity in perturbation response strength, where some perturbations induce dramatic distributional shifts while others lead to milder effects. Overall, these results indicate that perturbation responses are characterized by structured, context-dependent shifts in cellular populations.

**Cell-type composition and induced distributional shifts across perturbations.** Furthermore, perturbation datasets exhibit highly imbalanced cell-type compositions (Fig. 14). For example, in PBMC, the *CD4 Naive* population contains over one million cells, whereas the rare *Plasmablasts* population comprises fewer than ten thousand cells. Despite this extreme imbalance, the magnitude of inter-perturbation distributional shifts, measured by MMD within each cell type, shows no correlation with cell abundance. As illustrated in Fig. 14, although *CD4 Naive* cells dominate the dataset, their inter-perturbation MMD remains relatively small, whereas *CD14 Monocytes* exhibit substantially larger distributional shifts despite having fewer cells. This aligns with our understanding that perturbation sensitivity is governed by intrinsic, cell-type-specific regulatory programs while has no relationship with sample size or population frequency. It also helps explain why simple cell-type-specific mean baseline can perform reasonably well. More importantly, this observation highlights that perturbation responses emerge as structured, context-dependent transformations at the distribution level, necessitating modeling approaches that explicitly capture conditional distributional shifts.

# B. Method Details

## B.1. Diffusion over Cell Distributions in Reproducing Kernel Hilbert Space (RKHS)

Our goal is to model the distribution shift between control and perturbed single-cell populations. To this end, in App. B.1.1, we formally define cell distribution-valued random variables. In App. B.1.2, we begin by embedding the control and perturbed cell distributions into a structured RKHS (Hastie et al., 2009), which provides a principled space for defining diffusion processes directly at the distribution level. In App. B.1.3, we then connect this functional formulation to observed data by constructing empirical distributions from finite batches of single cells. These empirical kernel mean embeddings serve as Monte Carlo samples of the underlying distribution-valued random variables and constitute the objects to which noise is added and removed during diffusion. Since our Hilbert space is infinite-dimensional, in App. B.1.4, we introduce

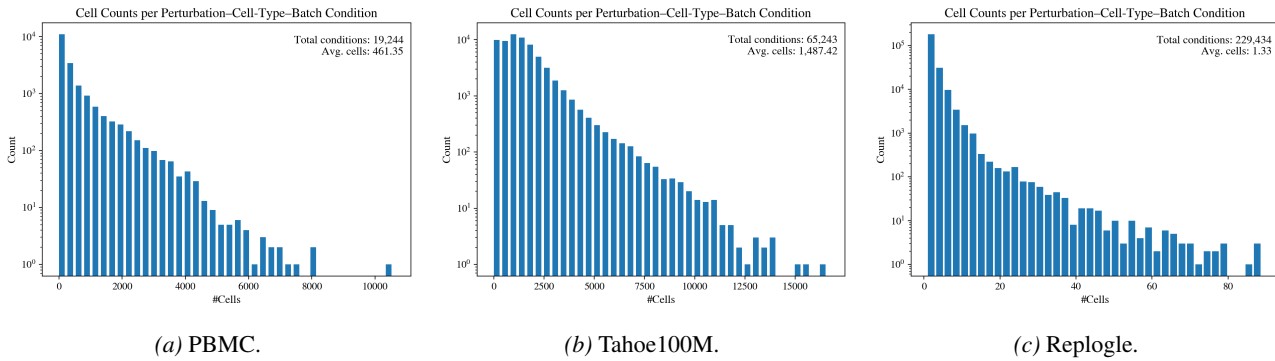

*(a)* PBMC.     *(b)* Tahoe100M.     *(c)* Replogle.

*Figure 12.* Distribution of cell counts per (perturbation, cell type, experimental batch) combination across datasets. Insets report the total number of perturbations and the average number of cells per combination. Lower within-combination MMD reflects intrinsic cellular variability under identical perturbation, cell-type, and batch conditions, while higher between-combination MMD indicates perturbation-induced distributional differences beyond this baseline variability.

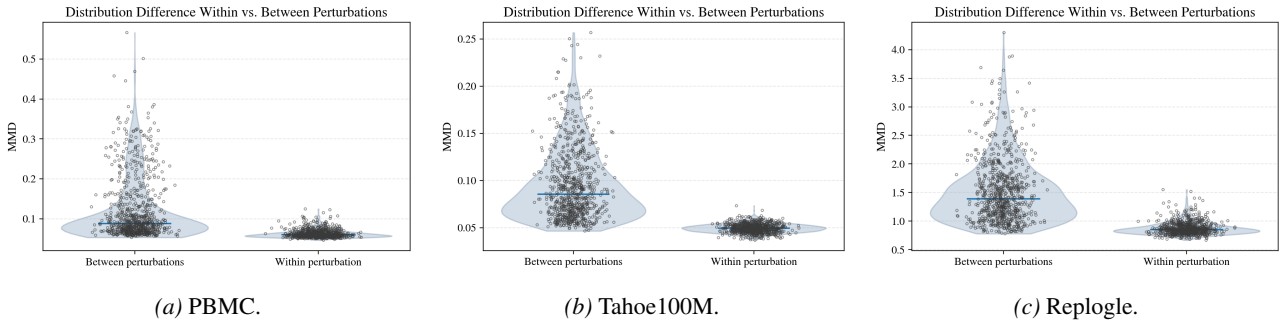

*(a)* PBMC.     *(b)* Tahoe100M.     *(c)* Replogle.

*Figure 13.* Distribution differences within and between perturbations. Violin plots show MMD values computed between cell distributions from the same perturbation (within) and from different perturbations (between) across perturbation datasets.

*Gaussian Random Elements* as a generalization of Gaussian measures to infinite-dimensional Hilbert spaces, which play an analogous role to Gaussian noise vectors in Euclidean diffusion models.

Building on this, in App. B.1.5 and App. B.1.6, we develop the forward and backward diffusion processes in RKHS, respectively. Finally, in App. B.1.7, we discuss a tractable implementation for the proposed diffusion framework over cell distributions. We show that both the reverse-time objective and the forward-time noise injection can be carried out through operations on empirical cell sets, while remaining consistent with the underlying RKHS diffusion formulation.

Taken together, these sections establish a theoretical framework for generative modeling directly over distributions in RHKS, along with a simple and effective procedure for carrying out the corresponding computations in cell space.

### B.1.1. CELL DISTRIBUTIONS AS RANDOM VARIABLES

For a fixed cellular context $c$ and perturbation $\tau$, we model the perturbed population not as a single deterministic distribution, but as a *distribution-valued random variable*

$$D_{c,\tau} \sim \mathbb{P}_{c,\tau}, \tag{10}$$

where $\mathbb{P}_{c,\tau}$ is the law of cell distributions. Analogously, we define the control population as a random variable $D_c \sim \mathbb{P}_c$.

This formulation reflects the fact that cell populations collected under the same observed condition $(c, \tau)$ may be influenced by unobserved biological and technical factors, resulting in a family of plausible response distributions. Under this view, the commonly assumed target distribution $P_{c,\tau}$ corresponds to the expectation of the random variable $D_{c,\tau}$, i.e. $P_{c,\tau} = \mathbb{E}[D_{c,\tau}]$.

### B.1.2. CELL DISTRIBUTIONS AS POINTS IN A HILBERT SPACE VIA KERNEL MEAN EMBEDDING.

Let $k : \mathcal{X} \times \mathcal{X} \to \mathbb{R}$ be a positive definite kernel. Then, by the Moore-Aronszajn theorem, there exists a unique reproducing kernel Hilbert space (RKHS) $\mathcal{H}_k$ of functions on $\mathcal{X}$, for which $k$ is a reproducing kernel. In particular, for all $x \in \mathcal{X}$, there exists a unique element $k_x \in \mathcal{H}_k$ such that $f(x) = L_x(f) = \langle f, k_x \rangle_{\mathcal{H}_k} \ \forall f \in \mathcal{H}_k$, where $L_x \in \mathcal{H}$ is the point-evaluation

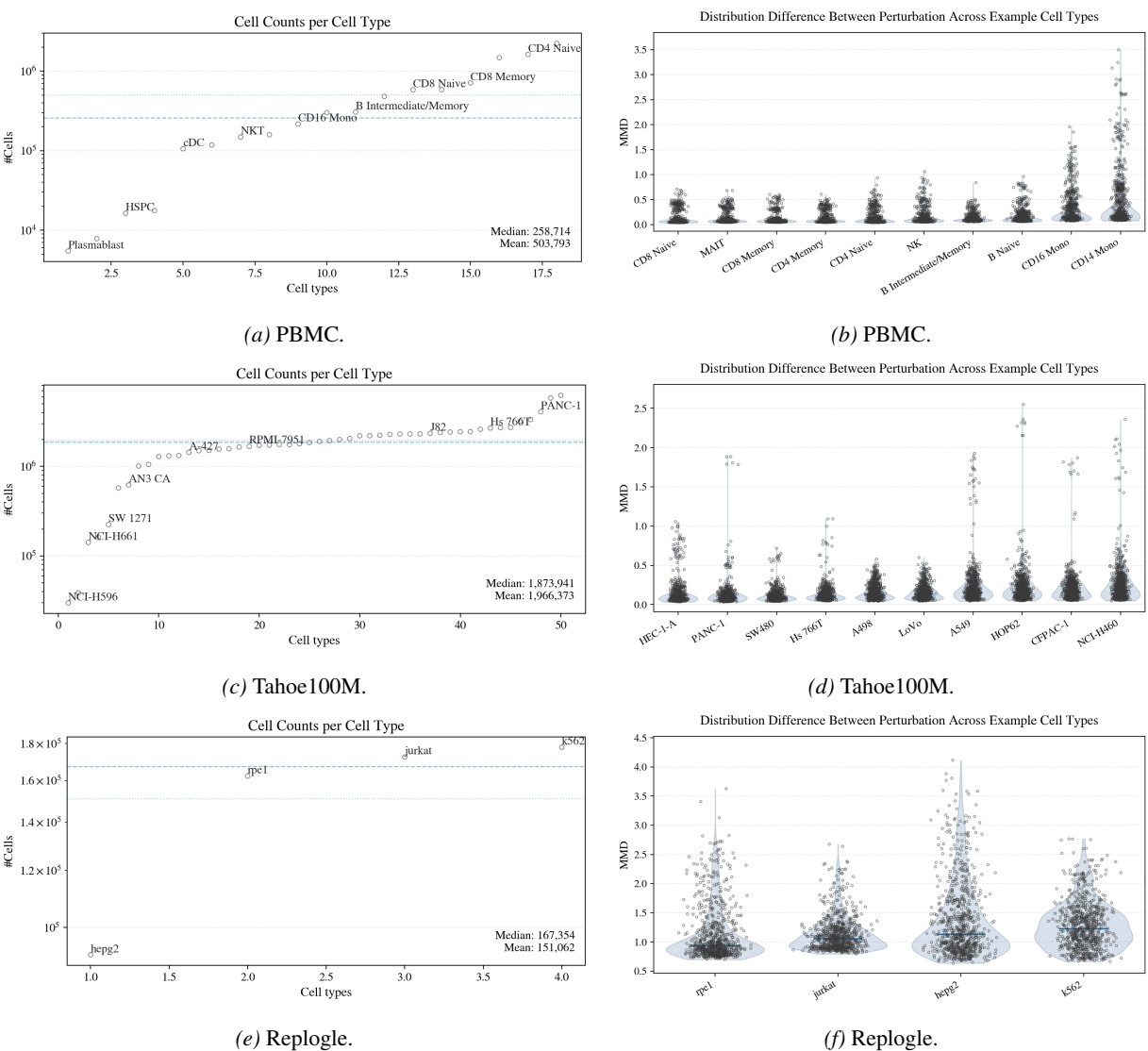

*Figure 14.* Cell counts per cell type (left panels (a), (c), and (e)) and between-perturbation MMD across example cell types (right panels (b), (d) and (f)), showing the heterogeneity in cell-type coverage and distributional differences across perturbations for specific cell types.

functional defined as $L_x(f) := f(x)$. For any probability measure $P$ over $\mathcal{X}$, its *kernel mean embedding* is defined as

$$\boldsymbol{\mu}_P := \mathbb{E}_{Z \sim P}[k(Z, \cdot)] \in \mathcal{H}_k, \tag{11}$$

provided that the expectation exists if $\mathbb{E}_{Z \sim P}[\sqrt{k(Z, Z)}] < \infty$. In our work, we employ the energy distance kernel, which satisfies this condition under mild assumptions on the data distribution. Specifically, although we formally denote cell states as $\mathbf{x} \in \mathbb{R}^{|\mathcal{G}|}$ (Sec. 2.1) for convenience, in practice, gene expression values are non-negative integers with finite dynamic range, implying that the support of the cell space $\mathcal{X}$ is bounded. Consequently, the kernel is bounded on $\mathcal{X}$, ensuring the existence of the kernel mean embedding $\boldsymbol{\mu}_P$.

This kernel mean embedding endows the space of probability measures with a Hilbert-space structure with the following standard properties (Muandet et al., 2017).

**Lemma B.1** (Basic properties of kernel mean embeddings)**.** *For any distribution $P$ and $Q$ on $\mathcal{X}$ and any $\alpha \in [0, 1]$, the following hold:*

*(1) Linearity under mixing: $\boldsymbol{\mu}_{(\alpha P + (1-\alpha)Q)} = \alpha\boldsymbol{\mu}_P + (1 - \alpha)\boldsymbol{\mu}_Q$, ensuring that convex combinations of distributions correspond to the convex combinations of the kernel mean embeddings in $\mathcal{H}_k$.*

*(2) Kernel geometry: $\langle \boldsymbol{\mu}_P, \boldsymbol{\mu}_Q \rangle_{\mathcal{H}_k} = \mathbb{E}_{Z \sim P, Z' \sim Q} k(Z, Z')$, meaning the Hilbert inner product coincides with an expected kernel similarity.*

*(3) Distributional distance through MMD: $\|\boldsymbol{\mu}_P - \boldsymbol{\mu}_Q\|_{\mathcal{H}_k} = MMD_k(P, Q)$, where $MMD_k$ denotes the maximum mean discrepancy induced by $k$ (Gretton et al., 2012), and gives the embedding space a well-understood geometric interpretation.*

Together, these properties allow us to represent entire cellular populations as single points in a Hilbert space, with inner produces and norms corresponding to population-level similarities and discrepancies. This makes $\mathcal{H}_k$ a natural state space for defining diffusion processes over distributions rather than individual cells.

### B.1.3. EMPIRICAL DISTRIBUTIONS AS INSTANTIATIONS OF DISTRIBUTION-VALUED RANDOM VARIABLES

In practice, the distribution-valued random variable $D_{c,\tau}$ is not directly observable. Instead, experiments conducted under a fixed cellular context $c$ and perturbation condition $\tau$ provide only *finite batches of single cells*, each offering a partial and noisy observation of one realization of $D_{c,\tau}$. Different cell batches may be implicitly influenced by different unobserved biological or technical factors, and thus correspond to different realizations of the underlying random variable.

Given a batch of perturbed cells $B_{\text{pert}} = \{\mathbf{x}_1, \ldots, \mathbf{x}_m\} \subset X_{\text{pert}}$, we construct the corresponding empirical distribution

$$\tilde{P}_{\text{pert}} := \frac{1}{m} \sum_{j=1}^{m} \delta_{\mathbf{x}_j}, \tag{12}$$

which serves as a Monte Carlo approximation of a realization drawn from $D_{c,\tau}$. Its associated kernel mean embedding is given by

$$\boldsymbol{\mu}_{\tilde{P}_{\text{pert}}} = \frac{1}{m} \sum_{j=1}^{m} k(\mathbf{x}_j, \cdot) \in \mathcal{H}_k. \tag{13}$$

In total, we consider $M_{\text{pert}}$ such perturbed batches, yielding a collection of empirical kernel mean embeddings $\{\boldsymbol{\mu}_{\tilde{P}_{\text{pert}}}\}$, which constitute empirical samples the random variable $\boldsymbol{\mu}_{c,\tau}$ in $\mathcal{H}_k$.

Similarly, control cells are sampled from realizations of the control distribution-valued random variable $D_c$. We denote $B_{\text{ctrl}} = \{\mathbf{x}_1, \ldots, \mathbf{x}_m\} \subset X_{\text{ctrl}}$ as a batch of control cells, with a total of $M_{\text{ctrl}}$ batches. Each batch induces an empirical distribution and its kernel mean embedding $\boldsymbol{\mu}_{\tilde{P}_{\text{ctrl}}}$, forming observable samples of the random variable $\boldsymbol{\mu}_c$ in $\mathcal{H}_k$.

Furthermore, our empirical cell distribution has exponential convergence in the supremum norm to the ground truth cell distribution as the size of the cell batch grows, the proof of which we refer to Naaman (2021).

**Proposition B.2.** *For some cell population distribution $P$ and its empirical cell distribution $P_m$ formed by $m$ cells $\mathbf{x}_1, \ldots, \mathbf{x}_m \overset{i.i.d.}{\sim} P$ and number of genes $|\mathcal{G}|$, we have*

$$\mathbb{P}_P[\sup_{x \in \mathbb{R}^{|\mathcal{G}|}} |F_n(x) - F(x)| > t] \le 2|\mathcal{G}| \exp(-2mt^2),$$

*for all $t \geq 0$ and $m$ sufficiently large, where $F$ and $F_n$ is the multivariate cumulative distribution function corresponding to $P$ and $P_m$, respectively.*

### B.1.4. GAUSSIAN RANDOM ELEMENTS AND GAUSSIAN MEASURES IN RKHS.

To define a diffusion model over $\mathcal{H}_k$, we require a notion of Gaussian noise acting directly on the elements of $\mathcal{H}_k$. In finite-dimensional spaces such as $\mathbb{R}^d$, Gaussian noise is represented by a multivariate normal random vector, whose distribution is fully characterized by a mean vector and a covariance matrix.

When moving to infinite-dimensional Hilbert spaces, such as the RKHS $\mathcal{H}_k$ in our case, we want to capture the essential properties of Gaussian measures in Euclidean space while maintaining computational tractability. Similar to defining a Gaussian process by restricting its projections to finite-dimensional joint distributions, we define $\Xi$, a $\mathcal{H}_k$-valued random variable, to be a *Gaussian random element* (GRE) if projecting $\Xi$ in any "direction" induces a Gaussian distribution in Euclidean space, *i.e.* the $\mathbb{R}$-valued random variable $\langle f, \Xi \rangle_{\mathcal{H}_k}$ is a Gaussian for any $f \in \mathcal{H}_k$. Note that for any GRE $\Xi \sim P_\Xi$, there exists a unique mean element $m \in \mathcal{H}_k$ given by $m := \int_{\mathcal{H}_k} \Xi \, dP_\Xi$. Moreover, there also exists a unique linear covariance operator $C : \mathcal{H}_k \to \mathcal{H}_k$ defined as

$$C : f \mapsto \int_{\mathcal{H}_k} \langle f, \Xi \rangle_{\mathcal{H}_k} dP_\Xi - \langle f, m \rangle_{\mathcal{H}_k} m.$$

Furthermore, the covariance operator $C$ is symmetric, positive semi-definite, compact, and is trace-class (Brezis, 2011).

**Definition B.3** (Symmetric operator). An operator $C : \mathcal{H}_k \to \mathcal{H}_k$ is symmetric if $\langle Cf, g \rangle_{\mathcal{H}_k} = \langle f, Cg \rangle_{\mathcal{H}_k}$ for all $f, g \in \mathcal{H}_k$.

**Definition B.4** (Positive semi-definite operator). An operator $C : \mathcal{H}_k \to \mathcal{H}_k$ is positive semi-definite if $\langle f, Cf \rangle_{\mathcal{H}_k} \geq 0$ for all $f \in \mathcal{H}_k$.

**Definition B.5** (Compact operator). An operator $C : \mathcal{H}_k \to \mathcal{H}_k$ is compact if the image of the closed unit ball $\{f \in \mathcal{H}_k : \|f\|_{\mathcal{H}_k} \leq 1\}$ under $C$ is relatively compact (i.e. its closure is compact) with respect to the topology induced by $\|\cdot\|_{\mathcal{H}_k}$.

**Definition B.6** (Trace-class operator). The covariance operator $C$ induced by a GRE $\Xi \sim P_\Xi$ is trace-class if it has finite trace, i.e. $\text{tr}(C) = \mathbb{E}_{\Xi \sim P_\Xi}[\|\Xi\|_{\mathcal{H}_k}^2] < \infty$.

The converse also holds, for any $m \in \mathcal{H}_k$ and symmetric, positive semi-definite, compact, and trace-class linear operator $C : \mathcal{H}_k \to \mathcal{H}_k$, there exists a $\mathcal{H}_k$-valued Gaussian measure with mean and covariance $m$ and $C$, respectively. Thanks to this one-to-one correspondence, we can write the distribution of any GRE as $\Xi \sim P_\Xi = \mathcal{N}_{\mathcal{H}_k}(m, C)$. Additionally, the following properties also hold because of the linearity of the inner product.

**Lemma B.7** (Affine transformations). *Let $\Xi \sim \mathcal{N}_{\mathcal{H}_k}(m, C)$ be a Gaussian random element. Then for any $\gamma \in \mathbb{R}$ and $g \in \mathcal{H}_k$,*

$$\gamma \Xi + g \sim \mathcal{N}_{\mathcal{H}_k}(\gamma m + g, \gamma^2 C). \tag{14}$$

**Lemma B.8** (Sum of independent Gaussian random elements). *If $\Xi_1 \sim \mathcal{N}_{\mathcal{H}_k}(m_1, C_1)$ and $\Xi_2 \sim \mathcal{N}_{\mathcal{H}_k}(m_2, C_2)$ are independent Gaussian random elements on $\mathcal{H}_k$, then*

$$\Xi_1 + \Xi_2 \sim \mathcal{N}_{\mathcal{H}_k}(m_1 + m_2, C_1 + C_2). \tag{15}$$

Together, Lemma B.7 and Lemma B.8 imply that the family of Gaussian measures on $\mathcal{H}_k$ is invariant under the forward diffusion operator. Each diffusion step consists of an affine transformation of the previous state, followed by the addition of an independent Gaussian random element (see Eqn. (16)). By induction over diffusion time steps, the marginal distribution of $\Xi_t$ remains Gaussian for all $t$ (see Eqn. (17)).

### B.1.5. FORWARD DIFFUSION ON KERNEL MEAN EMBEDDINGS IN RKHS.

We now define a DDPM-like forward process directly in $\mathcal{H}_k$. Starting from a "clean" kernel mean emebdding $\boldsymbol{\mu}_0 := \boldsymbol{\mu}_{c,\tau}$, we "noise" the embeddings by adding small increments of i.i.d. Gaussian random elements at each timestep. This produces $\boldsymbol{\mu}_0, \boldsymbol{\mu}_1, \ldots, \boldsymbol{\mu}_T$, a Markov chain on $\mathcal{H}_k$ that gradually approaches the fixed reference Gaussian measure $\mathcal{N}_{\mathcal{H}_k}$.

**Definition B.9** (Forward process on distribution kernel mean embeddings). Fix $T \in \mathbb{Z}_{>0}$ and a variance schedule $\{\beta_t\}_{t=1}^T \in [0,1]^T$. Choose a symmetric, positive semi-definite, compact, trace-class covariance operator $C : \mathcal{H}_k \to \mathcal{H}_k$. Let $\{\Xi_t\}_{t=1}^T$ be *i.i.d.* Gaussian random elements with $\Xi_t \sim \mathcal{N}_{\mathcal{H}_k}(0, C)$. We define the forward Markov chain on $\mathcal{H}_k$:

$$\boldsymbol{\mu}_t = \sqrt{1 - \beta_t}\boldsymbol{\mu}_{t-1} + \sqrt{\beta_t}\Xi_t, \quad t = 1, \ldots, T. \tag{16}$$

Following Lem. B.7, the forward chain satisfies: $\boldsymbol{\mu}_t|\boldsymbol{\mu}_{t-1} \sim P_{t|t-1} := \mathcal{N}_{\mathcal{H}_k}(\sqrt{1-\beta_t}\boldsymbol{\mu}_{t-1}, \beta_t C)$. Further following Lem. B.8, it holds true for

$$\boldsymbol{\mu}_t|\boldsymbol{\mu}_0 \sim P_{t|0} := \mathcal{N}_{\mathcal{H}_k}(\sqrt{\alpha_t}\boldsymbol{\mu}_0, (1-\alpha_t)C), \tag{17}$$

where $\alpha_t := \prod_{i=1}^t (1-\beta_i)$.

### B.1.6. REVERSE PROCESS AND DENOISING OBJECTIVE.

The essence of our generative model is its ability to simulate the "time-reversal" of the forward process, transporting elements from a Gaussian-like reference distribution to our data distribution. From an ancestral sampling point of view, we would like to sample $\boldsymbol{\mu}_T \sim \mathcal{N}_{\mathcal{H}_k}(0, C)$, then iteratively sample $\boldsymbol{\mu}_{t-1}$ conditioned on our current sample $\boldsymbol{\mu}_t$. However, Kerrigan et al. (2022) shows that while the posterior law $P_{t-1|t}(\boldsymbol{\mu}_{t-1} \mid \boldsymbol{\mu}_t) := \mathrm{Law}(\boldsymbol{\mu}_{t-1} \mid \boldsymbol{\mu}_t)$ is well-defined, it is intractable. The lack of a universal dominating reference measure (unlike the Lebesgue measure in Euclidean space) in the RKHS means that Radon-Nikodym densities do not exist in general. Without densities, it is not possible to apply Bayes' rule, not to mention the computationally intractable normalization constant.

Instead, similar to diffusion modeling over Euclidean spaces, we approximate the posterior measures with a variational family of Gaussian measures parametrized by some learnable neural network parameters $\theta$. Specifically, we define $Q_T^\theta := \mathcal{N}_{\mathcal{H}_k}(0, C)$ and $Q_{t-1|t}^\theta(\cdot \mid \boldsymbol{\mu}_t) = \mathcal{N}_{\mathcal{H}_k}(m_t^\theta(\boldsymbol{\mu}_t), C_t^\theta(\boldsymbol{\mu}_t))$, where $m_t^\theta(\boldsymbol{\mu_t}) \in \mathcal{H}_k$ is a learnable mean function and $C_t^\theta(\boldsymbol{\mu}_t) : \mathcal{H}_k \to \mathcal{H}_k$ is some valid covariance operator. Additionally, as shown in the following proposition, when further conditioning on the starting element $\boldsymbol{\mu}_0$, the posterior measure becomes tractable.

**Proposition B.10** (Reverse conditional measure $P_{t-1|t,0}(\boldsymbol{\mu}_{t-1}|\boldsymbol{\mu}_t, \boldsymbol{\mu}_0)$). *Let $t = 2, 3, \ldots, T$ and define $\tilde{\beta}_t := \frac{1-\alpha_{t-1}}{1-\alpha_t}$, the reverse-time conditional measure satisfies:*

$$P_{t-1|t,0}(\boldsymbol{\mu}_{t-1}|\boldsymbol{\mu}_t, \boldsymbol{\mu}_0) = \mathcal{N}_{\mathcal{H}_k}(\tilde{m}_t(\boldsymbol{\mu}_t, \boldsymbol{\mu}_0), \tilde{\beta}_t C), \tag{18}$$

*where $\tilde{m}_t : \mathcal{H}_k \times \mathcal{H}_k \to \mathcal{H}_k$ is an affine mapping:*

$$\tilde{m}_t(\boldsymbol{\mu}_t, \boldsymbol{\mu}_0) := \frac{\sqrt{\alpha_{t-1}}\beta_t}{1-\alpha_t}\boldsymbol{\mu}_0 + \frac{\sqrt{1-\beta_t}(1-\alpha_{t-1})}{1-\alpha_t}\boldsymbol{\mu}_t. \tag{19}$$

This is an RKHS analogue of the familiar DDPM posterior (Kerrigan et al., 2022). Similar to the Euclidean case, a corollary of the Feldman-Hàjek theorem allows us to compute, in closed form, the Kullback-Leibler divergence between two $\mathcal{H}_k$-valued Gaussian measures with same covariance operator.

**Lemma B.11** (KL between Gaussian measures with shared covariance). *Let $P = \mathcal{N}_{\mathcal{H}_k}(m_1, \bar{C})$ and $Q = \mathcal{N}_{\mathcal{H}_k}(m_2, \bar{C})$ be Gaussian measures on $\mathcal{H}_k$ with the same covariance operator $\bar{C}$. The KL divergence has the closed form (Kerrigan et al., 2022):*

$$KL(P||Q) = \frac{1}{2}\langle m_1 - m_2, \bar{C}^{-1}(m_1 - m_2)\rangle_{\mathcal{H}_k}. \tag{20}$$

Similar to denoising diffusion probabilistic models in Euclidean space, it is possible to derive a variational lower bound for the maximum likelihood objective that is amenable to gradient descent. We refer to Kerrigan et al. (2022) for the proof and state the result here.

**Proposition B.12.** *For some fixed cellular context and perturbation $(c, \tau)$, let $P_0 := T_\#\mathbf{P}_{c,\tau}$ be the $\mathcal{H}_k$-valued pushforward measure of the $\mathcal{P}(\mathcal{X})$-valued cell population distribution via the kernel mean embedding map $T : p \mapsto \mathbb{E}_{x\sim p}[k(\cdot, x)]$. Let $Q_{0:T}^\theta(d\boldsymbol{\mu}_{0:T}) := Q_T^\theta(d\boldsymbol{\mu}_T)\prod_{t=1}^T Q_{t-1|t}^\theta(d\boldsymbol{\mu}_{t-1} \mid \boldsymbol{\mu}_t)$ be the $\mathcal{H}_k^{T+1}$-valued path measure of the reverse process parametrized by our generative model $\theta$. Define the marginal probability under our generative model of observing some function $\boldsymbol{\mu}_0 \in \mathcal{H}_k$ as $q_\theta(\boldsymbol{\mu}_0) := \int_{\mathcal{H}_k^T} Q_{0:T}^\theta(d\boldsymbol{\mu}_{0:T})$. Specifically, we marginalize $\boldsymbol{\mu}_1, \ldots, \boldsymbol{\mu}_T$. Then, we have the variational lower bound for training*

$$\mathbb{E}_{\boldsymbol{\mu}_0\sim P_0}[-\log q_\theta(\boldsymbol{\mu}_0)] \leq \sum_{t=1}^T \mathbb{E}_{\boldsymbol{\mu}_0\sim P_0, \boldsymbol{\mu}_t\sim P_{t|0}}[KL(P_{t-1|t,0}(\boldsymbol{\mu}_{t-1} \mid \boldsymbol{\mu}_t, \boldsymbol{\mu}_0)||Q_{t-1|t}^\theta(\boldsymbol{\mu}_{t-1} \mid \boldsymbol{\mu}_t))] + c, \tag{21}$$

*where $c \in \mathbb{R}$ is a constant independent of $\theta$.*

Again, similar to the derivation of the loss function for diffusion models in Euclidean space, we sample a random time $t \sim \mathcal{U}\{1, \ldots, T\}$ and obtain our simple loss

$$\mathcal{L}_t(\theta) := \mathbb{E}_{t, \boldsymbol{\mu}_0, \boldsymbol{\mu}_t}[\mathrm{KL}(P_{t-1|t,0}(\boldsymbol{\mu}_{t-1} \mid \boldsymbol{\mu}_t, \boldsymbol{\mu}_0) || Q_{t-1|t}^\theta(\boldsymbol{\mu}_{t-1} \mid \boldsymbol{\mu}_t))]. \tag{22}$$

In order to compute the KL divergence term between these two Gaussian measures, we must first ensure they share the same covariance operator. To do so, we set $C_t^\theta = \tilde{\beta}_t C$. Now, applying Lem. B.11 and parameterizing our model to predict $\boldsymbol{\mu}_0$, our simple loss reduces even further to

$$\mathcal{L}_t \propto \mathbb{E}_{t, \boldsymbol{\mu}_0, \boldsymbol{\mu}_t}[\|\boldsymbol{\mu}_0 - \boldsymbol{\mu}_\theta(\boldsymbol{\mu}_t, t)\|_{\mathcal{H}_k}^2]. \tag{23}$$

**Conditional reverse process.** The derivation above defines an unconditional diffusion objective over perturbed cell distributions. To incorporate the conditions including the control cell distribution, cellular context $c$ and perturbation types $\tau$, we follow the standard formulation of conditional diffusion models and classifier-free guidance (Ho & Salimans, 2022), in which the data is drawn jointly with the conditioning information, and the only modification to the model are the additional conditions conditions as input:

$$\mathcal{L}_t \propto \mathbb{E}_{t, \boldsymbol{\mu}_0, \boldsymbol{\mu}_t}[\|\boldsymbol{\mu}_0 - \boldsymbol{\mu}_\theta(\boldsymbol{\mu}_t, t, \boldsymbol{\mu}_c, c, \tau)\|_{\mathcal{H}_k}^2]]. \tag{24}$$

B.1.7. TRACTABLE TRAINING AND SAMPLING.

**Equivalent objective in cell space.** As already discussed in Sec. 4.2 and App. B.1.3, in practice, we replace the probability distribution by empirical measures induced by batches of cell sets.

We recall that $\boldsymbol{\mu}_0 = \boldsymbol{\mu}_{c,\tau}$ (see App. B.1.5). Because $\boldsymbol{\mu}_0$ is a kernel mean embedding, the squared RKHS norm corresponds to the MMD between the underlying distributions back to the single-cell space under kernel $k$ (a property stated in Lem. B.1):

$$\|\boldsymbol{\mu}_0 - \boldsymbol{\mu}_\theta(\boldsymbol{\mu}_t, t, \boldsymbol{\mu}_c, c, \tau)\|_{\mathcal{H}_k}^2 = \mathrm{MMD}_k^2(\tilde{P}_{\mathrm{pert}}, \tilde{P}_\theta), \tag{25}$$

where $\tilde{P}_{\mathrm{pert}}$ denotes the empirical distribution for a perturbed cell batch $B_{\mathrm{pert}} = \{\mathbf{x}_1, \mathbf{x}_2, \ldots, \mathbf{x}_m\} \subset X_{\mathrm{pert}}$, and $\tilde{P}_\theta := \frac{1}{m} \sum_{j=1}^m \delta_{\mathbf{x}_j^\theta}$ refers to the empirical distribution for the neural network predicted cells $B_\theta = \{\mathbf{x}_1^\theta, \ldots, \mathbf{x}_m^\theta\}$.

This yields a principled, distribution-aware training signal that can be directly applied in the single-cell space, which directly penalizes density shifts, subpopulation reweighting, and other distributional effects that cannot be reduced to cell-wise reconstruction. We next describe how to tractably inject noise to sample $\boldsymbol{\mu}_t$ efficiently by only needing to add Gaussian noise in the batched cell space $\mathbb{R}^{m \times |\mathcal{G}|}$.

**Tractable sampling of Gaussian random elements.** The forward diffusion process requires adding a sequence of i.i.d. Gaussian random elements sampled from $\mathcal{N}_{\mathcal{H}_k}(0, C)$. Directly sampling such an object is intractable as the Gaussian measure is function-valued. In this section, we will show that adding $\mathbb{R}^{|\mathcal{G}|}$-valued Gaussian noise directly to the cells that form the distribution before being embedded in $\mathcal{H}_k$ is, up to a first-order approximation, distributionally the same as adding a Gaussian random element.

Specifically, consider the simplest case, a batch of cells $\mathbf{x}_1, \ldots, \mathbf{x}_m \in \mathbb{R}^{|\mathcal{G}|}$ whose kernel mean embedding $\boldsymbol{\mu}_0 := \frac{1}{m} \sum_{j=1}^m k(\mathbf{x}_j, \cdot) \in \mathcal{H}_k$ should be noised to $\boldsymbol{\mu}_0 + \Xi_t$, $\Xi_t \sim \mathcal{N}_{\mathcal{H}_k}(0, C)$. Now consider the embedded function constructed by adding i.i.d. Gaussian noise to each cell,

$$\tilde{\boldsymbol{\mu}} := \frac{1}{m} \sum_{j=1}^m k(\mathbf{x}_j', \cdot), \ \mathbf{x}_j' = \mathbf{x}_j + \varepsilon_j, \ \varepsilon_j \overset{i.i.d.}{\sim} \mathcal{N}(0, I) \ \forall j \in \{1, \ldots, m\}. \tag{26}$$

Now, it suffices to show that $\mathrm{Law}(\tilde{\boldsymbol{\mu}} - \boldsymbol{\mu}_0) \approx \mathcal{N}_{\mathcal{H}_k}(0, C)$. First, we assume that $k(\mathbf{x}, \cdot) : \mathcal{X} \to \mathbb{R}$ is differentiable with a $\|\cdot\|_{\mathcal{H}_k}$-bounded derivative in its first argument for all $\mathbf{x} \in \mathcal{X}$. This easily holds for all common choices of kernels, including the energy distance kernel. This is because for any gene, its gene count is necessarily upper bounded due to fundamental physical and biological constraints. Now, define the difference of the two functions $\Delta := \frac{1}{m} \sum_{j=1}^m \nabla_{\mathbf{x}} k(\mathbf{x}_j, \cdot)^\top \varepsilon_j \in \mathcal{H}_k$. It is important to note that $\nabla_{\mathbf{x}} k(\mathbf{x}_j, \cdot)$ is a vector of $\mathcal{H}_k$-valued elements of length $|\mathcal{G}|$. By expanding $\tilde{\boldsymbol{\mu}}$ in its first argument about $\mathbf{x}_j$ for each $j$, we have $\tilde{\boldsymbol{\mu}} - \boldsymbol{\mu}_0 = \Delta + \frac{1}{m} \sum_{j=1}^m o(\|\varepsilon_j\|) = \Delta + o(\|\varepsilon_1\|)$.

**Lemma B.13.** $\Delta$ *is a $\mathcal{H}_k$-valued Gaussian random element.*

*Proof.* Define the linear operator $T_j : \mathbb{R}^{|\mathcal{G}|} \to \mathcal{H}_k$ by

$$T_j v := \nabla_{\mathbf{x}} k(\mathbf{x}_j, \cdot)^\top v = \sum_{r=1}^{|\mathcal{G}|} v_r \partial_{\mathbf{x}^r} k(\mathbf{x}_j, \cdot),$$

where $\mathbf{x}^r$ denotes the $r$-th coordinate of the vector $\mathbf{x}$. We have boundedness of $T_j$ via an application of Cauchy-Schwarz with our bounded derivative assumption on $k(\mathbf{x}_j, \cdot)$:

$$\|T_j v\|_{\mathcal{H}_k} \leq \left( \sum_{r=1}^{|\mathcal{G}|} \|\partial_{\mathbf{x}^r} k(\mathbf{x}_j, \cdot)\|_{\mathcal{H}_k}^2 \right)^{1/2} \|v\| < \infty,$$

which is a sufficient condition for the adjoint $T_j^* : \mathcal{H}_k \to \mathbb{R}^{|\mathcal{G}|}$ to exist. Now, define $Z_j := T_j \varepsilon_j \in \mathcal{H}_k$ for each $j$, then we have $\Delta = \frac{1}{m} \sum_{j=1}^m Z_j$. Recall the definition of a Gaussian random element, let $h \in \mathcal{H}_k$ be arbitrary, we want to show that $\langle \Delta, h \rangle_{\mathcal{H}_k}$ is a centered Gaussian on $\mathbb{R}$.

$$\langle \Delta, h \rangle_{\mathcal{H}_k} = \frac{1}{m} \sum_{j=1}^m \langle T_j \varepsilon_j, h \rangle_{\mathcal{H}_k} = \frac{1}{m} \sum_{j=1}^m \langle \varepsilon_j, T_j^* h \rangle,$$

where $\langle \cdot, \cdot \rangle$ is the standard Euclidean inner product. Since each individual term is a linear functional of $\varepsilon_j$ and the sum of independent Gaussians is also a Gaussian, we have $\langle \Delta, h \rangle_{\mathcal{H}_k}$ is a Gaussian in $\mathbb{R}$. Also, since each $\varepsilon_j$ is mean-zero and hence $\langle \Delta, h \rangle_{\mathcal{H}_k}$ is a mean-zero Gaussian for every $h$, we know that $\Delta$ is a Gaussian random element with mean element $0$. ∎

Now, we find an explicit form for the covariance operator of this Gaussian measure and we are done since approximate sampling from $\mathcal{H}_k$-valued Gaussian measures with covariance operators $\beta C$ for some scalar $\beta > 0$ follows immediately from the linearity of the arguments in our proofs.

**Proposition B.14.** $\Delta \sim \mathcal{N}_{\mathcal{H}_k}(0, C)$, *where* $C = \frac{1}{m^2} \sum_{j=1}^m T_j T_j^*$.

*Proof.* Notice that the covariance operator $C$ is defined as

$$\langle Cf, g \rangle_{\mathcal{H}_k} = \mathbb{E}[\langle \Delta, f \rangle_{\mathcal{H}_k} \langle \Delta, g \rangle_{\mathcal{H}_k}],$$

for any $f, g \in \mathcal{H}_k$. By independence of cells in the batch, we have

$$\mathbb{E}[\langle \Delta, f \rangle_{\mathcal{H}_k} \langle \Delta, g \rangle_{\mathcal{H}_k}] = \frac{1}{m^2} \sum_{j=1}^m \mathbb{E}[\langle \varepsilon_j, T_j^* f \rangle \langle \varepsilon_j, T_j^* g \rangle] = \frac{1}{m^2} \sum_{j=1}^m \langle T_j^* f, T_j^* g \rangle,$$

where we used $\mathbb{E}[\langle \varepsilon, a \rangle \langle \varepsilon, b \rangle] = \langle a, b \rangle$ for any $a, b \in \mathbb{R}^{|\mathcal{G}|}$ and $\varepsilon \sim \mathcal{N}(0, I)$. Hence, we have the equality

$$\langle Cf, g \rangle_{\mathcal{H}_k} = \mathbb{E}[\langle \Delta, f \rangle_{\mathcal{H}_k} \langle \Delta, g \rangle_{\mathcal{H}_k}] = \frac{1}{m^2} \sum_{j=1}^m \langle T_j^* f, T_j^* g \rangle = \frac{1}{m^2} \sum_{j=1}^m \langle T_j T_j^* f, g \rangle_{\mathcal{H}_k} = \left\langle \frac{1}{m^2} \sum_{j=1}^m T_j T_j^* f, g \right\rangle_{\mathcal{H}_k}.$$

Hence, by matching terms, we have our result as needed. ∎

## B.2. Diffusion Implementation Details

As discussed in Sec. 4.4, we implement our diffusion framework directly in cell space for tractable computation. We adopt a variance-preserving (VP) diffusion and train an $x_0$-predictor to recover the clean perturbed cell batch from its noised version. Unless stated otherwise, we use the same diffusion configuration for (i) training from scratch, (ii) downstream fine-tuning, and (iii) marginal pretraining.

**Notation.** Let $B_{\text{pert}}$ be a sampled perturbed cell batch and $B_{\text{ctrl}}$ be a control batch, both associated with the same cellular context $c$; $B_{\text{pert}}$ is further associated with a perturbation type $\tau$. For convenience, we represent each batch as a matrix: $B_{\text{ctrl}} \in \mathbb{R}^{m \times |\mathcal{G}|}$ and $B_{\text{pert}} \in \mathbb{R}^{m \times |\mathcal{G}|}$, where $m$ is the batch size and $|\mathcal{G}|$ is the number of genes.

We denote the clean data as $B_0 := B_{\text{pert}}$, the noised batch at diffusion step $t$ as $B_t$, and the model predicted batch as $B_\theta$. For conditional generation, we write the conditioning information as $y := (B_{\text{ctrl}}, c, \tau)$, and our network $f_\theta$ is parameterized as an "$x_0$-predictor". For $f_\theta$, we also apply a non-negative output head (*e.g.*, `ReLU` or `softplus`) to enforce non-negativity of predicted expression values.

### B.2.1. TRAINING

**Forward noising process.** We discretize the VP diffusion into $T=1000$ steps with a linear variance schedule $\{\beta_t\}_{t=1}^T \subset (0, 1)$. Define the cumulative noise level $\alpha_t := \prod_{s=1}^t (1 - \beta_s)$. The forward noising process is given by

$$q(B_t \mid B_0) = \mathcal{N}\big(\text{vec}(B_t); \sqrt{\alpha_t}\text{vec}(B_0), (1 - \alpha_t)I_{m|\mathcal{G}|}\big), \quad B_t = \sqrt{\alpha_t}B_0 + \sqrt{1 - \alpha_t}\varepsilon, \quad \varepsilon_{ij} \overset{\text{i.i.d.}}{\sim} \mathcal{N}(0, 1). \quad (27)$$

During training, we sample timesteps uniformly: $t \sim \text{Unif}\{1, \ldots, T\}$.

$x_0$**-prediction parameterization.** We adopt an $x_0$-prediction parameterization and train the model to directly predict the clean perturbed cell batch. Given a noised batch $B_t$ at timestep $t$ and conditioning information $y := (B_{\text{ctrl}}, c, \tau)$, the model output is

$$B_\theta = f_\theta(B_t, t; y). \quad (28)$$

The corresponding noise prediction implied by $B_\theta$ is

$$\hat{\varepsilon}_\theta(B_t, t; y) = \frac{B_t - \sqrt{\alpha_t}\, B_\theta}{\sqrt{1 - \alpha_t}}. \quad (29)$$

**Objective.** The training objective combines a standard DDPM mean-squared error (MSE) loss with an MMD-based loss derived from our diffusion framework over cell distributions in $\mathcal{H}_k$:

$$\mathcal{L}_{\text{total}} = \mathcal{L}_{\text{MMD}} + \lambda_{\text{MSE}}\, \mathcal{L}_{\text{MSE}}, \qquad \lambda_{\text{MSE}} = 1 \text{ by default.} \quad (30)$$

The MSE term is a cell-wise reconstruction objective on the perturbed batch:

$$\mathcal{L}_{\text{MSE}} = \mathbb{E}_{B_0, y, t, \varepsilon}\Big[\|B_0 - f_\theta(B_t, t; y)\|_2^2\Big], \quad (31)$$

where $B_t$ is generated by Eqn. (27).

To align training with our distribution-level diffusion formulation in $\mathcal{H}_k$, we incorporate the MMD term using the energy distance kernel. Let $\tilde{P}_{B_0}$ and $\tilde{P}_{B_\theta}$ denote the empirical distributions induced by the true and predicted cell batches, respectively. The energy distance between them is

$$\text{ED}(\tilde{P}_{B_0}, \tilde{P}_{B_\theta}) = 2\,\mathbb{E}\|X - Y\| - \mathbb{E}\|X - X'\| - \mathbb{E}\|Y - Y'\|, \quad X, X' \sim \tilde{P}_{B_0}, \ Y, Y' \sim \tilde{P}_{B_\theta}, \quad (32)$$

and we set $\mathcal{L}_{\text{MMD}} := \text{ED}(\tilde{P}_{B_0}, \tilde{P}_{B_\theta})$ in Eqn. (30).

**Self-conditioning.** We employ self-conditioning with probability $p_{\text{sc}}$ to stabilize and improve training. Specifically, with probability $p_{\text{sc}}$, we first obtain a stop-gradient prediction $\bar{B}_\theta = \text{sg}(f_\theta(B_t, t; y))$ and feed it back to the network by concatenation:

$$B_\theta = f_\theta\big([B_t, \bar{B}_\theta], t; y\big), \qquad \bar{B}_\theta = \text{sg}(f_\theta(B_t, t; y)), \quad (33)$$

where $\text{sg}(\cdot)$ denotes the stop-gradient operator. When self-conditioning is disabled, the model reduces to the standard formulation $B_\theta = f_\theta(B_t, t; y)$.

**Classifier-free guidance (CFG) dropout during training.** To enable CFG at inference time, we randomly drop the *metadata* condition with probability $p_{\text{drop}}$. Specifically, the cellular context $c$ and perturbation label $\tau$ are masked, while the control cell batch $B_{\text{ctrl}}$ is always provided to the network and is not treated as a drop-able condition.

**Exponential Moving Average (EMA).** For evaluation and sampling, we maintain an exponential moving average (EMA) of model parameters with decay 0.99, updated every 10 steps.

### B.2.2. SAMPLING

**Reverse-time generation.** At inference, we initialize the reverse process from $\text{vec}(B_T) \sim \mathcal{N}(0, I_{m|\mathcal{G}|})$ and run the reverse process using DDIM, with an optional DDPM sampler. Given a decreasing timestep sequence $\{t_k\}_{k=1}^K$ (*e.g.*, $K=100$ for

fast sampling or $K=1000$ for full sampling), we perform DDIM sampling with noise parameter $\eta \in [0, 1]$, where $\eta = 0$ corresponds to deterministic sampling. The DDIM update with $\eta = 0$ is given by

$$B_\theta^{(k)} = f_\theta(B_{t_k}, t_k; y), \qquad \hat{\varepsilon}^{(k)} = \frac{B_{t_k} - \sqrt{\alpha_{t_k}} B_\theta^{(k)}}{\sqrt{1 - \alpha_{t_k}}}, \qquad B_{t_{k+1}} = \sqrt{\alpha_{t_{k+1}}} B_\theta^{(k)} + \sqrt{1 - \alpha_{t_{k+1}}} \, \hat{\varepsilon}^{(k)}. \qquad (34)$$

Self-conditioning is enabled during sampling by feeding the previous $B_\theta^{(k-1)}$ into the next step, consistent with Eqn. (33).

**Classifier-free guidance.** We apply CFG in $\varepsilon$-space:

$$\hat{\varepsilon}_{\text{cfg}} = (1 + w) \, \hat{\varepsilon}_c - w \, \hat{\varepsilon}_u, \qquad (35)$$

where $\hat{\varepsilon}_c$ and $\hat{\varepsilon}_u$ are the conditional and unconditional noise predictions, respectively. In our default setting, the unconditional branch drops the metadata conditioning (cellular context $c$ and perturbation label $\tau$), while *retaining* the control cell batch $B_{\text{ctrl}}$. This ensures that generation remains anchored to the matched control population even under guidance.

## B.3. Architecture Implementation Details

We instantiate our diffusion backbone as a multi-modal DiT (MM-DiT) transformer for perturbed cell generation. The model predicts the denoised perturbed cell batch $B_0$ (or equivalently the noise $\hat{\varepsilon}$ via Eqn. (29)) from a noised batch $B_t$, while conditioning on the matched control batch $B_{\text{ctrl}}$ and associated metadata $(c, \tau)$. Specifically, the perturbed batch and control batch are fed into the two token streams in MM-DiT, while $(c, \tau)$ are fed as conditioning encodings.

**Inputs and outputs.** Let $B_t \in \mathbb{R}^{m \times |\mathcal{G}|}$ denote a batch of noised perturbed cell profiles, where $m$ is the batch size, and $|\mathcal{G}|$ is the gene dimension (*e.g.*, $|\mathcal{G}|=2000$ HVGs). We condition on $y := (B_{\text{ctrl}}, c, \tau)$, where $B_{\text{ctrl}}$ is a sampled control batch, $c$ denotes cellular context (*i.e.*, cell type and optionally experimental batch), and $\tau$ denotes the perturbation type (and dosage when applicable). Note that both $B_{\text{ctrl}}$ and $B_{\text{pert}}$ are sampled under the same cellular context $c$, with $B_{\text{pert}}$ additionally associated with perturbation $\tau$. If the number of available cells is smaller than the batch size $m$, sampling is performed with replacement to ensure a consistent batch size.

The network is parameterized as an $x_0$-predictor

$$B_\theta = f_\theta(B_t, t; y), \qquad (36)$$

and we apply an `ReLU` head to $B_\theta$ when enforcing non-negative expression.

**Tokenization and embedding.** Each cell expression vector $\mathbf{x} \in \mathbb{R}^{|\mathcal{G}|}$ is first projected into the model dimension $d$ through a linear input projection, producing a sequence of tokens $h \in \mathbb{R}^{m \times d}$. Thus, $B_{\text{pert}}$ results in perturbed tokens $h_{\text{pert}}$ and $B_{\text{ctrl}}$ results in control tokens $h_{\text{ctrl}}$. We adopt this cell-wise tokenization scheme—treating each cell as a token and a batch of cells as a "sentence"—which is substantially more efficient when the gene dimension $|\mathcal{G}|$ is large.

Optionally, gene-level semantic information can be incorporated by associating each gene with an external embedding. Specifically, a short gene summary generated by `gpt-5-mini` is encoded using a pretrained text embedding model (`text-embedding-3-large`) and used as the corresponding gene embedding. When gene embeddings are disabled, all genes share a single dummy embedding. In practice, for the final reported model, we use the shared dummy embedding for simplicity.

The diffusion timestep $t$ is embedded by a standard sinusoidal timestep embedding followed by an MLP, producing $e_t \in \mathbb{R}^d$.

**Condition embedding.** We encode the metadata $(c, \tau)$ into a condition vector $e_y \in \mathbb{R}^d$ using a covariate encoder (App. B.3). When semantic embeddings are available (*e.g.*, drug/gene embeddings), they are linearly projected to dimension $d$ and combined with learned embeddings. The resulting representations are concatenated with the diffusion time embedding $e_t$ and passed through an MLP to form a global conditioning vector,

$$s := \text{MLP}\big([e_t, e_y]\big) \in \mathbb{R}^d, \qquad (37)$$

which is used to modulate every transformer block via AdaLN-Zero (described below).

**MM-DiT fusion blocks.** Our backbone consists of $L$ transformer blocks. Each block jointly updates the perturbed and control token streams via a symmetric MM-DiT-style fusion mechanism. Within each block, both streams are processed by two sequential sublayers: a multi-head self-attention (MSA) sublayer followed by an MLP sublayer.

For each stream and each sublayer, we use AdaLN-Zero (Peebles & Xie, 2023) conditioned on the context $s$ to produce a scale, shift, and a residual gate:

$$(\beta_\star^{\text{attn}}, \gamma_\star^{\text{attn}}, g_\star^{\text{attn}}, \beta_\star^{\text{mlp}}, \gamma_\star^{\text{mlp}}, g_\star^{\text{mlp}}) = f_{\theta,\star}(s), \quad \star \in \{\text{pert}, \text{ctrl}\}, \tag{38}$$

and define $\text{Mod}(h; \beta, \gamma) = \gamma \odot \text{LN}(h) + \beta$.

We first modulate the normalized inputs for the attention sublayer

$$\tilde{h}_{\text{pert}}^{\text{attn}} = \text{Mod}(h_{\text{pert}}; \beta_{\text{pert}}^{\text{attn}}, \gamma_{\text{pert}}^{\text{attn}}), \quad \tilde{h}_{\text{ctrl}}^{\text{attn}} = \text{Mod}(h_{\text{ctrl}}; \beta_{\text{ctrl}}^{\text{attn}}, \gamma_{\text{ctrl}}^{\text{attn}}), \tag{39}$$

and concatenate the two streams along the feature dimension

$$u = \left[\tilde{h}_{\text{pert}}^{\text{attn}}; \tilde{h}_{\text{ctrl}}^{\text{attn}}\right] \in \mathbb{R}^{m \times 2d}, \tag{40}$$

and processed by a shared multi-head self-attention module. The output is split back into perturbed and control components,

$$u' = \text{MSA}(u), \quad [\Delta h_{\text{pert}}; \Delta h_{\text{ctrl}}] = \text{Split}(u'). \tag{41}$$

The two streams are then updated by gated residual connections:

$$h_{\text{pert}} \leftarrow h_{\text{pert}} + g_{\text{pert}}^{\text{attn}} \odot \Delta h_{\text{pert}}, \qquad h_{\text{ctrl}} \leftarrow h_{\text{ctrl}} + g_{\text{ctrl}}^{\text{attn}} \odot \Delta h_{\text{ctrl}}. \tag{42}$$

Next, each stream is passed through an MLP sublayer independently, again preceded by AdaLN-Zero modulation and followed by a gated residual update:

$$h_\star \leftarrow h_\star + g_\star^{\text{mlp}} \odot \text{MLP}_\star\left(\text{Mod}(h_\star; \beta_\star^{\text{mlp}}, \gamma_\star^{\text{mlp}})\right), \quad \star \in \{\text{pert}, \text{ctrl}\}. \tag{43}$$

The separate residual gates for the attention $g_{\text{attn}}(s)$ and MLP sublayers $g_{\text{mlp}}(s)$ are used to control their information contributions.

Although both streams are updated throughout the network, only the perturbed stream $h_{\text{pert}}$ is connected to the denoising head and contributes to the diffusion loss. The control stream is updated within blocks to enhance conditioning capacity, but no denoising head or reconstruction loss is imposed on $h_{\text{ctrl}}$.

**AdaLN-Zero conditioning and initialization.** Each transformer block uses AdaLN-Zero to inject conditioning. For a block input $h$, AdaLN-Zero applies an affine modulation to LayerNorm-normalized activations, with scale and shift predicted from the conditioning variable $s$:

$$\text{AdaLN}(h; s) = \text{LN}(h) \odot \left(1 + \Gamma(s)\right) + \Delta(s), \tag{44}$$

where $\Gamma(\cdot)$ and $\Delta(\cdot)$ are linear projections of a small MLP on $s$.

Following standard AdaLN-Zero practice, the final linear layer of the modulation network is initialized to zero, such that the scale, shift, and residual gates are near zero at initialization. As a result, each block is initialized close to an identity mapping, which stabilizes optimization in deep conditional diffusion models.

**Classifier-free guidance compatibility.** To support CFG at sampling time (App. B.2), we apply conditional dropout during training: with probability $p_{\text{drop}}$ we drop metadata embeddings $e_y$ (while retaining the control context), producing an unconditional branch that shares the same architecture. This enables Eqn. (35) without requiring any auxiliary classifier.

**Covariate Encoder.** We implement a covariate encoder (`CovEncoder`) that maps experimental and biological metadata—including perturbation identity, dosage, cell type, and optional experimental batch information— into dense vectors. Each covariate is embedded independently, and all resulting embeddings are concatenated and linearly projected into the model dimension.

**Perturbation Identity Embeddings.** We provide multiple options to encode perturbation identity, depending on the perturbation type (cytokine, drug, or gene) and dataset characteristics. For perturbation identity, we support both semantic embeddings and simple one-hot encodings, allowing flexible choices across datasets.

- **Cytokine Embeddings (ESM2).** For cytokine perturbations, which correspond to protein molecules, we extract semantic embeddings using a pretrained protein language model (ESM2). Cytokine embeddings are loaded from disk, indexed by the cytokien identity.

- **Drug Semantic Embeddings (ChemBERTa + Dose).** For chemical perturbations, we load pre-computed semantic embeddings (e.g., the [CLS] representation from a ChemBERTa-like encoder (Chithrananda et al., 2020)) from disk and construct an embedding table indexed by the drug identity. The control perturbation embedding is set to the mean embedding across all drugs. To model dosage effects, we discretize observed dose values into a vocabulary and learn a corresponding dose embedding. The final perturbation representation is obtained by summing the drug semantic embedding and the associated dose embedding.

- **Gene Perturbation Embeddings (GenePT / LLM).** For gene perturbations, we support loading semantic gene embeddings from disk, including GenePT (Chen & Zou, 2024) embeddings or LLM-derived embeddings (*e.g.*, `text-embedding-3-large`) computed from LLM-generated gene summaries (`gpt-5-mini`). These embeddings are stored in a frozen lookup table aligned with gene identity, with the "non-targeting" control embedding set to the mean embedding.

**Cell Type Embeddings.** We support a frozen semantic cell type embedding loaded from disk, derived from LLM-generated cell type summaries using `gpt-5` and encoded by LLM like `text-embedding-3-large`.

**Experimental Batch Embeddings (Optional).** If batch covariates are provided, we use a learnable embedding table indexed by experimental batch identity. We assume that the batch IDs are mapped into a contiguous range $\{0, \ldots, N_{\text{batch}} - 1\}$ prior to feeding into `CovEncoder`.

### B.4. Score Matching Discussion Details

We also provide a perspective to interpret our MMD-based distribution alignment objective through the lens of score matching by analyzing its local behavior around a reference cell population. Let $x$ denote a fixed reference cell set (*e.g.*, a batch of training data), and let $x'$ be a generated cell set produced by the model. We consider the squared MMD, $\text{MMD}_k^2(x', x)$, as a function of $x'$ and study its Taylor expansion around $x' = x$.

$$
\begin{aligned}
\text{MMD}_k^2(x', x) = \ &\text{MMD}_k^2(x, x) \\
&+ \langle x' - x, \nabla_{x'}\text{MMD}_k^2(x', x)|_{x'=x} \rangle \\
&+ \frac{1}{2}\langle x' - x, \nabla_{x'}^2\text{MMD}_k^2(x', x)|_{x'=x}(x' - x) \rangle \\
&+ o(\|x' - x\|_2^2),
\end{aligned}
$$

The zeroth-order term is constant with respect to the model parameters and can be ignored during optimization. The first-order term vanishes at $x' = x$, since $\text{MMD}_k^2(\cdot, x)$ attains its minimum there and behaves locally as a convex, distance-like function. Consequently, the second-order term dominates in a neighborhood of the optimum, yielding the local approximation

$$
\text{MMD}_k^2(x', x) = \|x' - x\|_H^2 + o(\|x' - x\|_2^2),
$$

where $H := \nabla_{x'}^2\text{MMD}_k^2(x', x)|_{x'=x}$, and $\|x\|_H^2 := x^\top H x$. This result shows that, up to second order, minimizing MMD is equivalent to minimizing a matrix-weighted mean squared error, where the weighting matrix $H$ depends on the kernel and the local geometry of the reference population. Unlike a standard Euclidean MSE, this Hessian-induced norm emphasizes directions corresponding to informative higher-order population statistics encoded by the kernel, rather than treating all directions uniformly.

This local quadratic structure naturally connects MMD-based distribution alignment to score matching. In particular, minimizing a weighted quadratic deviation between generated and reference samples corresponds to matching the score of the data distribution under a non-Euclidean geometry induced by $H$. In Section B.4, we formalize this connection by showing that score matching under a general symmetric positive definite norm $\|\cdot\|_H$ admits an equivalent denoising score-matching formulation. To this end, we abstract the local second-order structure of the MMD objective into a generic sample-level matching function.

Specifically, let $x \in \mathbb{R}^d$ be a random variable with density $p$ and let $\tilde{x} := \alpha x + \beta \epsilon$, where $(\alpha, \beta) \in [0, 1]^2$, and $\epsilon \sim \mathcal{N}(0, I)$. We can factor the joint density as $p_{\tilde{x}, x} = p_{\tilde{x}|x} p_x$. Note that we can also write the marginal of $\tilde{x}$ as $p(\tilde{x}) = \int p(\tilde{x} \mid x) p(x) dx$.

To simplify the analysis, we introduce a generic smooth matching function $M : \mathbb{R}^d \times \mathbb{R}^d \to \mathbb{R}$, which captures the local second-order behavior of the squared MMD objective derived above. Let $M$ be infinitely differentiable. Suppose $M$ only depends on the 2-norm between the two arguments, *i.e.* $M(x, x') = M(\|x' - x\|_2)$. Also suppose $M$ is strongly convex with respect to the first argument with the second argument fixed. Fix some $x$ in the second argument of $M$, then we can compute the second-order Taylor expansion about $x$ as

$$M(x', x) = M(x, x) + \langle x' - x, \nabla_{x'} M(x', x)|_{x'=x} \rangle + \frac{1}{2} \langle x' - x, \nabla^2_{x'} M(x', x)|_{x'=x} (x' - x) \rangle + o(\|x' - x\|^2).$$

Importantly, from our assumptions, we have that the Hessian $\nabla^2_{x'} M(x', x)|_{x'=x}$ is symmetric and positive definite. Let $H = \nabla^2_{x'} M(x'x)|_{x'=x}$.

**Proposition B.15** (Augmented Denoising Score Matching)**.**

$$\arg \min_\theta \mathbb{E}_{\tilde{x}}[\|\nabla \log p(\tilde{x}) - f(\tilde{x})\|^2_H] = \arg \min_\theta \mathbb{E}_{\tilde{x}, x}[\|\nabla_{\tilde{x}} \log p(\tilde{x} \mid x) - f(\tilde{x})\|^2_H]$$

*Proof.* We want to show $L(f) := \frac{1}{2} \mathbb{E}_{\tilde{x}}[\|\nabla \log p(\tilde{x}) - f(\tilde{x})\|^2_H]$ is equal, up to a constant not dependent on $f$, to $L'(f) = \frac{1}{2} \mathbb{E}_{\tilde{x}, x}[\|\nabla_{\tilde{x}} \log p(\tilde{x} \mid x) - f(\tilde{x})\|^2_H]$, where $\|x\|^2_H = x^\top H x$.

We start with $L(f)$:

$$L(f) = \frac{1}{2} \mathbb{E}_{\tilde{x}}[\|\nabla \log p(\tilde{x}) - f(\tilde{x})\|^2_H]$$

$$= \frac{1}{2} \int f(\tilde{x})^\top H f(\tilde{x}) p(\tilde{x}) d\tilde{x} - \int \langle f(\tilde{x}) H^\top, \underbrace{p(\tilde{x}) \nabla \log p(\tilde{x})}_{=\nabla p(\tilde{x})} \rangle d\tilde{x} + \underbrace{\frac{1}{2} \int \|\nabla \log p(\tilde{x})\|^2_H p(\tilde{x}) d\tilde{x}}_{=:C_1}$$

$$= \frac{1}{2} \mathbb{E}_{\tilde{x}}[f(\tilde{x})^\top H f(\tilde{x})] - \int \langle f(\tilde{x}) H^\top, \nabla p(\tilde{x}) \rangle d\tilde{x} + C_1$$

Focusing on the second term, we have

$$\int \langle f(\tilde{x}) H^\top, \nabla p(\tilde{x}) \rangle d\tilde{x} = \int \langle f(\tilde{x}) H^\top, \underbrace{\nabla_{\tilde{x}} \int}_{\text{swap}} p(\tilde{x} \mid x) p(x) dx \rangle d\tilde{x}$$

$$= \iint \langle f(\tilde{x}) H^\top, \underbrace{\nabla_{\tilde{x}} p(\tilde{x} \mid x)}_{p(\tilde{x}|x) \nabla_{\tilde{x}} \log p(\tilde{x}|x)} \rangle p(x) dx d\tilde{x}$$

$$= \iint \langle f(\tilde{x}) H^\top, \nabla_{\tilde{x}} \log p(\tilde{x} \mid x) \rangle \underbrace{p(\tilde{x} \mid x) p(x)}_{=p(\tilde{x}, x)} dx d\tilde{x}$$

$$= \mathbb{E}_{(\tilde{x}, x)} \left[ f(\tilde{x})^\top H (\nabla_{\tilde{x}} \log p(\tilde{x} \mid x)) \right]$$

Plugging this back in, and also adding in and subtracting a 'complete the square' term, we get

$$L(f) = \frac{1}{2} \mathbb{E}[f(\tilde{x})^\top H f(\tilde{x})] - \mathbb{E}[f(\tilde{x})^\top H (\nabla_{\tilde{x}} \log p(\tilde{x} \mid x))]$$

$$+ \frac{1}{2} \mathbb{E}[(\nabla_{\tilde{x}} \log p(\tilde{x} \mid x))^\top H (\nabla_{\tilde{x}} \log p(\tilde{x} \mid x))] - \underbrace{\frac{1}{2} \mathbb{E}[(\nabla_{\tilde{x}} \log p(\tilde{x} \mid x))^\top H (\nabla_{\tilde{x}} \log p(\tilde{x} \mid x))]}_{=:C_2} + C_1$$

$$= \frac{1}{2} \mathbb{E}[\|f(\tilde{x}) - \nabla_{\tilde{x}} \log p(\tilde{x} \mid x)\|^2_H] + C$$

$$= L'(f) + C,$$

where $C := C_1 - C_2$ doesn't depend on $f$, and we are done. ∎

# C. Experimental Details

## C.1. Training Details

**Pretraining and downstream data.** Detailed discussion for pretraining and dowmstream data composition and preprocessing can be found in App. A.2 and App. A.3, respectively.

**Optimization.** We optimize all models using AdamW with $(\beta_1, \beta_2) = (0.9, 0.98)$ and weight decay $10^{-2}$. The learning rate follows a cosine decay schedule with 200 warmup steps and a minimum plateau at $0.1\times$ the initial learning rate. We use a learning rate of $2 \times 10^{-4}$ for PBMC and Tahoe100M, $2 \times 10^{-3}$ for Replogle in downstream training and fine-tuning, and $2 \times 10^{-4}$ for pretraining. Gradients are clipped to a maximum global norm of 1.0.

**Early stopping for model checkpoint selection (training from scratch / finetuning).** As shown in Fig. 7 and Fig. 20, diffusion models are sensitive to the number of training steps, or equivalently, the training compute budget. This behavior is also commonly observed in image diffusion models (Ho et al., 2022; Baptista et al., 2025). To ensure a fair and consistent evaluation, we train all model configurations for a fixed 20k steps and select the best checkpoint based on performance evaluated every 2k steps on validation set. This applies to both training from scratch and finetuning from a pretrained model.

**Model checkpoint selection for pretraining.** As shown in Fig. 6, the pretraining dynamics differ across datasets. On PBMC, $R^2$ steadily improves with training steps, reflecting stable gains in capturing marginal perturbation patterns. In contrast, Replogle shows a sharp drop in $R^2$ after the first epoch, followed by stagnation. This discrepancy motivates our checkpoint selection strategy: for downstream finetuning, we select the end-of-first-epoch checkpoint, which consistently balances performance and stability across datasets.

**Input/output layer initialization during finetuning.** We consider three strategies for transferring the pretrained model to downstream finetuning: (1) directly leveraging the pretrained $12,626$-gene input/output layer weights for finetuning; (2) replacing the pretrained input/output layers for the $12,626$ gene vocabulary with randomly initialized layers; and (3) replacing the pretrained layers with newly initialized input/output layers restricted to a 2k-gene vocabulary. Empirically, we find that strategy strategy (1) performs best on PBMC, strategy (2) on Replogle, and strategy (3) on Tahoe-100M.

**Zero control stream during pretraining.** To enable a unified model architecture for both pretraining and finetuning, we simplify the input relative to training from scratch: the control token stream is set to all zeros, and conditioning is restricted to cellular context only.

**Dataset-specific linear transformation layer for pretraining.** To project high-dimensional gene expression vectors into a shared low-dimensional feature space, we apply a linear transformation $W \in \mathbb{R}^{|\mathcal{G}| \times D}$, where $D$ denotes the model feature dimension. Because datasets originate from different sources and exhibit distinct expression distributions, we use dataset-specific linear transformations rather than a single shared projection. Concretely, we instantiate separate linear layers for cells from PBMC, Tahoe100M, Replogle, and CellxGene.

**More details.** Besides the details above, App. B.2 describes diffusion implementation, and App. B.3 details the architecture.

## C.2. Metrics

We adopted the $R^2$ metric from CellFlow (Klein et al., 2025) and the entire evaluation framework Cell-Eval (Adduri et al., 2025) (version 0.6.6) from STATE to comprehensively assess perturbation predictions. This framework addresses a key challenge: since individual cells cannot be directly compared to a specific ground truth (due to biological variability and the destructive nature of sequencing), evaluation must rely on population-level statistics.

We categorize the leveraged metrics in two ways: (1) **Across the dimension of cells**: it measures **expression-level accuracy** by comparing statistical distributions between the population of predicted cells and the population of real observed cells; (2) **Across the dimension of genes**: it assesses **biologically meaningful differential patterns** by comparing the differentially expressed genes sets derived from predictions versus those derived from ground truth data.

To ensure fair comparison, we fix the control cells to be identical across predicted and ground-truth evaluations and across all our methods and baselines, so that differences arise solely from perturbed-cell predictions.

**Notations.** Let $M$ be the total number of perturbations. For each perturbation condition $\lambda$, we denote by $\{\mathbf{x}_{\lambda,i}^{\text{pert}}\}_{i=1}^{N_{\text{pert}}^{\lambda}}$ and $\{\mathbf{x}_{\lambda,i}^{\text{ctrl}}\}_{i=1}^{N_{\text{ctrl}}^{\lambda}}$ the ground-truth expression profiles for perturbed and control cells, respectively, and by $\{\hat{\mathbf{x}}_{\lambda,i}^{\text{pert}}\}_{i=1}^{N_{\text{pert}}^{\lambda}}$ the

model-predicted expression profiles for the perturbed cells. Empirically, Cell-Eval chooses $N_{\text{ctrl}}^\lambda = N_{\text{pert}}^\lambda$. We define the **relative perturbation effects** as the difference between the pseudobulk profiles for perturbed and control cells. Specifically, the pseudobulk profiles for ground-truth perturbed and control cells are defined as

$$\bar{\mathbf{x}}_\lambda^{\text{pert}} = \frac{1}{N_{\text{pert}}^\lambda} \sum_{i=1}^{N_{\text{pert}}^\lambda} \mathbf{x}_{\lambda,i}^{\text{pert}}, \quad \bar{\mathbf{x}}_\lambda^{\text{ctrl}} = \frac{1}{N_{\text{ctrl}}^\lambda} \sum_{i=1}^{N_{\text{ctrl}}^\lambda} \mathbf{x}_{\lambda,i}^{\text{ctrl}}, \tag{45}$$

respectively. And the pseudobulk profiles for the predicted perturbed cells are defined as

$$\bar{\hat{\mathbf{x}}}_\lambda^{\text{pert}} = \frac{1}{N_{\text{pert}}^\lambda} \sum_{i=1}^{N_{\text{pert}}^\lambda} \hat{\mathbf{x}}_{\lambda,i}^{\text{pert}}. \tag{46}$$

Then we can formally define the ground-truth perturbation effect as

$$\Delta\mathbf{x}_\lambda := \bar{\mathbf{x}}_\lambda^{\text{pert}} - \bar{\mathbf{x}}_\lambda^{\text{ctrl}}, \tag{47}$$

and the predicted perturbation effect as

$$\Delta\hat{\mathbf{x}}_\lambda := \bar{\hat{\mathbf{x}}}_\lambda^{\text{pert}} - \bar{\mathbf{x}}_\lambda^{\text{ctrl}}. \tag{48}$$

### C.2.1. AVERAGED EXPRESSION ACCURACY

**Coefficient of Determination ($R^2$).** $R^2$ measures the fraction of variance in the ground-truth data explained by model predictions, relative to the empirical mean baseline:

$$R^2 = 1 - \frac{\sum_i (y_i - \hat{y}_i)^2}{\sum_i (y_i - \bar{y})^2}. \tag{49}$$

Higher values indicate better predictive performance.

**Perturbation Discrimination Score (PDS).** To evaluate whether models's ability to distinguish different perturbations, the PDS score is defined as the normalized rank of the ground truth from the predicted perturbation with respect to all ground truth perturbations:

$$\text{PDS} = 1 - \frac{1}{M} \sum_{\lambda=1}^M \frac{r_\lambda}{M}, \quad r_\lambda = \sum_{p \neq \lambda} \mathbb{1}[d(\Delta\hat{\mathbf{x}}_\lambda, \Delta\mathbf{x}_p) < d(\Delta\hat{\mathbf{x}}_\lambda, \Delta\mathbf{x}_\lambda)]. \tag{50}$$

Intuitively, the prediction for perturbation $\lambda$ should be closest to its own ground truth, and farther from the ground truths of other perturbations. A PDS value of $1$ indicates perfect discrimination, while a value of approximately $0.5$ corresponds to random performance. We report PDS using three distance functions: L1 distance ($\text{PDS}_{\text{L1}}$), L2 distance ($\text{PDS}_{\text{L2}}$), and cosine distance ($\text{PDS}_{\text{cos}}$).

**Pearson Delta Correlation (PDCorr).** For each perturbation $\lambda$, the PDCorr metric evaluates the Pearson correlation coefficient between the predicted and the ground-truth relative perturbation effects:

$$\text{PDCorr} = \frac{1}{M} \sum_{\lambda=1}^M \text{PearsonR}(\Delta\hat{\mathbf{x}}_\lambda, \Delta\mathbf{x}_\lambda). \tag{51}$$

**Mean Absolute Error (MAE).** The MAE metric evaluates how accurately a model preserves the average expression shift induced by each perturbation. Specifically, it measures the discrepancy between the predicted and ground-truth pseudobulk profiles. Formally, for a given perturbation $\lambda$, the MAE is defined as:

$$\text{MAE} = \frac{1}{M} \sum_{\lambda=1}^M ||\Delta\hat{\mathbf{x}}_\lambda - \Delta\mathbf{x}_\lambda||. \tag{52}$$

**Mean Squared Error (MSE) .** The MSE metric follows the same principle as MAE but penalizes larger deviations more strongly by measuring the squared Euclidean distance between predicted and ground-truth perturbation effects. Formally, the MSE is defined as:

$$\text{MAE} = \frac{1}{M} \sum_{\lambda=1}^{M} ||\Delta \hat{\mathbf{x}}_\lambda - \Delta \mathbf{x}_\lambda||_2^2. \tag{53}$$

### C.2.2. BIOLOGICALLY MEANINGFUL DIFFERENTIAL PATTERNS

The most important criterion for evaluating generated cells is their biological relevance. Cell-Eval applies a standard differential expression (DE) analysis pipeline based on the Wilcoxon rank-sum test (Hollander et al., 2013). To control for false positives arising from multiple hypothesis testing across thousands of genes, $p$-values are adjusted using the Benjamini–Hochberg (BH) procedure (Giraud, 2021), which controls the false discovery rate (FDR). This DE analysis is applied independently to both the ground-truth cells and the model-predicted cells.

**Notations related to DEGs.** We consider the top 2,000 HVG set used for perturbation prediction as $\mathcal{G}$. And we perform differential expression (DE) analysis on both the ground-truth and the predicted cells for each perturbation $\lambda$. For each gene $g \in \mathcal{G}$, it's considered significantly differentially expressed if its BH-adjusted $p$-value $p_{\text{adj}} < 0.05$, where $p_{\text{adj}}$ denotes the $p$-value after correction for multiple testing. We denote by $\mathcal{G}_\lambda^{\text{DE}}$ and $\hat{\mathcal{G}}_\lambda^{\text{DE}}$ the complete sets of significant DE genes detected for the ground-truth and predicted cells, respectively. Furthermore, DE genes can be ranked by the absolute log-fold change $|\log \text{FC}_{\lambda,g}| := |\log_2 \frac{\bar{\mathbf{x}}_{\lambda,g}^{\text{pert}} + \epsilon}{\bar{\mathbf{x}}_{\lambda,g}^{\text{ctrl}} + \epsilon}|$, where $\bar{\mathbf{x}}_{\lambda,g}^{\text{pert}}$ and $\bar{\mathbf{x}}_{\lambda,g}^{\text{ctrl}}$ denote the mean expression of gene $g$ in ground-truth perturbed and control cells, respectively, and $\epsilon$ is a small float constant (e.g., $10^{-8}$) added for numerical stability. The same definition is applied to the predicted perturbed cells by replacing $\bar{\mathbf{x}}_{\lambda,g}^{\text{pert}}$ with the corresponding predicted mean to calculate $|\widehat{\log \text{FC}}_{\lambda,g}|$. We define $\mathcal{G}_\lambda^k$ and $\hat{\mathcal{G}}_\lambda^k$ as the top-$k$ differentially expressed (DE) genes identified from the ground-truth and predicted cells, respectively.

**DE Overlap (DEOver).** For each perturbation $\lambda$, we evaluate the agreement between predicted and ground-truth differentially expressed (DE) genes by measuring the overlap among the top-ranked genes. And the DEO metric is defined as the fraction of overlapping genes:

$$\text{DEOver}_k = \frac{1}{M} \sum_{\lambda=1}^{M} \frac{\left| \mathcal{G}_\lambda^k \cap \hat{\mathcal{G}}_\lambda^k \right|}{k}. \tag{54}$$

Here, $k$ is chosen as the number of significant DE genes in the ground-truth set, i.e., $k = |\mathcal{G}_\lambda^{\text{DE}}|$, ensuring that the comparison is normalized with respect to the true perturbation signal.

**DE Precision (DEPrec).** For each perturbation $\lambda$, we evaluate how many of the top $k$ DEGs from the ground truth appear in the top $k$ DEGs from the predicted cells. That is,

$$\text{DEPrec}_k = \frac{1}{M} \sum_{\lambda=1}^{M} \frac{\left| \mathcal{G}_\lambda^k \cap \hat{\mathcal{G}}_\lambda^k \right|}{|\hat{\mathcal{G}}_\lambda^k|}. \tag{55}$$

Here, $k$ is chosen as the number of significant DE genes in the predicted set, i.e., $k = |\hat{\mathcal{G}}_\lambda^{\text{DE}}|$, ensuring that the comparison is normalized with respect to the predicted perturbation signal.

**Direction Agreement (DirAgr).** For genes that are identified as differentially expressed in both the ground-truth and predicted cells, we further assess whether the model correctly captures the direction of the perturbation effect. Specifically, DirAgr is defined as the fraction of overlapping DE genes for which the predicted and ground-truth log-fold changes have the same sign. Let $\mathcal{G}_\lambda^\cap = \hat{\mathcal{G}}_\lambda^{\text{DE}} \cap \mathcal{G}_\lambda^{\text{DE}}$ denote the set of DE genes shared between prediction and ground truth for perturbation $\lambda$. The DirAgr metric is then defined as

$$\text{DirAgr} = \frac{1}{M} \sum_{\lambda=1}^{M} \frac{\left| \left\{ g \in \mathcal{G}_\lambda^\cap \; : \; \text{sgn}\left( \widehat{\log \text{FC}}_{\lambda,g} \right) = \text{sgn}(\log \text{FC}_{\lambda,g}) \right\} \right|}{|\mathcal{G}_\lambda^\cap|}, \tag{56}$$

where $\text{sgn}(\cdot)$ denotes the sign function.

**Log Fold-change Spearman Correlation (LFCSpear).** This metric evaluates how well a model preserves the relative ordering of gene-level perturbation effects. For each perturbation $\lambda$, we compute the Spearman rank correlation between the

predicted and ground-truth log fold changes over the set of significantly differentially expressed genes identified from the ground truth, and report the average correlation across all perturbations.

$$\text{LFCSpear} = \frac{1}{M} \sum_{\lambda=1}^{M} \text{SpearmanR}\left(\left(\log \text{FC}_{\lambda,g}\right)_{g \in \mathcal{G}_\lambda^{\text{DE}}}, \left(\widehat{\log \text{FC}}_{\lambda,g}\right)_{g \in \mathcal{G}_\lambda^{\text{DE}}}\right). \tag{57}$$

**ROC-AUC (AUROC).** This metric evaluates the model's ability to distinguish truly differentially expressed genes from non-differentially expressed ones. For each perturbation $\lambda$, genes identified as significant DEGs in the ground truth ($\mathcal{G}_\lambda^{\text{DE}}$) are treated as positive samples, while all remaining genes are treated as negatives. The predicted gene-level significance scores are given by the negative log-transformed adjusted p-values, and the area under the ROC curve (AUROC) is computed to quantify how well the model separates significant from non-significant genes across all possible thresholds. The final score is obtained by averaging AUROC across perturbations. Essentially, AUROC measures the probability that a randomly chosen true DEG is assigned a higher confidence score than a randomly chosen non-DEG.

**PR-AUC (AUPRC).** Besides AUROC, the AUPRC metric is similarly defined to evaluate the model's ability to prioritize truly differentially expressed genes among all predicted significant genes. For each perturbation $\lambda$, genes identified as significant DEGs in the ground truth ($\mathcal{G}_\lambda^{\text{DE}}$) are treated as positive samples, while all remaining genes are treated as negatives. The predicted gene-level significance scores are given by the negative log-transformed adjusted p-values. AUPRC is calculated as the area under the precision-recall (PR) curve, which summarizes the trade-off between precision and recall across all possible thresholds. The final score is obtained by averaging PR-AUC across perturbations.

**Effect sizes (ES).** To compare the relative effect sizes of perturbations, this ES metric computes the Spearman rank correlation on the number of differentially expressed genes between predicted and ground-truth for each perturbation.

$$\text{ES} = \text{SpearmanR}\left(\left(|\mathcal{G}_\lambda^{\text{DE}}|\right)_{\lambda=1}^{M}, \left(|\hat{\mathcal{G}}_\lambda^{\text{DE}}|\right)_{\lambda=1}^{M}\right). \tag{58}$$

## C.3. More Perturbation Prediction Results

**Full numerical results for perturbation prediction performance in Fig. 3.** The radar plots in Fig. 3 report the relative comparison across diverse methods and multiple metrics. We provide their corresponding detailed numerical results in Tab. 4 for reference.

**MSE and MAE results.** The radar plots in Fig. 3 report higher-is-better metrics, while the lower-is-better metrics (MSE and MAE) are summarized in Tab. 4. Across all three datasets, PerturbDiff (From Scratch) achieves low reconstruction errors comparable to or better than strong baselines, while STATE and Linear remain competitive on MSE/MAE especially in lower-variance settings.

**Mean baseline variants performance.** Mean baselines predict perturbed cells by assigning all cells the same averaged expression profile computed from observed data. The averaging can be performed at different levels: (1) Mean - per perturbation (*i.e.*, the reported "Mean"): average expression across cells sharing the same perturbation and use this mean to predict cells under that perturbation; (2) Mean - per cell type: average across cells of the same cell type and use this mean for corresponding perturbed cells; (3) Mean - per batch: average across cells from the same experimental batch and use this mean for corresponding perturbed cells; (4) Mean - overall: average across all cells and use this global mean for all predictions. Among these variants, "Mean - per perturbation" achieves the best performance and is therefore reported in Sec. 5.2, while performance for all mean baseline variants are summarized in Tab. 4.

## C.4. More Scatter Plot Comparison Results

Fig. 15 and Fig. 16 report per-perturbation scatter plot comparisons between PerturbDiff (From Scratch) and STATE on PBMC and Tahoe100M, respectively. Each point corresponds to a single perturbation, and the diagonal indicates equal performance. ES and $R^2$ are not included, as they are not defined at the perturbation level.

On PBMC, PerturbDiff (From Scratch) consistently outperforms STATE across perturbations on key DE-metrics including PRAUC, DEOver, and DEPrec, with points tightly concentrated above the diagonal. In addition, PerturbDiff (From Scratch) achieves high win rates for distributional metrics, exceeding 88% on $\text{PDS}_{\text{L1}}$, $\text{PDS}_{\text{L2}}$, and $\text{PDS}_{\text{cos}}$. These results indicate that our method provides more reliable perturbation-level predictions, particularly in capturing distributional shifts rather than

*Table 4.* Detailed numerical results for perturbation prediction across metrics and datasets (the same as reported in Fig. 3).

| Model | $R^2$ | DEOver | DEPrec | ES | DirAgr | LFCSpear | AUPRC | AUROC | PDCorr | MSE | MAE | $PDS_{L1}$ | $PDS_{L2}$ | $PDS_{cos}$ |
|---|---|---|---|---|---|---|---|---|---|---|---|---|---|---|
| **PBMC** | | | | | | | | | | | | | | |
| PerturbDiff (Scratch) | 0.997 | **0.564** | 0.581 | 0.288 | 0.751 | 0.519 | 0.607 | 0.603 | **0.816** | 1.91e-4 | 0.006 | **0.972** | **0.975** | **0.986** |
| PerturbDiff (Finetuned) | 0.991 | 0.533 | **0.588** | **0.387** | 0.701 | 0.431 | **0.669** | **0.647** | 0.705 | 3.15e-4 | 0.008 | 0.941 | 0.959 | 0.972 |
| STATE | 0.998 | 0.512 | 0.547 | -0.363 | **0.789** | 0.602 | 0.547 | 0.513 | 0.796 | **1.41e-4** | 0.005 | 0.954 | 0.943 | 0.985 |
| Mean | 0.959 | **0.564** | 0.544 | 0.011 | 0.740 | 0.478 | 0.545 | 0.506 | 0.642 | 8.49e-4 | 0.013 | 0.730 | 0.777 | 0.787 |
| CPA | 0.919 | 0.488 | 0.515 | -0.154 | 0.539 | 0.369 | 0.515 | 0.500 | 0.181 | 1.10e-2 | 0.052 | 0.576 | 0.613 | 0.597 |
| Linear | 0.997 | 0.549 | 0.581 | 0.194 | 0.666 | 0.366 | 0.591 | 0.556 | 0.646 | 4.44e-4 | 0.009 | 0.658 | 0.670 | 0.903 |
| CellFlow | 0.994 | 0.270 | 0.350 | 0.042 | 0.652 | 0.377 | 0.354 | 0.502 | 0.628 | 3.58e-4 | 0.009 | 0.639 | 0.689 | 0.725 |
| Squidiff | -218.00 | 0.359 | 0.547 | NaN | 0.379 | 0.022 | 0.547 | 0.500 | 0.033 | 4.387 | 2.062 | 0.508 | 0.508 | 0.508 |
| Mean Variant (per Cell Type) | 0.992 | 0.506 | 0.542 | -0.108 | 0.635 | 0.247 | 0.542 | 0.500 | 0.400 | 6.39e-4 | 0.009 | 0.625 | 0.596 | 0.612 |
| Mean Variant (per Batch) | **0.999** | 0.554 | 0.542 | 0.019 | 0.727 | 0.489 | 0.543 | 0.501 | 0.517 | 5.59e-4 | 0.007 | 0.612 | 0.587 | 0.655 |
| Mean Variant (Overall) | 0.973 | 0.557 | 0.542 | NaN | 0.656 | 0.250 | 0.543 | 0.500 | 0.408 | 1.01e-3 | 0.013 | 0.508 | 0.508 | 0.508 |
| **Tahoe100M** | | | | | | | | | | | | | | |
| PerturbDiff (Scratch) | 0.963 | 0.522 | 0.572 | **0.531** | 0.734 | 0.445 | 0.584 | 0.621 | 0.686 | 6.30e-4 | 0.012 | **0.970** | **0.978** | **0.975** |
| PerturbDiff (Finetuned) | 0.893 | **0.580** | **0.598** | 0.478 | 0.641 | **0.691** | **0.618** | **0.658** | 0.648 | 1.04e-3 | 0.017 | 0.784 | 0.883 | 0.903 |
| STATE | 0.807 | 0.499 | 0.522 | 0.301 | 0.665 | 0.644 | 0.520 | 0.506 | 0.535 | 2.11e-3 | 0.021 | 0.789 | 0.846 | 0.854 |
| Mean | 0.703 | 0.430 | 0.504 | 0.350 | 0.595 | 0.280 | 0.506 | 0.502 | 0.205 | 3.09e-3 | 0.026 | 0.515 | 0.508 | 0.508 |
| CPA | 0.946 | 0.502 | 0.504 | 0.190 | 0.710 | 0.466 | 0.505 | 0.500 | 0.425 | 8.89e-4 | 0.011 | 0.654 | 0.593 | 0.585 |
| Linear | **0.993** | 0.505 | 0.576 | 0.382 | **0.760** | 0.514 | 0.605 | 0.632 | **0.723** | 4.15e-4 | 0.009 | 0.887 | 0.881 | 0.912 |
| CellFlow | 0.682 | 0.183 | 0.313 | 0.346 | 0.638 | 0.298 | 0.320 | 0.500 | 0.269 | 3.10e-3 | 0.027 | 0.608 | 0.564 | 0.550 |
| Squidiff | -616.98 | 0.420 | 0.581 | NaN | 0.501 | 0.276 | 0.581 | 0.500 | 0.011 | 5.627 | 2.244 | 0.505 | 0.506 | 0.505 |
| Mean Variant (per Cell Type) | 0.976 | 0.504 | 0.508 | 0.023 | 0.688 | 0.578 | 0.508 | 0.508 | 0.461 | 6.96e-4 | 0.010 | 0.682 | 0.694 | 0.760 |
| Mean Variant (per Batch) | 0.705 | 0.427 | 0.504 | 0.475 | 0.587 | 0.270 | 0.505 | 0.500 | 0.185 | 3.11e-3 | 0.026 | 0.492 | 0.496 | 0.498 |
| Mean Variant (Overall) | 0.707 | 0.429 | 0.504 | 0.469 | 0.591 | 0.275 | 0.505 | 0.500 | 0.193 | 3.06e-3 | 0.026 | 0.501 | 0.501 | 0.501 |
| **Replogle** | | | | | | | | | | | | | | |
| PerturbDiff (Scratch) | 0.984 | 0.190 | 0.174 | 0.623 | 0.702 | 0.342 | 0.210 | 0.625 | 0.340 | 1.47e-2 | 0.081 | 0.690 | 0.700 | 0.575 |
| PerturbDiff (Finetuned) | 0.988 | **0.214** | 0.192 | 0.762 | 0.723 | 0.388 | **0.257** | **0.652** | 0.376 | 1.46e-2 | 0.079 | 0.758 | 0.762 | 0.639 |
| STATE | 0.998 | 0.196 | **0.193** | **0.818** | **0.778** | **0.506** | 0.239 | 0.633 | **0.437** | 6.40e-3 | 0.055 | **0.788** | **0.802** | **0.676** |
| Mean | 0.742 | 0.127 | 0.094 | 0.417 | 0.532 | 0.077 | 0.092 | 0.467 | 0.048 | 9.90e-2 | 0.206 | 0.582 | 0.599 | 0.608 |
| CPA | **1.000** | 0.173 | 0.087 | 0.637 | 0.746 | 0.499 | 0.081 | 0.401 | 0.418 | 7.50e-2 | **0.054** | 0.582 | 0.581 | 0.521 |
| Linear | 0.991 | 0.068 | 0.074 | 0.353 | 0.535 | 0.056 | 0.085 | 0.435 | 0.058 | 1.30e-2 | 0.074 | 0.519 | 0.517 | 0.667 |
| CellFlow | 0.757 | 0.110 | 0.093 | 0.442 | 0.472 | -0.033 | 0.086 | 0.434 | -0.003 | 9.49e-2 | 0.205 | 0.507 | 0.507 | 0.495 |
| Squidiff | -10.19 | 0.039 | 0.091 | 0.419 | 0.432 | -0.000 | 0.079 | 0.366 | 0.089 | 4.641 | 2.077 | 0.500 | 0.501 | 0.501 |
| Mean Variant (per Cell Type) | **1.000** | 0.178 | 0.087 | 0.417 | 0.743 | 0.492 | 0.078 | 0.404 | 0.412 | 8.45e-3 | 0.057 | 0.502 | 0.505 | 0.502 |
| Mean Variant (per Batch) | 0.798 | 0.110 | 0.092 | 0.421 | 0.475 | -0.024 | 0.088 | 0.450 | 0.002 | 6.99e-2 | 0.174 | 0.505 | 0.505 | 0.506 |
| Mean Variant (Overall) | 0.794 | 0.111 | 0.093 | 0.416 | 0.474 | -0.027 | 0.089 | 0.453 | -0.001 | 7.11e-2 | 0.175 | 0.502 | 0.503 | 0.501 |

isolated cell-level effects.

On the larger and more diverse Tahoe100M dataset, PerturbDiff (From Scratch) similarly demonstrates strong advantages, achieving win rates above 87% on PRAUC, DEPrec, MAE, MSE, and all PDS variants. Moreover, PerturbDiff (From Scratch) attains win rates above 50% on all reported metrics except LFCSpear, indicating broadly consistent improvements across perturbations despite increased dataset heterogeneity.

Together, these scatter comparisons show that the gains of PerturbDiff (From Scratch) are not driven by a small subset of perturbations, but instead reflect systematic and robust improvements across diverse perturbation settings, particularly on metrics that emphasize distribution-level response fidelity and perturbation discrimination.

## C.5. More Zero-shot Results

We further analyze zero-shot behavior under different amounts of marginal pretraining, where inference is performed without access to perturbation labels or control cells, and the model relies solely on the pretrained marginal cell manifold.

As shown in Fig. 17 and Fig. 18, metrics such as MAE and MSE remain relatively stable across pretraining steps on both PBMC and Replogle, indicating limited sensitivity of pointwise reconstruction errors to marginal pretraining. In contrast, many distribution- and perturbation-aware metrics (*e.g.*, DEOver, DEPrec, LFCSpear, DirAgr, PDCorr, and ES) exhibit a consistent U-shaped trend: zero-shot performance initially degrades as pretraining proceeds, followed by a gradual recovery with increased training steps. This pattern suggests that early stages of marginal pretraining may temporarily distort task-relevant directions, while longer pretraining progressively organizes biologically meaningful structure that can be reused for perturbation inference.

Notably, on PBMC, the randomly initialized model attains seemingly non-trivial scores on several metrics (*e.g.*, AUPRC, AUROC, DEPrec, and ES), despite the absence of any training. This phenomenon is not observed on Replogle, and likely

reflects dataset-specific structure or metric sensitivity rather than genuine perturbation understanding. Indeed, on more stringent metrics—including DEOver, $R^2$, PDCorr, DirAgr, MAE, and MSE—the randomly initialized model substantially underperforms pretrained counterparts. These observations highlight the necessity of multi-metric evaluation when assessing zero-shot perturbation performance, as isolated metrics may overestimate capability in the absence of meaningful biological modeling.

### C.6. More Limited Data Results

Fig. 19 evaluates few-shot adaptation performance across training steps on downsampled PBMC, comparing training from scratch and finetuning a marginally pretrained model at sample ratios of 1% and 5%.

Across all metrics, finetuning consistently leads to faster convergence and substantially improved training stability. In contrast, models trained from scratch exhibit pronounced sensitivity to training steps, with large fluctuations and frequent performance degradation in early and mid training stages, particularly under the 1% sample regime. Finetuned models reach stable performance within fewer training steps and maintain more robust throughout optimization.

The advantages of finetuning are especially pronounced on many metrics, including DEOver, DEPrec, DirAgr, LFCSpear, and all PDS variants. On these metrics, finetuned models consistently outperform training from scratch across training steps, with larger gaps observed under more extreme low-data settings (ratio=1%). This suggests that marginal pretraining provides a structured initialization that preserves perturbation-relevant geometry, enabling effective reuse of distribution-level information during adaptation.

Overall, these results demonstrate that marginal pretraining substantially improves data efficiency, optimization stability, and robustness under low-data regimes.

### C.7. More Scaling Results

Fig. 20 reports the rest metrics for the scaling behavior of PerturbDiff (From Scratch) by varying both model size and training compute. Overall, scaling exhibits non-monotonic behavior along both model size and compute dimensions. Increasing training compute does not uniformly improve performance. These results indicate that effective perturbation modeling benefits from balanced model capacity and compute, rather than aggressive scaling.

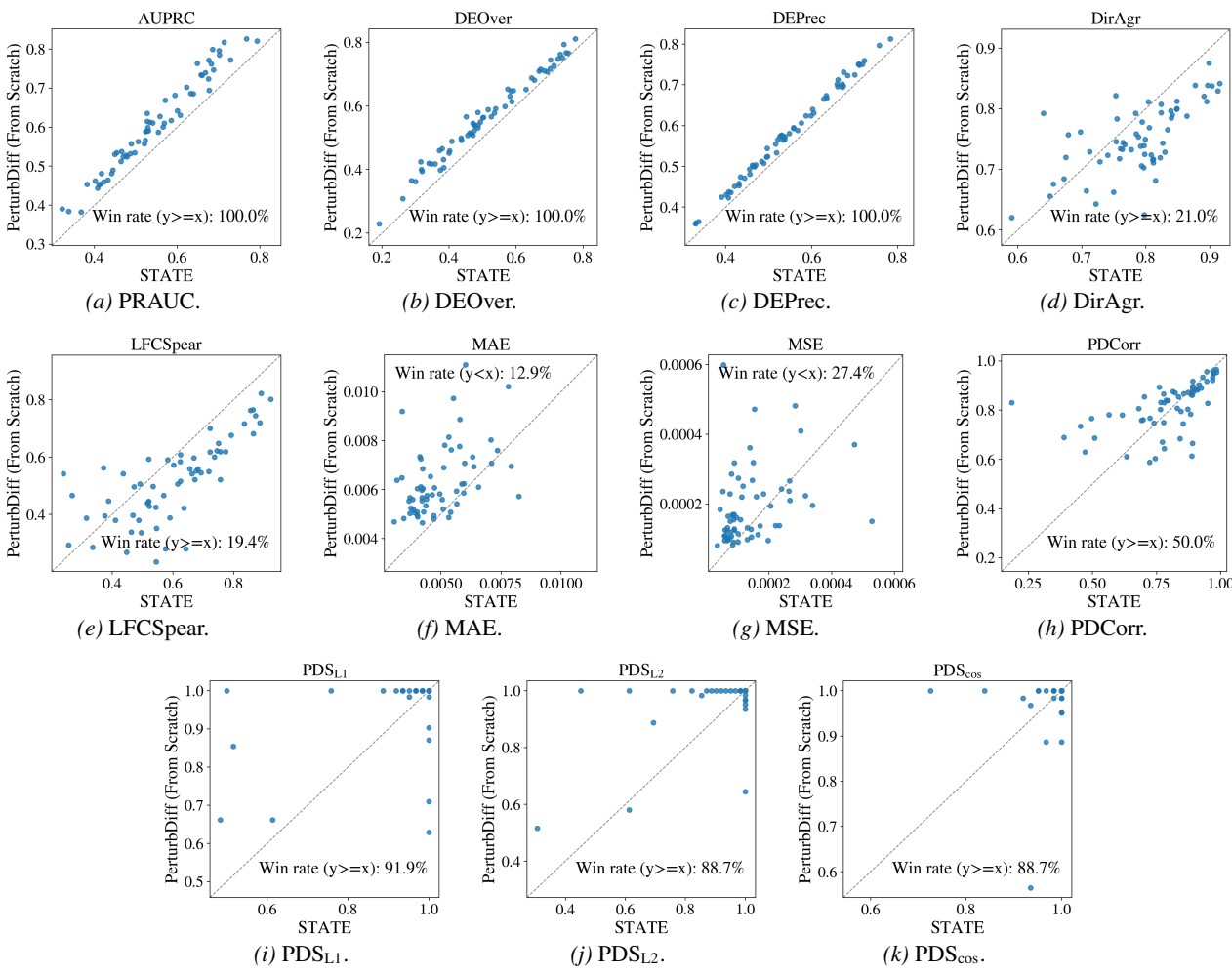

*Figure 15.* Per-metric scatter plot comparison between PerturbDiff (From Scratch) and STATE on PBMC dataset. Each panel corresponds to a different evaluation metric, plotting STATE (x-axis) against PerturbDiff (From Scratch) (y-axis) across held-out perturbations. The dashed diagonal indicates equal performance; points above the diagonal indicate higher performance for PerturbDiff (From Scratch) (reversed for MAE and MSE). Reported win rates denote the fraction of perturbations where PerturbDiff (From Scratch) achieves equal or better performance.

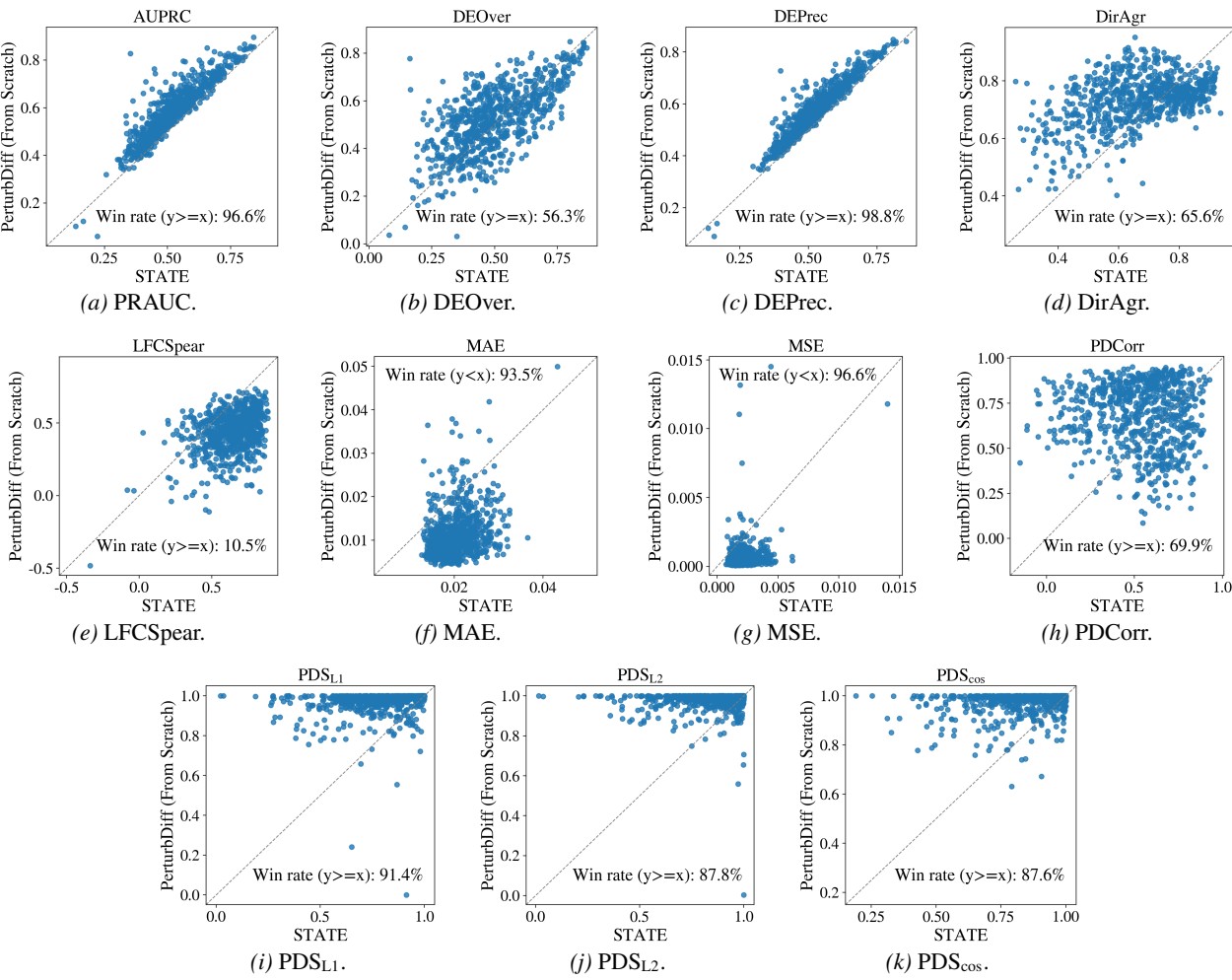

*Figure 16.* Per-metric scatter plot comparison between PerturbDiff (From Scratch) and STATE on Tahoe100M dataset. Each panel corresponds to a different evaluation metric, plotting STATE (x-axis) against PerturbDiff (From Scratch) (y-axis) across held-out perturbations. The dashed diagonal indicates equal performance; points above the diagonal indicate higher performance for PerturbDiff (From Scratch) (reversed for MAE and MSE). Reported win rates denote the fraction of perturbations where PerturbDiff (From Scratch) achieves equal or better performance.

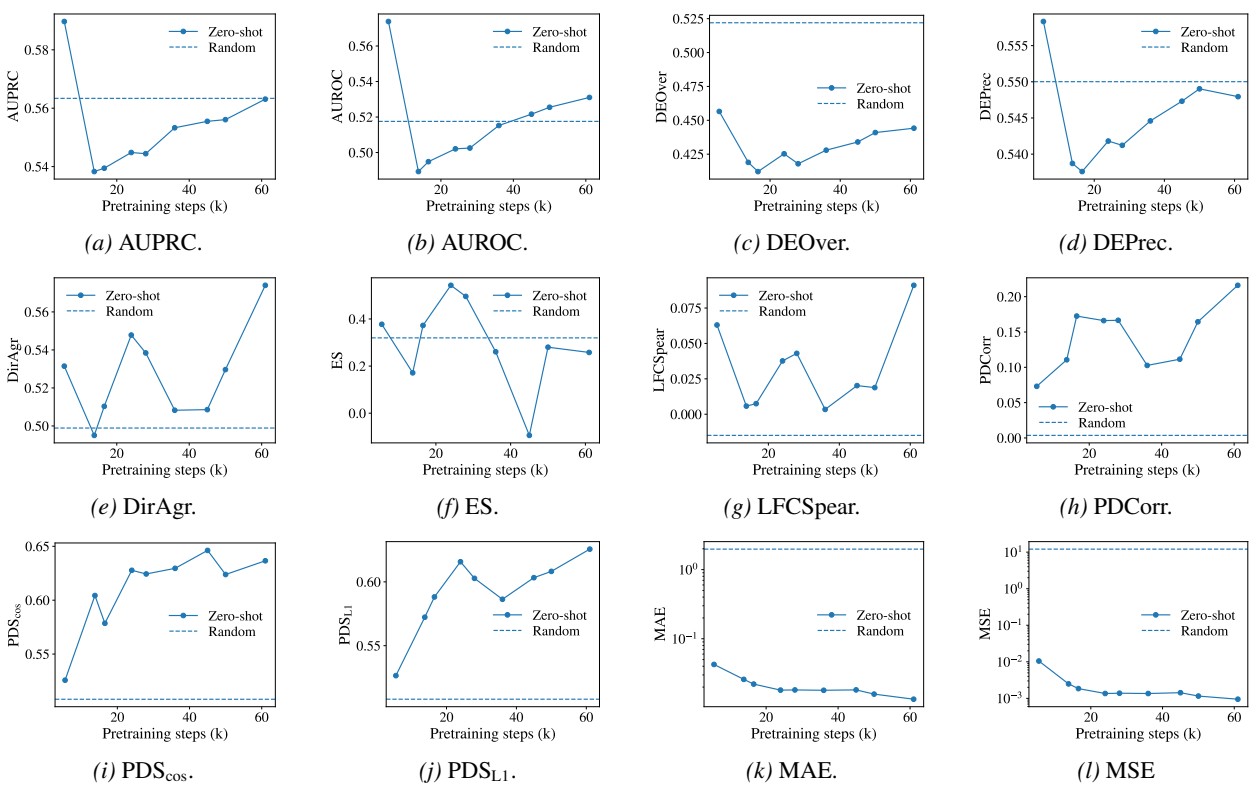

*Figure 17.* Zero-shot performance on PBMC across pretraining steps, compared to a random baseline.

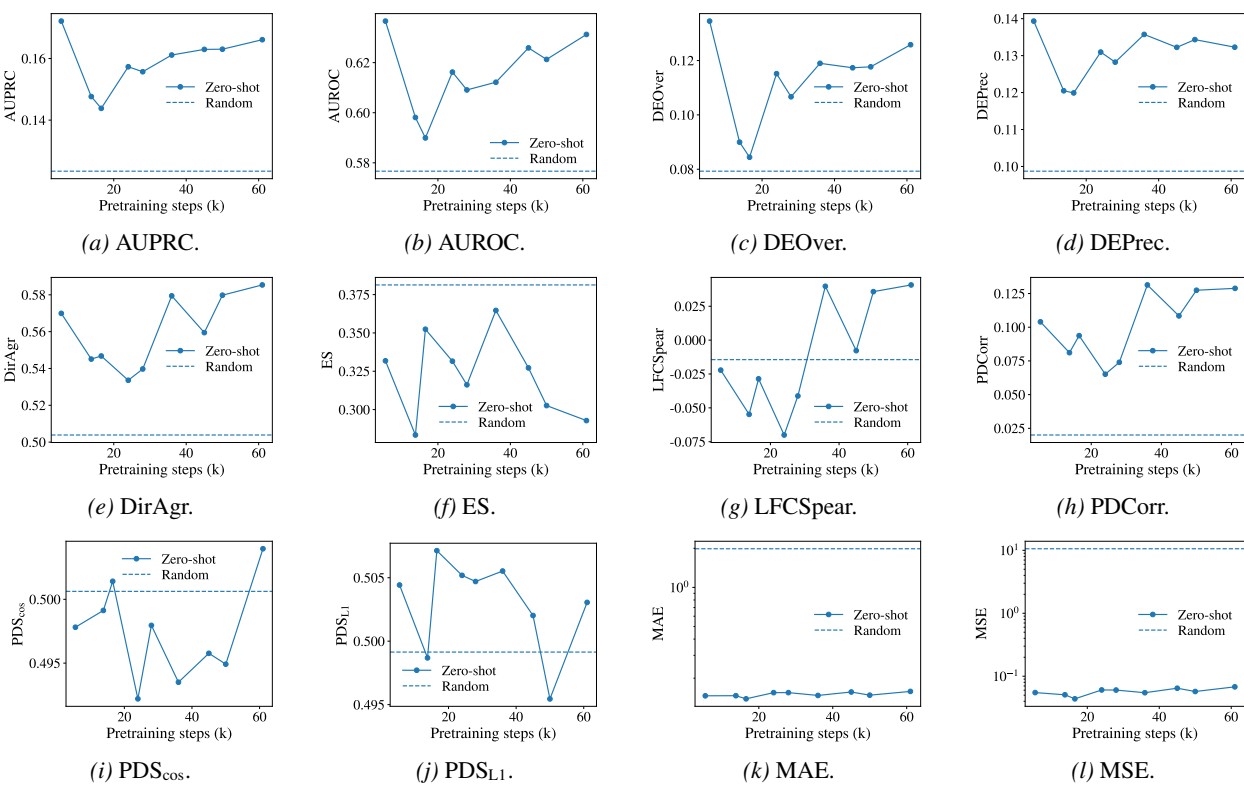

*Figure 18.* Zero-shot performance on Replogle across pretraining steps, compared to a random baseline.

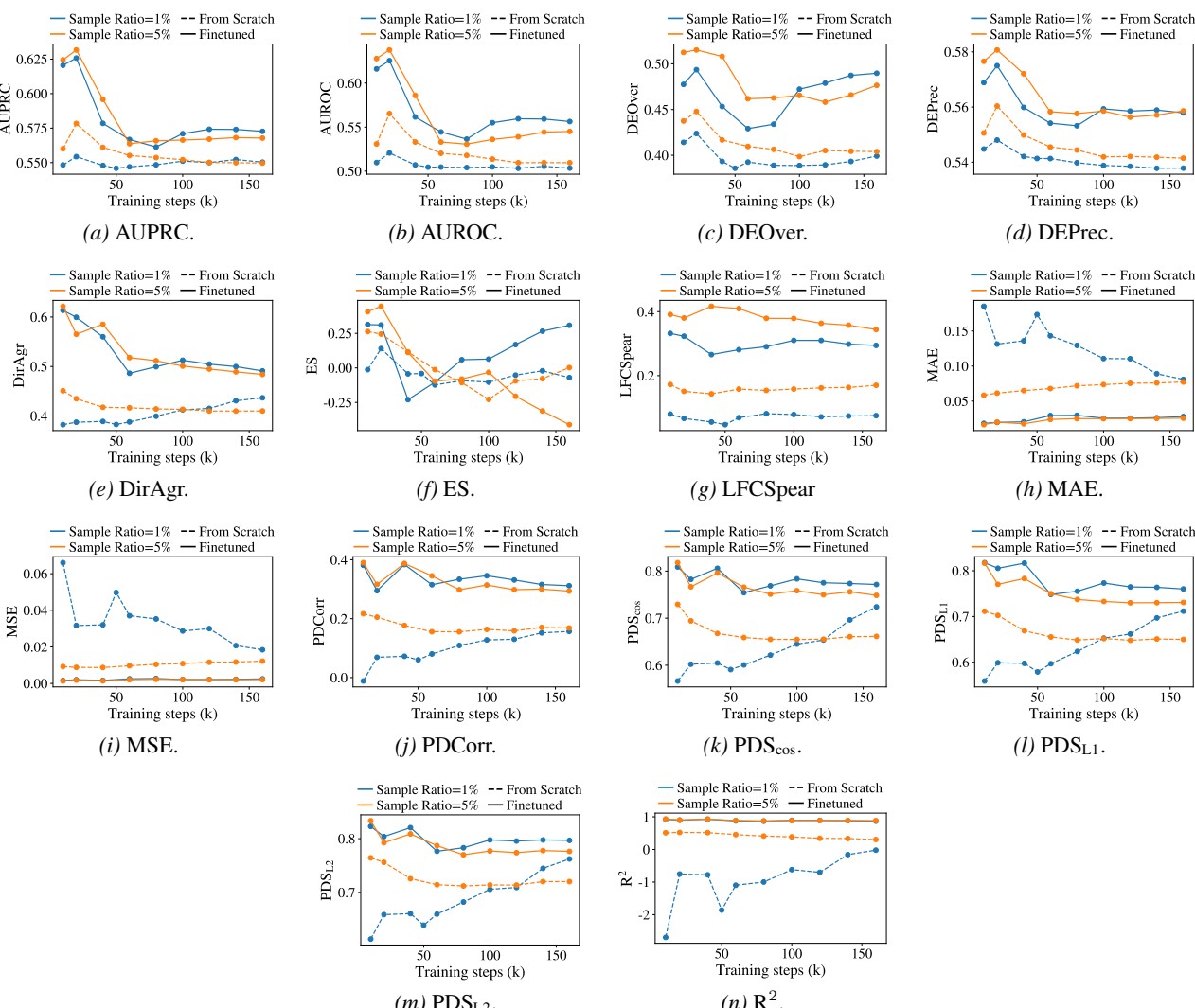

*Figure 19.* Performance on downsampled PBMC (sample ratio 1% and 5%) across training steps for diverse metrics.

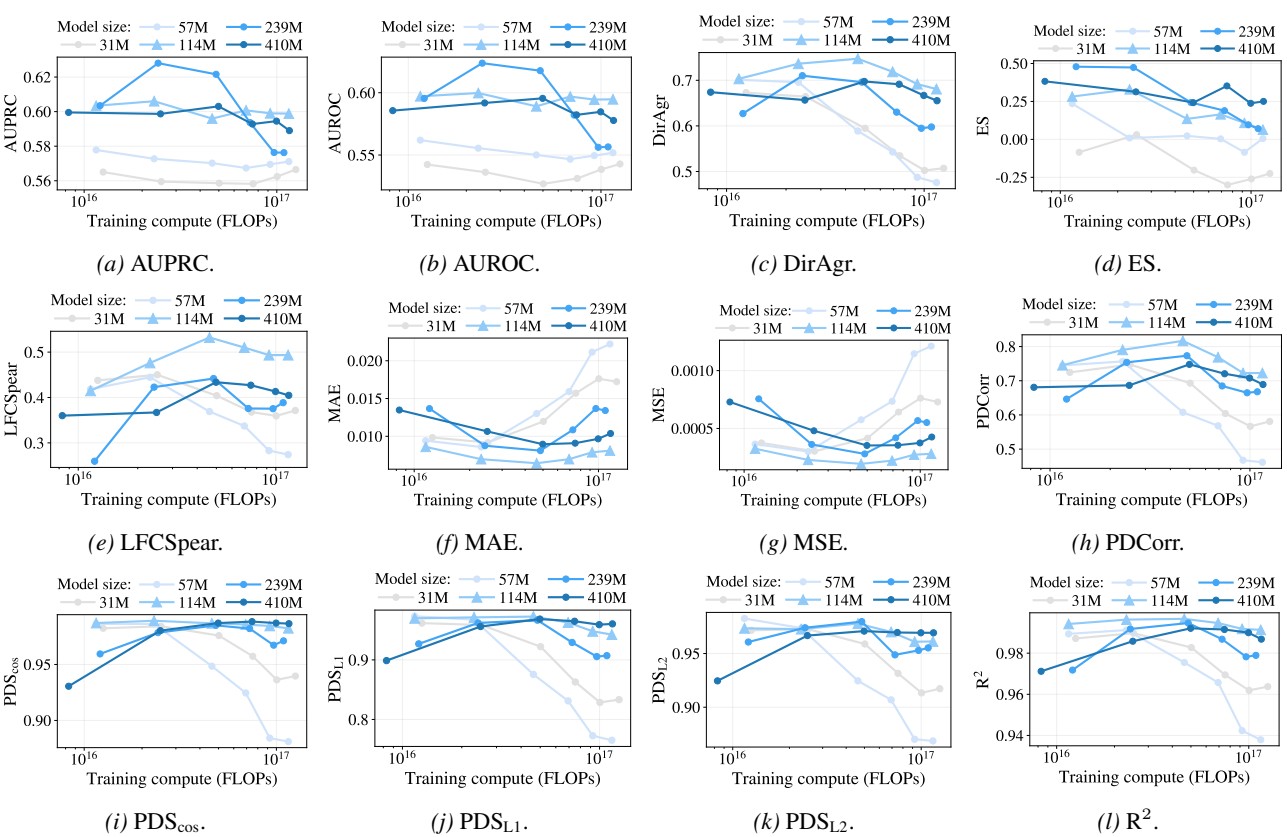

*Figure 20.* Compute and model-size scaling of PerturbDiff on PBMC for diverse metrics.

---

**Algorithm 1** One training step of PerturbDiff.

---

**Require:** Clean perturbed batch $B_0$, matched control batch $B_{\text{ctrl}}$, cellular context $c$, perturbation label $\tau$, denoiser $f_\theta$, noise schedule $\{\alpha_t\}_{t=1}^T$, and MSE weight $\lambda_{\text{MSE}}$.
1: Sample $t \sim \text{Unif}\{1, \ldots, T\}$ and $\varepsilon \sim \mathcal{N}(0, I)$.
2: Form the noised batch $B_t = \sqrt{\alpha_t} B_0 + \sqrt{1 - \alpha_t} \varepsilon$.
3: Apply classifier-free metadata dropout to $(c, \tau)$ with probability $p_{\text{drop}}$; always keep $B_{\text{ctrl}}$.
4: With probability $p_{\text{sc}}$, compute a stop-gradient self-conditioning batch $\bar{B}_\theta = \text{sg}\{f_\theta(B_t, t; B_{\text{ctrl}}, c, \tau)\}$; otherwise set $\bar{B}_\theta = \varnothing$.
5: Predict the clean perturbed batch $B_\theta = f_\theta(B_t, t; B_{\text{ctrl}}, c, \tau, \bar{B}_\theta)$.
6: Compute $\mathcal{L}_{\text{MMD}} = \text{ED}(\tilde{P}_{B_0}, \tilde{P}_{B_\theta})$ and $\mathcal{L}_{\text{MSE}} = \|B_0 - B_\theta\|_2^2$.
7: Update $\theta$ using $\mathcal{L}_{\text{total}} = \mathcal{L}_{\text{MMD}} + \lambda_{\text{MSE}} \mathcal{L}_{\text{MSE}}$ and update EMA weights.

---

**Algorithm 2** DDIM sampling of a perturbed population.

---

**Require:** Matched control batch $B_{\text{ctrl}}$, cellular context $c$, perturbation label $\tau$, decreasing DDIM timesteps $t_1 > \cdots > t_K$, EMA denoiser $f_\theta$, and guidance weight $w$.
1: Initialize $B_{t_1} \sim \mathcal{N}(0, I)$ and set self-conditioning state $\bar{B}_\theta = \varnothing$.
2: **for** $k = 1, \ldots, K - 1$ **do**
3:     Predict $B_{\theta,c} = f_\theta(B_{t_k}, t_k; B_{\text{ctrl}}, c, \tau, \bar{B}_\theta)$ with metadata conditioning.
4:     Predict $B_{\theta,u} = f_\theta(B_{t_k}, t_k; B_{\text{ctrl}}, \varnothing, \varnothing, \bar{B}_\theta)$ with metadata dropped but control retained.
5:     Convert $B_{\theta,c}$ and $B_{\theta,u}$ into noise predictions $\hat{\varepsilon}_c$ and $\hat{\varepsilon}_u$ using Eqn. (29).
6:     Apply classifier-free guidance $\hat{\varepsilon}_{\text{cfg}} = (1 + w)\hat{\varepsilon}_c - w\hat{\varepsilon}_u$.
7:     Update $B_{t_{k+1}} = \sqrt{\alpha_{t_{k+1}}} B_{\theta,c} + \sqrt{1 - \alpha_{t_{k+1}}} \hat{\varepsilon}_{\text{cfg}}$.
8:     Set $\bar{B}_\theta = B_{\theta,c}$ for self-conditioning.
9: **end for**
10: **return** the final denoised batch $B_{t_K}$ as the generated perturbed population.

---

# D. Additional Experiment Results

## D.1. Implementation pseudocode.

Algorithms 1 and 2 summarize the cell-space realization used in our implementation. The clean perturbed batch $B_0$ and matched control batch $B_{\text{ctrl}}$ are Monte Carlo representatives of their corresponding empirical cell distributions. Thus, although the diffusion process is derived over RKHS kernel mean embeddings in App. B.1, the model can be trained and sampled directly with empirical batches: $B_t$ realizes the noised perturbed distribution, $B_\theta$ realizes the predicted denoised distribution, and the MMD term is the empirical RKHS distance between the induced distributions.

## D.2. Additional Generalization, Robustness, and Ablation Results

This section complements the main evaluation in Sec. 5.2 and the extended results above with additional analyses of held-out cell-line generalization, population-level discrepancies, diffusion-specific ablations, kernel and batch-size sensitivity, sampling cost, and seed-wise variability.

### D.2.1. EVALUATION PROTOCOL FOR HELD-OUT CONTEXTS

For each test condition $y = (c, \tau)$, where $c$ denotes the observed cellular context including cell type and experimental batch and $\tau$ denotes the perturbation type, we generate a batch of perturbed cells by DDIM sampling conditioned on the matched control population and covariates. We package generated cells into a predicted AnnData object with matched perturbation labels, control labels, and gene order, and compare it to the real AnnData object using the same Cell-Eval protocol as in Sec. C.2. For fair comparison across methods, the control-cell set used by the predicted and real AnnData objects is kept identical and fixed across all compared baselines. This evaluation produces both cell-level metrics based on averaged expression profiles and gene-level metrics based on differential-expression recovery.

We emphasize that the main unseen-context benchmark is different from the strict unseen-perturbation-identity setting. In the main benchmark, held-out contexts retain partial perturbation coverage, whereas a strict unseen-perturbation-identity

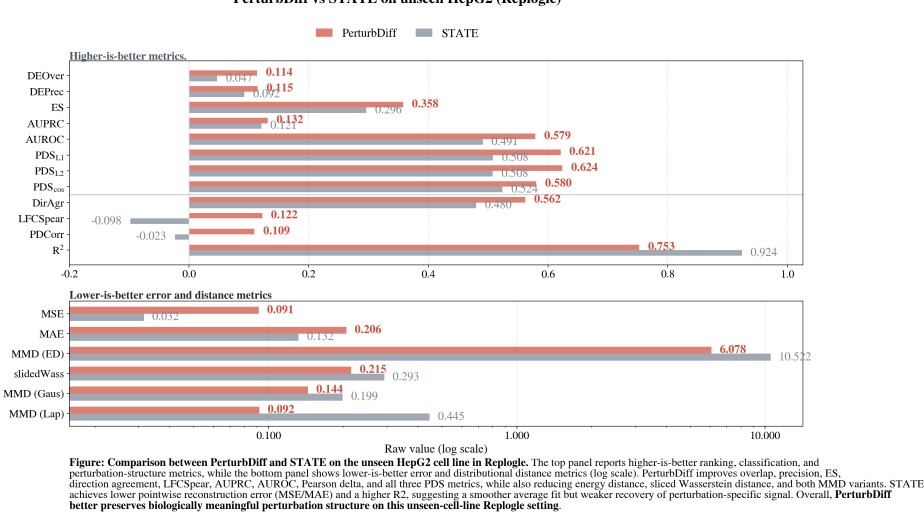

*Figure 21.* Unseen-cell-line generalization on Replogle. HepG2 is held out during training and all perturbations from HepG2 are used for testing. PerturbDiff outperforms STATE on most reported metrics under the original unseen-context hyperparameters of each method.

*Table 5.* Additional population-level discrepancy metrics on Tahoe100M in the from-scratch downstream setting. Lower is better for all metrics.

| Method | ED ↓ | SlicedWass ↓ | MMD-Gaussian ↓ | MMD-Laplacian ↓ |
|---|---|---|---|---|
| STATE | 0.6177 | 0.0693 | 0.0269 | 0.0250 |
| PerturbDiff (Scratch) | **0.0681** | **0.0216** | **0.0033** | **0.0025** |

benchmark would remove an entire perturbation identity from training. The latter setting is possible in principle through semantic perturbation embeddings, but is not part of the present benchmark.

### D.2.2. UNSEEN-CELL-LINE GENERALIZATION ON REPLOGLE

To further evaluate generalization beyond the main unseen-context split, we consider the unseen-cell-line setting used in prior STATE evaluations, where one cell line is held out and all perturbations from that cell line are used for testing. We hold out HepG2 in Replogle and compare PerturbDiff with STATE using each method's original unseen-context hyperparameters, without additional tuning. As shown in Fig. 21, PerturbDiff outperforms STATE on most metrics in this setting, indicating that the distribution-level diffusion formulation transfers to cellular contexts not observed during training.

### D.2.3. POPULATION-LEVEL DISCREPANCY METRICS

The main text emphasizes DE-related metrics because they directly measure biologically meaningful perturbation responses. We additionally evaluate population-level discrepancies between predicted and observed cell populations. These metrics provide a complementary view of whether the generated population matches the ground-truth distribution beyond averaged expression profiles. On Tahoe100M, PerturbDiff (From Scratch) substantially improves over STATE under energy distance, sliced Wasserstein distance, Gaussian-kernel MMD, and Laplacian-kernel MMD (Tab. 5).

### D.2.4. ABLATION: DIRECT CONDITIONAL GENERATOR WITHOUT DIFFUSION

To isolate the contribution of diffusion dynamics from the population-level MMD objective, we ablate diffusion and time conditioning while keeping the same backbone and MMD loss. This variant becomes a direct conditional generator trained to map the conditioning information to a perturbed batch in one step. As shown in Fig. 22, removing diffusion substantially harms biological recovery: the ES metric becomes negative, indicating that the predicted DEG ranking is inversely correlated with the ground-truth DEG ranking. This supports that the diffusion process contributes beyond simply adding an MMD

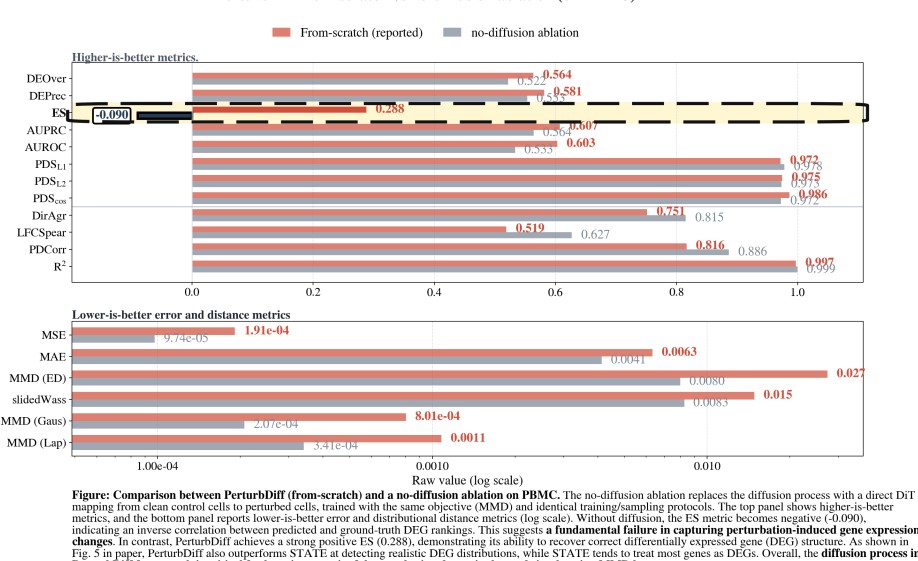

Figure: Comparison between PerturbDiff (from-scratch) and a no-diffusion ablation on PBMC. The no-diffusion ablation replaces the diffusion process with a direct DiT mapping from clean control cells to perturbed cells, trained with the same objective (MMD) and identical training/sampling protocols. The top panel shows higher-is-better metrics. The bottom panel reports lower-is-better error and distributional distance metrics (log scale). Without diffusion, the ES metric becomes negative (-0.090), indicating an inverse correlation between predicted and ground-truth DEG rankings. This suggests a **fundamental failure in capturing perturbation-induced gene expression changes**. In contrast, PerturbDiff achieves a strong positive ES (0.288), demonstrating its ability to recover correct differentially expressed gene (DEG) structure. As shown in Fig. 5 in paper, PerturbDiff also outperforms STATE at detecting realistic DEG distributions, while STATE tends to treat most genes as DEGs. Overall, the **diffusion process in PerturbDiff framework is critical for learning meaningful perturbation dynamics beyond simply using MMD loss**.

*Figure 22.* No-diffusion ablation. The ablated model keeps the same backbone and MMD loss but removes diffusion and time conditioning, reducing the model to a direct conditional generator. The performance drop, especially the negative ES, indicates failure to recover biologically meaningful differential-expression rankings.

population-matching term to a conditional generator.

### D.2.5. KERNEL SENSITIVITY

Our default implementation uses the energy-distance kernel because it does not require a bandwidth hyperparameter and matches the assumptions used in the tractable cell-space approximation in App. B.1.7. We additionally test Gaussian, Laplacian, and simple multi-kernel variants. These kernels are compatible with the RKHS derivation, but introduce bandwidth choices that can affect empirical performance. Fig. 23 summarizes the sensitivity analysis. Overall, the results motivate the default energy-distance choice as a simple, bandwidth-free option while suggesting that learned or adaptively selected kernels may be a useful future extension.

### D.2.6. RECOVERING LATENT BATCH VARIABILITY WITHOUT BATCH LABELS

Directly evaluating a "distribution over distributions" is difficult because the benchmarks do not provide ground-truth labels for all latent factors, and the RKHS objects are infinite-dimensional. As a proxy, we remove experimental batch labels from the conditioning metadata and test whether generated samples still preserve batch-associated variability. Fig. 24 compares generated populations in PCA space. STATE collapses into a compact region with near-zero variability, whereas PerturbDiff preserves a batch-dependent spread closer to the real data. This suggests that the functional diffusion formulation can retain latent distributional variability even when an observed batch label is not explicitly provided.

### D.2.7. SAMPLING STEPS AND COMPUTATIONAL COST

Unless otherwise specified, we use DDIM sampling with $K = 100$ steps for evaluation, which is substantially faster than full $K = 1000$-step sampling while retaining similar performance. Fig. 25 summarizes the runtime, memory, and sampling-step trade-off. These results contextualize the computational cost of using diffusion for population prediction and motivate the default fast-sampling configuration.

### D.2.8. BATCH-SIZE EFFECTS

Because each empirical batch induces a Monte Carlo estimate of a cell distribution and its KME, batch size affects both the quality of the distribution estimate and the cost of MMD computation. We therefore vary the batch size on Replogle, where

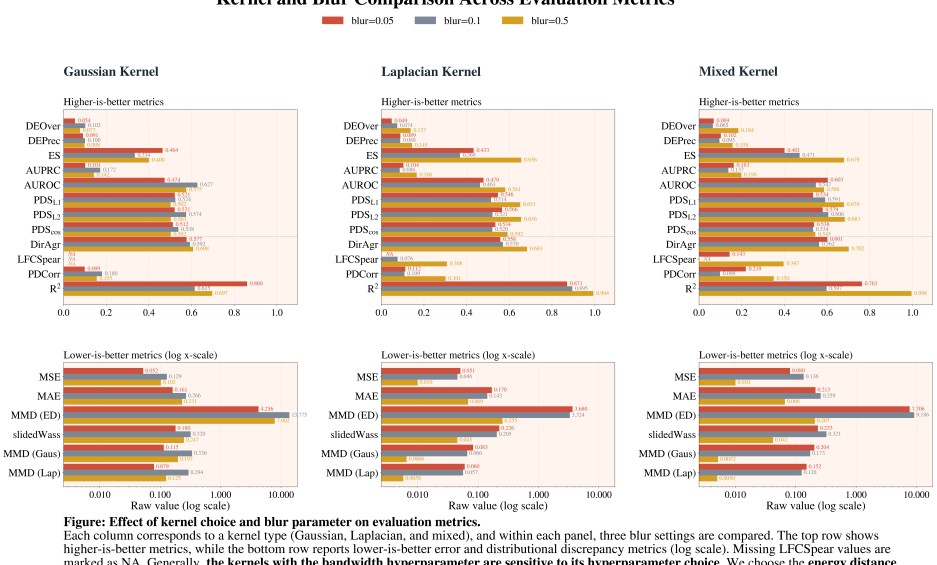

**Figure: Effect of kernel choice and blur parameter on evaluation metrics.**
Each column corresponds to a kernel type (Gaussian, Laplacian, and mixed), and within each panel, three blur settings are compared. The top row shows higher-is-better metrics, while the bottom row reports lower-is-better error and distributional discrepancy metrics (log scale). Missing LFCSpear values are marked as NA. Generally, **the kernels with the bandwidth hyperparameter are sensitive to its hyperparameter choice**. We choose the **energy distance** MMD kernel as it does **not** have any hyperparameter therefore is simple.

*Figure 23.* Kernel sensitivity analysis for MMD-based population matching. Gaussian, Laplacian, and multi-kernel choices are theoretically compatible with the RKHS formulation, but bandwidth selection affects empirical behavior.

the default setting uses batch size 32 for both training and sampling. Fig. 26 shows that batch size influences metrics, but no single value dominates across all metrics. This is consistent with the trade-off between estimator stability, stochastic regularization, and compute.

### D.2.9. SEED-WISE VARIABILITY

Diffusion training on large-scale perturbation datasets is computationally expensive, making repeated-seed evaluation costly. We nevertheless report seed-wise variability in the feasible setting shown in Fig. 27. The results indicate that the qualitative performance trends are stable across random seeds, supporting that the main conclusions are not artifacts of one training run.

### D.3. Explicit RKHS Branch via Nyström Kernel Mean Embeddings

The base implementation realizes the RKHS diffusion objective in cell space for scalability. To test whether making the RKHS representation more explicit can provide additional empirical benefit, we implement a finite-rank Nyström approximation to the KME. This variant is intended as a diagnostic experiment rather than the default model, because it introduces additional PCA, anchor selection, and loss-weight search costs that are prohibitive for very large datasets such as Tahoe100M and PBMC.

**Anchor selection.** We first project training and validation cells into PCA space, group cells by metadata such as cell line or batch, allocate the anchor budget proportionally to group size, run $k$-means within each group, and choose the nearest real cell to each cluster center. This yields a set of real anchor cells $A = \{a_r\}_{r=1}^R$ that covers the data manifold.

**Nyström KME.** For a cell batch $B = \{x_i\}_{i=1}^m$, we define a finite-dimensional KME coordinate vector

$$\hat{\phi}_A(B) = \frac{1}{m} \sum_{i=1}^m \left[ k(x_i, a_1), \ldots, k(x_i, a_R) \right]^\top \in \mathbb{R}^R, \tag{59}$$

where $k$ is the same energy-distance kernel family used in the MMD objective. The vector $\hat{\phi}_A(B)$ is a tractable proxy for the infinite-dimensional KME of the empirical distribution induced by $B$.

**Model and loss.** We extend the MM-DiT with an RKHS branch. The noisy perturbed KME $\hat{\phi}_A(B_t)$ and the control KME $\hat{\phi}_A(B_{\mathrm{ctrl}})$ are encoded as batch-global tokens and prepended to the perturbed and control cell-token streams, respectively.

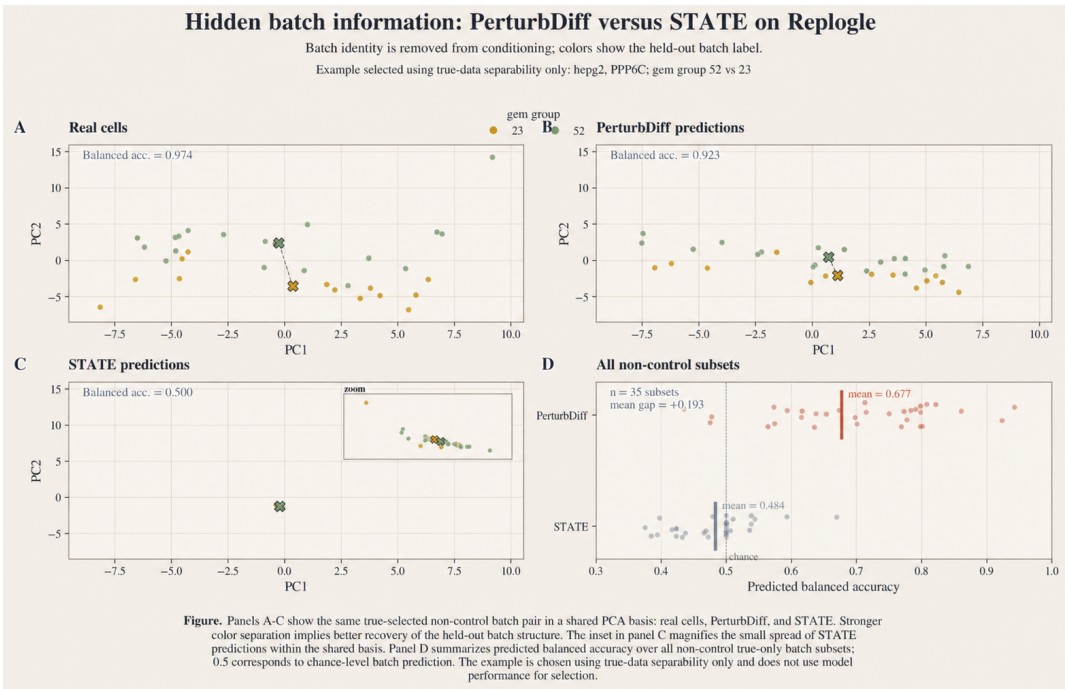

*Figure 24.* Proxy analysis for latent distributional variability. Batch labels are removed from conditioning. PerturbDiff preserves batch-dependent spread in PCA space, while STATE collapses to a much smaller region.

*Table 6.* Explicit RKHS branch on Replogle. Higher is better for all metrics except MSE, MAE, ED, SlicedWass, MMD-Gaussian, and MMD-Laplacian, where lower is better. The Nyström KME branch improves most metrics, suggesting that the RKHS formulation is not merely interpretive, although this variant is more computationally expensive than the default cell-space realization.

| Method | DEOver | DEPrec | ES | DirAgr | LFCSpear | AUPRC | AUROC | PDCorr | MSE | MAE | PDS$_{L1}$ | PDS$_{L2}$ | PDS$_{cos}$ | R$^2$ | ED | SlicedWass | MMD-Gaus | MMD-Lap |
|---|---|---|---|---|---|---|---|---|---|---|---|---|---|---|---|---|---|---|
| **PerturbDiff + RKHS** | **0.202** | **0.187** | **0.722** | **0.722** | **0.380** | **0.244** | **0.645** | **0.352** | **0.013** | **0.074** | **0.764** | **0.771** | **0.698** | **0.998** | **0.167** | **0.044** | **0.004** | **0.004** |
| PerturbDiff (reported) | 0.190 | 0.174 | 0.623 | 0.702 | 0.343 | 0.210 | 0.625 | 0.340 | 0.015 | 0.081 | 0.690 | 0.700 | 0.575 | 0.984 | 0.297 | 0.059 | 0.008 | 0.007 |

The model predicts both a denoised cell batch $B_\theta$ and a denoised KME vector $z_\theta$. We add two explicit RKHS losses:

$$\mathcal{L}_{\text{KME}} = \|z_\theta - \hat{\phi}_A(B_0)\|_2^2, \tag{60}$$

$$\mathcal{L}_{\text{align}} = \|z_\theta - \hat{\phi}_A(B_\theta)\|_2^2, \tag{61}$$

and train with

$$\mathcal{L}_{\text{total}}^{\text{RKHS}} = \mathcal{L}_{\text{total}} + \lambda_{\text{KME}}\mathcal{L}_{\text{KME}} + \lambda_{\text{align}}\mathcal{L}_{\text{align}}. \tag{62}$$

We select $\lambda_{\text{KME}}$ and $\lambda_{\text{align}}$ from $\{0.03, 0.1, 0.3\}$ and keep the sampling configuration unchanged.

Taken together, the Nyström variant provides a more explicit finite-dimensional realization of the RKHS KME, enables direct RKHS supervision, and improves most Replogle metrics. We keep the cell-space realization as the default method because it is substantially more scalable, while this experiment supports the practical value of the RKHS formulation beyond a purely interpretive derivation.

| $t$ | DEPrec | ES |
|------|--------|------|
| 10   | 0.15   | 0.53 |
| 50   | 0.17   | 0.60 |
| 100  | 0.17   | 0.62 |
| 1000 | 0.17   | 0.71 |

| $t$ | 1000 | 100 | 10 |
|----------|-------|-------|-------|
| Replogle | 5.4h  | 37min | 6min  |
| PBMC     | 36.8h | 3.6h  | 26min |

Sampling time by DDIM steps.

DEPrec and ES by DDIM steps.

| Method | #Params | Train | Mem | Sampling |
|--------|---------|-------|-----|----------|
| PerturbDiff | 111.0M | 14.5h (200k step, 4*V100) | 4.18GiB | 37.0min |
| STATE | 47.30M | 23.3h (80k step, 1*V100) | 3.1GiB | 2.1min |
| Linear | 2.30M | 47.0s (1*V100) | 108MiB | 61.8s |

Training and inference cost.

*Figure 25.* Sampling-step and computational-cost analysis. We use DDIM with $K = 100$ steps by default, which provides a practical trade-off between inference cost and predictive performance.

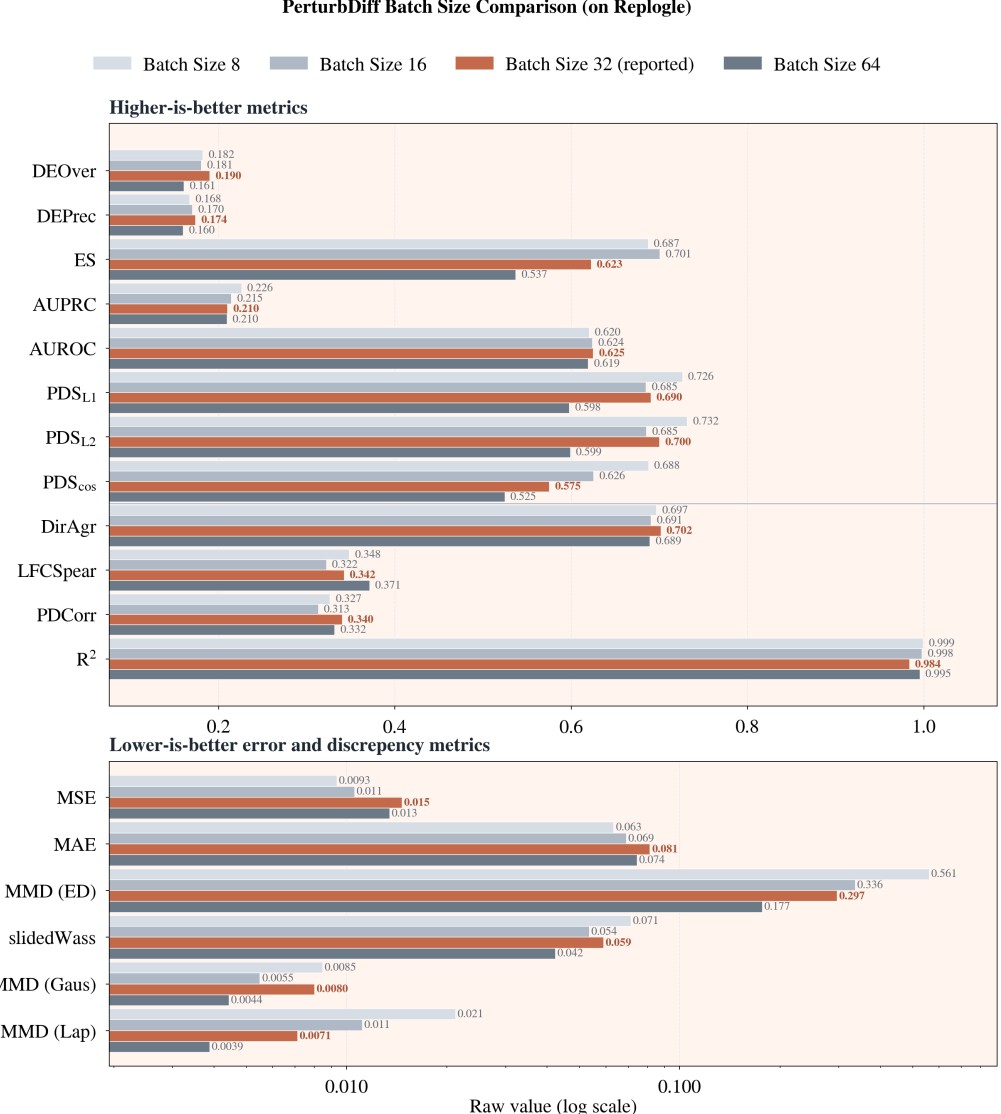

**Figure: Effect of batch size on PerturbDiff performance (Replogle, trained from scratch).** The top panel reports higher-is-better metrics, and the bottom panel shows lower-is-better error and distributional discrepancy metrics (log scale). Batch size 32 corresponds to the reported configuration and is highlighted in orange; other batch sizes (8, 16, 64) are shown in gray. All other training settings are kept identical. We did not tune the batch size and used the same value for both training and sampling. While it does affect performance, **the impact is metric-dependent**: larger batch sizes improve distributional metrics (e.g., ED), DEOver peaks at a moderate size (32), and metrics like AUROC and DirAgr show no clear trend. **Overall, no single batch size consistently optimizes all metrics.**

*Figure 26.* Batch-size sensitivity on Replogle. The default batch size is 32. Batch size changes the empirical distribution estimator and MMD computation, leading to metric-dependent trade-offs.

**PBMC metric comparison across random seeds**

**Figure: PBMC evaluation metrics across four random seeds.** The top panel reports higher-is-better metrics, while the bottom panel shows lower-is-better error and distributional distance metrics (log scale). Seed 42 corresponds to the reported run and is highlighted in orange; other seeds (0, 3407, 233333) are shown in gray, with dashed lines indicating the mean across seeds. Overall, PerturbDiff demonstrates **rather consistent performance** across random initializations, with limited variance across most metrics.

*Figure 27.* Seed-wise variability analysis. Repeated runs show that the qualitative performance trends are stable despite the stochasticity of diffusion training and sampling.

