# OpenReview forum: "PerturbDiff: Functional Diffusion for Single-Cell Perturbation Modeling"
_ICML.cc/2026/Conference — ICML 2026 regular_

### Official Review · Reviewer_rvRj · 2026-03-04

**Soundness:** 4
**Presentation:** 3
**Significance:** 4
**Originality:** 4
**Overall Recommendation:** 5
**Confidence:** 4

**Summary:**

This paper solves the problem of single-cell perturbation prediction: predicting gene expression of a single cell population under perturbation.

Instead of working on the space of gene expressions, the method works on the space of probability distributions of gene expressions, characterized by a reproducing kernel Hilbert space (RKHS) on which the induced norm is the Maximum Mean Discrepancy (MMD).

The paper then defines a diffusion model with noising and denoising processes tailored for the RKHS and uses MMD for the loss.

They also introduce a pretraining process on large unlabeled data that was shown to improve the performance.

They benchmarked the method against strong and up-to-date baselines on three large-scale datasets and showed strong performance. Ablation study was also conducted to justify design choices.

**Compliance With Llm Reviewing Policy:**

Affirmed.

**Final Justification:**

The authors addressed my original concerns and follow-up questions adequately, so I will maintaining my score.

**Key Questions For Authors:**

1. Is the model’s output is a function in the RKHS, parameterized by a vector? How to recover the predicted populations, i.e. actual cells? Is it through an additional sampling process? If so, how is the uncertainty introduced by this process handled in the evaluation? Is it computationally efficient to sample these high dimensional samples of gene expression?
2. can you explain a bit more on how the kernel is chosen for this RHKS, and the motivation for that choice. Does it scale for the high-dimensional gene input? Does it suffer from the curse of dimensionality?
3. Suppose I am interested in not only the changes of distributions, but also the transport, for example, how will a certain cell or a subpopulation of cells will react to perturbation, is the method able to do it, or is it a limitation? I guess this is not possible because the method operates on the space of distributions and uses MM-DiT from noise to data, but please explain. For example, a diffusion model learning the transport from unperturbed data to perturbed data, such as CellFlow among the baselines, would be able to answer these questions.
4. Is it possible to also evaluate the methods on population metics, such as EMD and MMD/Wasserstein distance? Cell average methods wouldn’t be able to show the effectiveness of single-cell level modeling, while the DEG metrics are not directly evaluating how the predicted populations match the ground truth.
5. The results are presented with single numbers without uncertainty quantification (e.g. stdev over a few runs). Can you add that uncertainty or justify why it was not presented?

**Limitations:**

Yes

**Strengths And Weaknesses:**

**Soundness**

The paper is technically sound, with well motivated method design, rigorous theory and proofs, and extensive ablation studies to justify the design choices. The results are presented with single numbers without uncertainty quantification (e.g. stdev over a few runs).

**Presentation**

The paper is well written, with clear explanation of the method, results, and a through overview of related works. The paper explains the architectures and details clearly, but did not provide source code.

**Significance**

The paper addresses the perturbation prediction problem, which is a hot topic recently and has promise for improving biology understanding and benefit drug design.

**Originality**

The viewpoint of modeling populations instead of individual cells is original, and the use of RKHS and the diffusion models on the RKHS are novel not just in concept but also in the detailed derivation and execution.

---

> ### Author Rebuttal · Authors · 2026-03-31
>
> Thank you for recognizing the paper as technically sound, with well-motivated design, strong performance against baselines, and supporting ablations. We also appreciate your positive feedback on clarity, significance, and the originality of population-level RKHS-based diffusion. We respond to your questions below
>
> > R1: What’s the model’s output? How are generated cells sampled?
>
> Our contribution is a functional diffusion framework with a tractable cell-space realization. **We distinguish theory from implementation**. **In theory**, PerturbDiff defines diffusion over **function-valued random variables** represented by kernel mean embeddings $\mu_{c,\tau}$ and $\mu_c$ in an RKHS $H_k$; the forward process adds Gaussian random elements (“noise”) in $H_k$, and the reverse model predicts a denoised embedding $\mu_\theta(\mu_t, t, \mu_c, c, \tau)$. **In practice**, we do not implement $\mu_{t}$ and $\mu_c$ as an infinite-dimensional object, due to the intractable dimension. Both training and sampling are done in cell space with empirical cell batches.
>
> A perturbed batch $B_{pert}$ is the Monte-Carlo estimate of an empirical distribution, which theoretically induces $\mu_{c,\tau}$; a control batch $B_{ctrl}$ similarly induces $\mu_{c}$. Hence the model outputs a generated perturbed cell batch, not an explicit RKHS function.
>
> Inference also runs directly in cell space, with no extra RKHS-to-cell step. Starting from Gaussian noise ($B_T\sim\mathcal N(0,I)$), the model runs reverse DDIM conditioned on the matched control batch and metadata $(c,\tau)$. The denoiser is an $x_0$-predictor, and the final denoised batch is returned as the generated perturbed population.
>
> Appendix B.1.1–B.1.7 formalizes this theory-to-implementation bridge with rigorous proofs. More proofing details for the tractable cell-space realization can be found in App. B.1.7.
>
> We further provide the pseudocode for both real implementation and the corresponding equivalent logics in RKHS for both training and sampling in https://anonymous.4open.science/r/rebfig-1F63/pseudocode.pdf
>
> > R2: How is uncertainty handled in sampling?
>
> Uncertainty comes only from the initial Gaussian noise. With DDIM ($\eta=0$), sampling is deterministic given that noise, and diversity comes from resampling the initial noise/random seed. One can also use stochastic sampling.
>
> > R3: Uncertainty measures across random seeds
>
> We add the results in https://anonymous.4open.science/r/rebfig-1F63/random_seed.png. We did not initially include repeated-seed statistics mainly because diffusion training is expensive, especially on very large datasets such as Tahoe100M (100M cells). Still, we agree that seed-wise variability is informative and will add it where feasible.
>
> > R4: Is it efficient for high-dimensional gene expression?
>
> As explained in **R1 rvRj**, the method avoids explicit infinite-dimensional computation and works directly on empirical batches in cell space, so it remains tractable for high-dimensional gene expression. Computational cost is discussed in **R9 pao9**
>
> > R5: Kernel choice, motivation, and scalability in high dimensions.
>
> We use the energy distance kernel (GeomLoss; Eq. (32)) because (1) it is simple and needs no bandwidth hyperparameter, unlike Gaussian/RBF kernels; and (2) it matches our theory by ensuring existence of the KME (App. B.1.2), while differentiability with bounded first-order derivative is exactly the assumption needed for our tractable approximation of Gaussian random elements (App. B.1.7).
>
> If the gene-expression dimension is |G| and batch size is |B|, the cost scales as O(|B|^2|G|), which is manageable for |G|=2000 and |B|=256 or 32 in our case.
>
> > R6: Can PerturbDiff model transport between cell subpopulations?
>
> This is a limitation of PerturbDiff, and we will clarify it. PerturbDiff learns diffusion from Gaussian noise to the data distribution, while control distributions are only conditioning information. A possible direction is to combine the same “distribution as random variable” idea with flow matching instead of diffusion. But this would require careful proofs for the continuity equation, transition path, and velocity field. We have not tried this and are unsure whether it works theoretically.
>
> > R7: Add more discrepancy measures like energy distance, MMD and Wasserstein distance
>
>
> Thanks for the suggestion. We have added more metrics and updated results in all the newly added rebuttal links. From-scratch downstream performance for Tahoe100M are shown below:
> |Method|EnergyDistance(ED)|SlicedWasserstein (slicedWass)|MMD(Gaussian)|MMD(Laplasion)|
> |-|-|-|-|-|
> |STATE|0.6177|0.0693|0.0269|0.0250|
> |PerturbDiff (from scratch)|0.0681|0.0216|0.0033|0.0025|
>
> Still, we focused on DE-related metrics because they directly reflect biological relevance, and it's hard to set a threshold for discrepancy measures to tell if it's good already or too good.
>
> > R8: provide source code
>
> See https://anonymous.4open.science/r/Anouymous_PerturbDiff-28F7

---

> > ### Author Rebuttal · Reviewer_rvRj · 2026-04-01
> >
> > Thank you for the rebuttal!
> >
> > My concerns in R2-R8 have been addressed.
> >
> > However, after reading the clarification on R1, I have a new question. Could you clarify if my following understanding is correct: The paper develops the perspective of sampling in the space of distributions, with RKHS, and theory of diffusion models in RKHS, however, the actual computation is a standard diffusion model with the MMD loss and DiT.
> > If this is correct, I have new concerns on the novelty of this paper. The theoretical novelty is there, but the computational novelty is limited, since both DDIM and MMD loss (as in STATE) have been established before.

---

> > > ### Author Response · Authors · 2026-04-05
> > >
> > > We thank the reviewer for the thoughtful follow-up question. We clarify this point below.
> > >
> > > **First**, we thank the reviewer for recognizing the **novelty of our theoretical formulation**. The key shift in our work is to **model cell populations as distribution-valued random variables** and define diffusion over their RKHS kernel mean embeddings (KME), rather than model individual cells as random variables as in prior works (Squidiff in Sec. 5). This is the core of our work and drives the theory, training objective, and implementation.
> > >
> > > **Second**, although the diffusion is defined in RKHS, **it has to be tractably realized as RKHS objects are infinite-dimensional and intractable**. Thus, we use a theoretically grounded cell-space realization, using standard components like DDIM sampling and MM-DiT denoiser, where:
> > >   * KME is estimated via **Monte Carlo over cell batches**
> > >   * Gaussian random elements are approximated by **additive Gaussian noise under first-order kernel assumptions**
> > >     (Sec 4.1–4.7, App B.1.1–B.1.7, App B.2)
> > >   * Exact loss equivalence induces MMD, which is not interchangeable, while STATE's MMD is in principle a replaceable population matching loss
> > >
> > > **Third**, beyond the theoretical novelty and RKHS-grounded realization, PerturbDiff also contributes:
> > > * **marginal pretraining** on single-cell data, enabling non-trivial zero-shot generalization and improving from-scratch performance
> > > * **strong empirical performance** across datasets and metrics for many perturbations (drug, cytokine, genetic)
> > > * detect a **failure mode** of the prior SoTA STATE: over-prediction of differential expressed genes with poor ES
> > > * **scaling** study: perturbation modeling benefits from **moderate model capacity and compute** rather than aggressive scaling
> > >
> > > **Finally**, inspired by the reviewer's comment, **to test whether the RKHS formulation has more computational value beyond the base cell-space realization**, we **implemented an explicit RKHS variant**. The results show that **making RKHS KME more explicit via Nystrom KME approximation can further improve performance**. But this variant is too computationally expensive for big data like Tahoe100M and PBMC, due to PCA compute and careful loss weight grid search. We detail this variant and its results below.
> > > ## 1) Explicit RKHS via Nyström KME
> > > We use a **finite-rank Nyström KME** (https://arxiv.org/abs/2201.13055), as a proxy for infinite-dimensional RKHS KME. The idea is to build a **finite-dimensional RKHS coordinate system** using anchor cells:$$\Phi(X) \in \mathbb{R}^R$$ Each dimension measures similarity to one anchor (R anchors in total)
> > > ## 2)Anchor Cells
> > > Anchors are selected via:
> > > 1) Project training&validation cells into PCA space
> > > 2) Group by metadata (cell line/batch)
> > > 3) Allocate the anchor budget for group size
> > > 4) Run k-means in each group
> > > 5) Select the nearest real cell to each cluster center
> > >
> > > Nyström anchors are **real, diverse cells** that cover the data
> > > ## 3) Nyström KME
> > > For a cell set $X$={$x_1,...,x_S$}, Nyström KME is $$\Phi_r(X) = \frac{1}{S} \sum_{s=1}^S k(x_s, z_r)$$
> > > with energy kernel$$k(x, z)=\frac{1}{2}(|x|_2+|z|_2-|x-z|_2)$$
> > > ## 4) Model
> > > We extend an RKHS branch:
> > >
> > > **Input**
> > > * Noisy perturbed cell $x_t$, with clean version $x_0=x_{pert}\in X_{pert}$
> > > * Control cell $x_{ctrl}\in X_{ctrl}$
> > > * Noisy RKHS KME $u_t$, with clean version $u_0=u_{pert}:=\Phi(X_{pert})$
> > > * Control RKHS KME $u_{ctrl}:=\Phi(X_{ctrl})$
> > >
> > > **Pipeline**
> > > * $u_t$, $u_{ctrl}$ are encoded as batch-global **tokens** and prepended to cell tokens $x_t$ and $x_{ctrl}$, respectively
> > >
> > > **Output**
> > > * Denoised cell $\hat{x}$
> > > * Denoised KME $\hat{u}$
> > >
> > > ## 5) Loss
> > > We design **two RKHS losses**:
> > >
> > > (1) Train the model to predict the true KME:
> > > $$L_{head}=\frac{1}{R}\sum_r(\hat{u}_r-u_r)^2$$
> > >
> > > (2) Align
> > > * RKHS prediction $\hat{u}$
> > > * RKHS embedding of generated cells
> > >
> > > Let $\hat{X}=${$\hat{x}_1,..., \hat{x}_S$},
> > >
> > > $$L_{cons}=\frac{1}{R} \sum_r (\Phi_r(\hat{X}) -\mathrm{sg}( \hat{u}_{r} ))^2$$
> > >
> > > **Final Loss**
> > > $$L = L_{base} + \lambda_{head} L_{head}+\lambda_{cons} L_{cons}$$
> > >
> > > ## 6)Replogle Result
> > > * $\lambda_{head}=0.1$, $\lambda_{cons}=0.1$, both selected via grid search: {0.03, 0.1, 0.3}
> > > * Sampling config is unchanged
> > >
> > > |Method|DEOver|DEPrec|ES|DirAgr|LFCSpear|AUPRC|AUROC|PDCorr|MSE|MAE|PDS L1|PDS L2|PDScos|R2|MMD (ED)|slicedWass|MMD (Gaus)|MMD (Lap)|
> > > |-|-|-|-|-|-|-|-|-|-|-|-|-|-|-|-|-|-|-|
> > > |**PerturbDiff + RKHS**|0.202|0.187|0.722|0.722|0.380|0.244|0.645|0.352|0.013|0.074|0.764|0.771|0.698|0.998|0.167|0.044|0.004|0.004|
> > > |**PerturbDiff (reported)**|0.190|0.174|0.623|0.702|0.343|0.210|0.625|0.340|0.015|0.081|0.690|0.700|0.575|0.984|0.297|0.059|0.008|0.007|
> > >
> > > ## 7) Takeaway
> > > These results suggest that the RKHS formulation is not merely interpretive: beyond the base cell-space realization, **making KME more explicit in the model can further improve performance**.
> > > * Nyström KME adds a **tractable realization of RKHS**
> > > * It enables **explicit RKHS supervision**
> > > * It improves performance **across most metrics**

---

### Official Review · Reviewer_pao9 · 2026-03-09

**Soundness:** 2
**Presentation:** 2
**Significance:** 2
**Originality:** 2
**Overall Recommendation:** 2
**Confidence:** 4

**Summary:**

The paper proposes PerturbDiff, a diffusion framework that models single-cell perturbation responses at the level of distributions rather than individual cells. Distributions of cells are embedded into a reproducing kernel Hilbert space (RKHS) via kernel mean embeddings, a forward diffusion is defined in this function space, and training reduces to aligning predicted and observed batches with a principled MMD objective while conditioning on control populations and covariates. Empirically, the method is evaluated on three large-scale benchmarks and, together with a marginal pretraining strategy using unperturbed atlases, reports strong performance on a broad suite of metrics, particularly for differential expression recovery.

**Compliance With Llm Reviewing Policy:**

Affirmed.

**Key Questions For Authors:**

1. How exactly is $\mu_t$ instantiated in practice, and what goes into the denoising network at each timestep? Please provide pseudocode for a single training step, clarifying the roles of noised batches, $\mu_c$, and conditioning tokens.

2. What kernel(s) are used for MMD, how are bandwidths selected, and how sensitive are results to these choices? Did you consider multi-kernel MMD or learning kernel parameters?

3. Can you include an ablation that removes the diffusion/time-conditioning but keeps the same backbone and MMD loss (i.e., a direct conditional generator trained with MMD), to isolate the contribution of diffusion?

4. How is the “distribution over distributions” claim evaluated? Can you show that multiple samples for a fixed $(c, \tau)$ reproduce batch-to-batch variability observed empirically, with calibration metrics and uncertainty quantification?

5. The figures show anomalies (e.g., negative AUPRC; many values at 1.00; identical values across variants). Are these normalized or clipped scores? Please provide raw numeric tables (with variance over seeds) and fix plotting issues.

6. What is the computational cost (training/inference time, GPU memory) relative to STATE and Linear? How many sampling steps are used, and how does reducing steps affect performance?

7. How do you handle gene vocabulary mismatches between pretraining and downstream (especially for Replogle with ~45% overlap)? Is there a learned projection, imputation, or masking strategy?

8. How many predicted cells per condition are generated for evaluation, and how does batch size influence MMD loss and metric stability?

9. Could you comment on potential overfitting or memorization when pretraining on massive corpora that include related contexts? What safeguards are in place to avoid leakage across splits?

10. Do results hold under alternative discrepancy measures (e.g., energy distance, sliced Wasserstein) or with MMD computed on learned embeddings rather than raw counts?

**Limitations:**

**Technical limitations or concerns**

- The functional diffusion in RKHS appears partially aspirational: training and sampling are ultimately carried out in cell space with an MMD loss; the paper approximates the forward RKHS noise by noising cells and recomputing embeddings, and sampling is performed via DDIM in cell space. This undermines how much of the infinite-dimensional diffusion is truly realized beyond motivating the loss function.
- The core training objective reduces to MMD between batches, which brings PerturbDiff close in spirit to STATE; the incremental gain from the diffusion machinery versus “simply” training a conditional generator with MMD is not fully disentangled.
- The “distribution over distributions” claim is not directly evaluated; no experiments demonstrate calibrated diversity across replicates or latent-factor-induced variability for the same ($c, \tau$).

**Experimental gaps or methodological issues**

- Some figures exhibit questionable values (e.g., a negative AUPRC in a radar chart; many metrics saturating at 1.0), suggesting normalization or plotting artifacts; raw unnormalized metrics, confidence intervals, and significance tests are missing.
- Kernel choice for MMD (type, bandwidth, multi-kernel setup) is not specified in the main text, nor is a sensitivity analysis provided; MMD performance heavily depends on this.
- No runtime, memory, or computational cost comparisons are provided; diffusion architectures can be substantially heavier than baselines.
- Missing baselines that are close in spirit: conditional mean embedding regression, MMD-GANs, or distribution-regression/deep-set approaches that also operate on batches/distributions.

**Clarity or presentation issues**

- The relationship and practical transitions between RKHS diffusion variables ($\mu_t$) and the actual inputs to the DiT (noised cell batches) are not fully spelled out with pseudocode/equations; the role of $\mu_t$ in the implemented training loop remains opaque.
- The mapping between pretraining and downstream gene vocabularies (particularly for ~45% overlap in Replogle) needs more concrete detail on how missing genes are handled in fine-tuning and inference.

**Missing related work or comparisons**

- Prior work on conditional mean embeddings, distribution regression, and MMD-based generative models (e.g., MMD-GAN, kernel herding, deep sets for distribution-to-distribution mappings) is not adequately discussed or compared, despite conceptual proximity.

**Strengths And Weaknesses:**

**Technical novelty and innovation**

- The shift from cell-level random variables to distribution-valued random variables is a clear conceptual innovation for unpaired single-cell perturbation modeling.
- Using kernel mean embeddings to place distributions in an RKHS provides a clean pathway to define a “functional” diffusion, and the equivalence between squared RKHS distance and MMD naturally yields a distribution-aware training objective.
- The Multi-Modal DiT architecture that jointly attends to control and perturbed batches is a sensible design to capture structured deviations under perturbation.
- The marginal pretraining strategy leveraging large unperturbed single-cell corpora is well-motivated and addresses data scarcity.

**Experimental rigor and validation**

- Broad evaluation across three challenging datasets representing signaling, drugs, and genetic perturbations, with many metrics (including DE-focused metrics) and per-perturbation analyses.
- Ablation removing the MMD term demonstrates that distribution-aware training is pivotal, especially for DE-centric metrics.
- Scaling and low-data studies provide additional insight into model behavior and data efficiency.

**Clarity of presentation**

- The high-level motivation, connection to unobserved latent factors, and the RKHS formulation are explained clearly.
- The link from RKHS geometry to MMD provides a coherent story for why the loss aligns with the goals of population-level modeling.

**Significance of contributions**

- Perturbation modeling is an important problem with growing impact in systems biology and drug discovery.
- If borne out, a principled, distribution-aware approach that generalizes across contexts and improves DE recovery would be of substantial interest to the ICML community.

---

> ### Author Rebuttal · Authors · 2026-03-31
>
> Thank you for recognizing the novel idea of modeling perturbation response as distribution-valued random variables, the RKHS framework, MM-DiT design, and marginal pretraining. We appreciate your positive feedback on the broad evaluation (ablations, scaling, low-data). We address the issues below
>
> > R1: Figure anomaly and raw value
>
> **Raw values are in Tab. 4 in paper**. There is no "negative AUPRC", "many values at 1", or "identical values across variants". The anomaly comes from each axis in the radar plot being scaled separately, so the same ring does not mean the same value. We'll clarify
>
> > R2: Variance across seeds
>
> See **R3 rvRj**
>
> > R3: realization of $\mu_t, \mu_c$ and training
>
> See **R1 rvRj**
>
> > R4: RKHS diffusion vs. tractable cell-space realization
>
> We **do not directly operate on infinite-dimensional RKHS objects due to intractability**. RKHS instead gives the derivation: empirically, the denoising reduces to MMD between cell batches, and the Gaussian random element is approximated in cell space by additive Gaussian noise with the first-order approximation in App. B.1.7. Hence RKHS diffusion grounds the practical objective and noising scheme
>
> > R5: MMD makes PerturbDiff close to STATE
>
> Our novelty is **not** MMD alone. In STATE, MMD is a model-agnostic, replaceable population-matching loss (Sec. 4). In PerturbDiff, MMD is the empirical form of RKHS denoising objective and not interchangeable
>
> > R6: “no-diffusion/same-backbone/same-MMD” ablation
>
> As shown in https://anonymous.4open.science/r/rebfig-1F63/no_diffusion.png, **without diffusion, ES metric becomes negative**, indicating failure to generate biologically meaningful cells. By Eqn. (58), negative ES means the predicted DEG ranking is inversely correlated with the ground truth, i.e., failing to capture true DE patterns. STATE also fails on ES (Tab. 4), as explained in Fig. 5
>
> > R7: kernel choice and hyperparameter effects. Add multi-kernel MMD or learning kernel parameters
>
> See **R5 rvRj** for kernel choice. Gaussian, Laplacian, and simple multi-kernel MMD (0.5* Gaussian+0.5* Laplasion) satisfy the theory. Results in https://anonymous.4open.science/r/rebfig-1F63/kernel.png show bandwidth sensitivity. Learning kernel parameters via meta-learning is beyond our focus
>
> > R8: "distribution over distribution" claim
>
> This is a **key motivation and part of RKHS derivation**. Treating distributions as random variables enables theoretical distributional variability. We did not test this directly: benchmarks **lack latent-factor labels and RKHS objects are infinite-dimensional and not computable/visualizable**. Evidence is thus indirect via stronger downstream performance, especially DE metrics
>
> As a proxy, we remove experimental batch labels from conditioning and test whether generated samples still recover batch info in https://anonymous.4open.science/r/rebfig-1F63/remove_batch.png. This compares which method (PerturbDiff/STATE) better captures batch variability: STATE collapses into a tiny region in PCA (near-zero variance, chance-level separability), while PerturbFlow preserves the batch-dependent spread in real data
>
> > R9: Sampling steps / Runtime / Memory
>
> We use DDIM with t=100, faster than full sampling t=1000 with limited performance loss. See details and computational cost in https://anonymous.4open.science/r/rebfig-1F63/cost.png
>
> > R10: Gene-vocabulary mismatch between pretraining and downstream
>
> Pretraining uses a unified 12626-gene vocabulary (App. 3.2), with missing genes zero-imputed. In fine-tuning, we use dataset-specific transfer for input/output projections (App. C.1). For PBMC, we fine-tune and sample in the full 12626-gene space, then keep the 2000 HVGs. For Replogle, we reinitialize projections and do the same. For Tahoe100M, we replace the 12626-dim projections with 2000-dim ones due to memory limits
>
> > R11: Per-condition batch size and effects
>
> Each batch shares $(c,\tau)$. We add Replogle batch-size study in https://anonymous.4open.science/r/rebfig-1F63/batch_size.png. For Replogle, bsz=32 is used in training and sampling. Batch size affects performance, with no single best value across metrics
>
> > R12: Leakage
>
> To avoid leakage, perturbation datasets keep the same train/validation/test split for pretraining, and pretraining uses only the train split
>
> > R13: Discrepancy metrics on raw counts or embeddings
>
> See example results in **R7 rvRj**. We focused on DE metrics as they directly reflect biological relevance, while discrepancy metrics lack clear thresholds for interpretation
>
> PerturbDiff operates in expression space and avoids encoder-decoder information loss, so embedding-space evaluation is less central and mainly relevant to latent-space methods like CellFlow
>
> > R14: More baselines: MMD-GAN, conditional mean embedding regression, and distribution-regression/deep-set methods
>
> We'll expand related work, though some of them are not for perturbation modeling and not SoTA for this task

---

> > ### Author Rebuttal · Reviewer_pao9 · 2026-04-03
> >
> > The rebuttal seems to miss the opportunity to meaningfully engage with the points I raised, therefore I will keep my scores.

---

> > > ### Author Response · Authors · 2026-04-05
> > >
> > > Dear Reviewer,
> > >
> > > Thank you for your detailed review and thoughtful questions. **During Rebuttal Stage 1, we made our best effort to address your concerns point by point and to provide as much supporting experimental evidence as was feasible within the rebuttal window**.
> > >
> > > Specifically, we:
> > > - clarified the concern regarding the figure anomaly and pointed to the raw values already reported in the paper [**R1 pao9**];
> > > - added random-seed variance analyses [**R2 pao9 / R3 rvRj**, https://anonymous.4open.science/r/rebfig-1F63/random_seed.png];
> > > - included pseudocode for both training and sampling [**R3 pao9 / R1 rvRj**, https://anonymous.4open.science/r/rebfig-1F63/pseudocode.pdf];
> > > - clarified the difference between STATE and PerturbDiff [**R5 pao9**; see also **Sec. 4** of the paper];
> > > - added a no-diffusion ablation [**R6 pao9**, https://anonymous.4open.science/r/rebfig-1F63/no_diffusion.png];
> > > - reported kernel choices and sensitivity analyses [**R7 pao9**, https://anonymous.4open.science/r/rebfig-1F63/kernel.png];
> > > - provided runtime, memory, and sampling-step details [**R9 pao9**, https://anonymous.4open.science/r/rebfig-1F63/cost.png];
> > > - clarified gene-vocabulary transfer [**R10 pao9**], batch-size effects [**R11 pao9**], data-leakage safeguards [**R12 pao9**], and additional discrepancy metrics [**R13 pao9 / R7 rvRj**]; and
> > > - regarding the concern about modeling a “distribution over distributions,” provided a proxy experiment [**R8 pao9**, https://anonymous.4open.science/r/rebfig-1F63/remove_batch.png]. A fully direct experiment is computationally infeasible because the RKHS is infinite-dimensional, so its objects cannot be manipulated explicitly in a straightforward way [**R4 pao9, R8 pao9**].
> > >
> > > **In Rebuttal Stage 2**, motivated by a follow-up question from reviewer **rvRj** raised during the discussion, **we further investigated whether the RKHS formulation offers more practical value beyond the base cell-space realization**. To test this, we implemented a more explicit RKHS variant using a Nyström approximation to the KME. **The results suggest that making the RKHS KME more explicit can further improve performance**. However, this variant is **computationally prohibitive for large datasets** such as Tahoe100M and PBMC because of the additional PCA cost and the need for careful loss-weight grid search. We provide details and results for this variant in the response to **rvRj**.
> > >
> > > **We also appreciate your recognition of several strengths of our work**, including: (1) the conceptual novelty, (2) the principled RKHS formulation, (3) the sensible MM-DiT design, (4) the broad evaluation across datasets and metrics, (5) the useful ablations and scaling studies, (6) the clear presentation, and (7) the potential significance to the community.
> > >
> > > **To restate our contributions as clearly as possible, we would like to emphasize three points**.
> > >
> > > First, the core **conceptual contribution of our work is to model cell populations as distribution-valued random variables** and to define diffusion over their RKHS kernel mean embeddings (KMEs), rather than modeling individual cells as random variables as in prior work such as Squidiff (Sec. 5). This shift is central to our theory, training objective, and implementation.
> > >
> > > Second, although the diffusion process is defined in RKHS, RKHS objects are infinite-dimensional and therefore not directly tractable in practice. We thus **develop a theoretically grounded realization in cell space**, implemented with standard components such as DDIM sampling and an MM-DiT denoiser. In particular, the KME is estimated via Monte Carlo over cell batches, and Gaussian random elements are approximated through additive Gaussian noise under first-order kernel assumptions (Sec. 4.1–4.7; App. B.1.1–B.1.7; App. B.2).
> > >
> > > Third, beyond the RKHS-based formulation and its tractable realization, PerturbDiff **empirically contributes**:
> > > - an **MM-DiT** architecture that jointly models perturbed and control cells;
> > > - **marginal pretraining** on single-cell data, which enables non-trivial zero-shot generalization and improves performance over training from scratch;
> > > - **strong empirical performance** across datasets, metrics, and perturbation types (drug, cytokine, and genetic);
> > > - identification of a **failure mode in the prior state-of-the-art method STATE**, namely over-prediction of differentially expressed genes, which leads to poor ES; and
> > > - a **scaling study** showing that perturbation modeling benefits from moderate model capacity and compute, rather than aggressive scaling.
> > >
> > > To help us address any remaining issues as precisely as possible, we would greatly appreciate more specific guidance on **which concerns still remain unresolved**, and **why the current rebuttal does not yet address them to your satisfaction**. We would welcome any further detail before the rebuttal period ends.

---

### Official Review · Reviewer_mDR4 · 2026-03-10

**Soundness:** 3
**Presentation:** 3
**Significance:** 3
**Originality:** 3
**Overall Recommendation:** 5
**Confidence:** 3

**Summary:**

Virtual cell models are a class of models that predict the effects of perturbations on a cell’s transcriptome. These models are typically trained on single-cell gene expression data. The same cells cannot be assayed before and after a perturbation due to the destructive nature of the sequencing experiment. Thus, we get a distinct population of control and perturbed cells. Most existing models assume a single distribution each for the control and perturbed cells. However, this ignores batch effects and other latent sources of variability that can lead to different cell distributions each time an experiment is performed, essentially leading to existing models marginalizing over these variables. To alleviate this problem and improve our ability to generalize, the authors propose a novel diffusion model called PerturbDiff that directly models distribution-level variability by learning a diffusion process over the space of cell distributions. Here, cell distributions are the random variables and they are represented using their kernel mean embeddings. In the forward diffusion process, Gaussian random elements are added to the kernel mean embedding of the perturbed cells based on a noise schedule. In the reverse process, the model is asked to predict the denoised kernel mean embedding of the perturbed cells given the kernel mean embedding of control cells, some conditioning information (cell type, perturbation info, etc.), and the timestep. The authors derive training and sampling methods that make the learning process tractable. Additionally, they also propose a pretraining step that trains the diffusion model on large single-cell datasets before fine-tuning it to model perturbation data. The authors compare PerturbDiff to many existing models including STATE, a popular recent virtual cell model. Overall, PerturbDiff outperforms or matches STATE on most datasets. It is also shown to be better at detecting differentially expressed genes compared to STATE, an important goal for perturbation experiments. Finally, the authors also illustrate the benefits of pretraining - largely improved performance, and better sample-efficiency.

**Compliance With Llm Reviewing Policy:**

Affirmed.

**Final Justification:**

All of my concerns were addressed in the rebuttal, including the unseen cell type evaluation which showed that PerturbDiff was better that STATE in this setting. Hence, I keep my recommendation to accept this paper.

**Key Questions For Authors:**

1. Why do the authors not benchmark their model for unseen cell type prediction? Could these benchmarks be performed during the rebuttal period?
2. How does the trained model perform on unseen perturbation prediction when prompted with an embedding representing an unseen perturbation?

**Limitations:**

yes

**Strengths And Weaknesses:**

Overall, I think this is a strong paper and recommend its acceptance – the method is well-motivated and novel, it largely outperforms existing methods, and is likely to be of interest to both ML researchers working on genomics and diffusion models. The paper is also well-written and is relatively easy to follow even though the methods are mathematically involved. I mention two other benchmarks below that could further strengthen this paper.

**Soundness**

The paper is technically sound and all empirical claims made by the authors are supported by their results. I was not able to fully verify the authors’ theoretical results but they seem broadly correct. The experiments are also well-designed and relevant baselines have been benchmarked. A few more benchmarking results could further strengthen the paper:
The paper seems to be missing benchmarks evaluating model performance on unseen cell types (State is evaluated in this setting to the best of my knowledge). This setting is a particularly important use case for virtual cell models. The authors could potentially use data from the Virtual Cell Challenge to further compare performance vs. many other existing methods.
I feel the description of the unseen perturbation prediction task is inadequate. Also, since the model uses perturbation embeddings, it could be evaluated on unseen perturbation prediction by prompting using embeddings for these perturbations.

**Presentation**

This submission is well-structured and is very clear. Prior work has been adequately cited and it is clear how this work builds on them. As mentioned above, the unseen perturbation prediction task’s description could be improved with more details on the exact methodology used to extract predictions and on metrics computation.

**Significance and Originality**

The diffusion-based methods presented in this work seem novel and could be of interest to both computational biologists and those working on diffusion models more generally. The methods are also well-motivated and it is clear how and where they improve on existing modelling paradigms. As virtual cell models are a very popular research area, this paper is likely to appeal to a wide audience and it broadly improves our understanding of the perturbation prediction task.

---

> ### Author Rebuttal · Authors · 2026-03-31
>
> Thank you for acknowledging our work as novel, well-motivated, technically sound, clearly written, largely outperforming existing methods, and broadly impactful across computational biology and diffusion modeling. We address the concerns below
>
> > R1: Unseen perturbation task setting
>
> Our current setting follows STATE's Figure 2 in their paper: it is a **unseen-context generalization setting** with partial perturbation coverage in the held-out-context, **rather than a strict unseen-perturbation-identity setting**, where a perturbation is completely absent from training. Concretely, 4 of the 12 donors are held out in PBMC, 5 cell lines are held-out in Tahoe100M and Hepg2 is held-out in Replogle. Within these held-out contexts, 30% of perturbations are included in training and the remaining 70% are used for testing.
> **Details are also provided in App. A.2.3**, with train/validation/test cell counts summarized **in Tab. 2(a)**. We’ll clarify it in the main text.
>
> > R2: Whether PerturbDiff supports unseen-perturbation-identity setting
>
> While possible in principle, the **strict unseen-perturbation-identity setting hasn’t been evaluated** in SoTA methods like STATE or CellFlow, likely because it is **highly challenging**. PerturbDiff can in principle support this setting via a pretrained covariate encoder. As described in App. B.3, PerturbDiff **can incorporate semantic perturbation representations** such as ESM2 protein embeddings for cytokines, ChemBERTa-style drug embeddings, and GenePT/LLM-derived gene embeddings from texts.  We agree  this is an important and challenging setting, and will explore in future work due to the rebuttal time limit
>
>
>
> > R3: Evaluation on unseen-cell-type setting
>
> The **unseen-cell-type setting** is reported in STATE's Figure 3 in their paper, where a cell line is hold out and all perturbations for that cell line are used for testing. We evaluated the unseen “hepg2” cell line in Replogle.
>
> As shown in https://anonymous.4open.science/r/rebfig-1F63/unseen_cellline.png, **PerturbDiff outperforms STATE for most metrics**. For both methods, we used each method’s original unseen-context hyperparameters as specified in their respective implementations, without any additional tuning.
>
> We did not include this setting in the main paper, as the unseen-context benchmark already provides broad and systematic coverage across datasets, metrics, and perturbation types.
>
> > R4: Add Virtual Cell Challenge (VCC) benchmark
>
> Thanks for the suggestion. We agree that the VCC is an important and increasingly influential benchmark for this community, and evaluating PerturbDiff on VCC would be valuable. A fair VCC comparison would require reproducing a substantially different preprocessing/evaluation pipeline and, in some winning solutions, hybrid statistical or pseudo-bulk features beyond the scope of our current setup; we therefore could not complete a complete benchmark within the rebuttal window.
>
>
> Based on the public description, VCC includes ~ 183k training cells across 150 gene perturbations (~1200 cells / perturbation), which is quite sparse, similar to Replogle. This also suggests that marginal pretraining may be important. We also note that the winning VCC solution combines deep learning with statistical or pseudo-bulk features, which would be interesting to evaluate in our setting. We'll explore in future work.
>
> > R5: Exact methodology for prediction extraction and metrics computation for the unseen perturbation task
>
>
> As explained in **R1 mDR4**, we used a unseen-context setting where partial perturbations are observable in the held-out cell lines.
>
> For each test condition ($c$=(cell type, experimental batch), $\tau$=perturbation type), we generate a batch of perturbed cells conditioned on the matched control population and covariates via **DDIM sampling**, package the generated cells into a predicted AnnData with matched labels and gene order, and evaluate it against the real AnnData using Cell-Eval v0.6.6 from STATE as evaluation protocol.
>
> Cell-Eval is also described in **Sec. 5.1 "Evaluation Metrics"**, and detailed in **App. C.2. "Metrics"**. Cell-Eval compares a predicted AnnData and a real AnnData object with the same perturbation labels, control label, and gene order. It reports two types of metrics: **(1) cell-level metrics**, which compare the averaged expressions between predicted and observed cell populations; and **(2) gene-level metrics**, which compare differential-expression patterns, including DEG sets derived from predictions and ground truth.
>
> An important detail is that although Cell-Eval itself does not automatically force the control cells in the predicted and real AnnData objects to be identical, we explicitly use the same control-cell set for both the predicted and ground-truth cells, and we **keep this control reference fixed across all compared baselines for fair comparison**.

---

> > ### Author Rebuttal · Reviewer_mDR4 · 2026-04-03
> >
> > Thank you for the detailed response! All of my concerns have been resolved and I will keep my rating. It would be great if the authors could add the unseen cell type evaluation to the final version of the paper since it is an important use-case for these models.

---

> > > ### Author Response · Authors · 2026-04-05
> > >
> > > Dear Reviewer,
> > >
> > > Thanks so much for your support on our project! In the final version, we will include the unseen cell type experiment results and discussions in appendix according to your great suggestions.
> > >
> > >
> > > Best,
> > > Authors

---

### Decision · Program_Chairs · 2026-04-30

**Decision:**

Accept (regular)

**Comment:**

Reviewers largely agree that this paper represents a well-motivated and novel approach, with an extensive empirical evaluation that supports the claims of the paper. During the rebuttal period, several reviewers asked for follow-up experiments or additional ablations, which the authors performed, alleviating many of the reviewer's concerns. I highly encourage the authors to incorporate these new results in an updated version of the paper. While one reviewer maintains their final score of (2=reject), the specific concerns raised in their review appear to be successfully addressed by the author's rebuttal. From my own read of the paper, I do not share these concerns, or believe that they are fully addressed by the rebuttal. Therefore, the paper is suitable for acceptance.